



# Constraining the soil carbon source to cave-air $CO_2$: evidence from the high-time resolution monitoring soil $CO_2$, cave-air $CO_2$ and its $\delta^{13}C$ in Xueyudong, Southwest China

Min Cao, Yongjun Jiang*, Jiaqi Lei, Qiufang He, Jiaxin Fan, Ze Zeng

Chongqing Key Laboratory of Karst Environment & School of Geographical Sciences, Southwest University, Chongqing 400715, China

*Correspondence to: Yongjun Jiang (jiangjyj@swu.edu.cn), +86 023-68253446*

**Abstract.** Cave $CO_2$ plays an important role in carbon cycle in a karst system, which also largely influences the formation of speleothems in caves. The partial pressure of $CO_2$ ($pCO_2$) of the cave air and cave water (cave stream and drip water) in Xueyu

Cave was monitored from 2015 to 2016. The $pCO_2$ for cave air and stream over two years showed very similar variations in seasonal patterns, with fluctuated high $CO_2$ concentrations in the wet season and steady low $CO_2$ concentrations in the dry season. Soil $CO_2$ which is largely controlled by soil temperature and soil water content as well as stream degassing are main origins for the Xueyu cave air $pCO_2$. The average values of $\delta^{13}C_{soil}$, $\delta^{13}C_{DIC}$ in June were -23.9‰ and -13.4‰, respectively; $\delta^{13}C_{CO2}$ of atmospheric air was -10.0‰ and $\delta^{13}C_{CO2}$ of cave air was -23.3‰. The average values of $\delta^{13}C_{soil}$, $\delta^{13}C_{DIC}$ in November

were -18.0‰ and -12.2‰, respectively; $\delta^{13}C_{CO2}$ of atmospheric air was -9.6‰ and $\delta^{13}C_{CO2}$ of cave air was -18.8‰. Moreover, the contribution from soil $CO_2$ is higher in June (78.8%) than in November (67.1%) based on the model of carbon stable isotopes. The contribution of C from the soil was larger in summer than in winter. The very similar (negative) values of carbon isotopes between soil and cave air $CO_2$ suggests that there were no potential geological/deeper sources with more positive $\delta^{13}C_{CO2}$. Stream $pCO_2$ degases from upper stream to downstream in the cave, resulting in slightly decreased $pCO_2$ but increased

carbon isotope values in the downstream. The influence of these regional controls on stalagmite records requires a better understanding of modern interaction between cave $CO_2$ sources, transport paths and mechanisms.

## 1 Introduction

In karst regions, carbon dioxide ($CO_2$) concentrations in epikarst (especially from soils) largely affect karst landscapes (Ford and Williams, 2007). Shallow caves are widely distributed in the terrestrial environment and contain a significant volume of

underground air with high concentrations of $CO_2$ (Wood, 1985; Faimon *et al.*, 2006; Bourges *et al.*, 2014). $CO_2$ concentrations in temperate karst soils range from 1000 to 15000 ppm, always with higher values in summer and lower values in winter (Spötl *et al.*, 2005; Frisia *et al.*, 2011). The $CO_2$ inputs that penetrate caves and become part of the karstic atmosphere via directly in gaseous form, dissolved $CO_2$ in infiltrated waters from the soil matter (Wood, 1985; Baldini *et al.*, 2006; Cuezva *et al.*, 2011). Caves with low $CO_2$ concentrations are considered as better ventilated caves (Šebela and Turk, 2011; Bourges *et al.*, 2014). In





shallow or ventilated caves $CO_2$ concentrations are generally lower than in the overlying soils, ranging from 500 to 10000 ppm, most of the cave $CO_2$ concentrations showed low ranges of values no more than 6500 ppm (Ek and Gewelt, 1985; Spötl *et al*., 2005; Faimon and Ličbinská, 2010). Although a few studies revealing very high $CO_2$ concentrations exist in deep and confined karst caves, e.g. the identification of average vadose $CO_2$ in the range of 10,000–40,000 ppm, with a maximum of nearly 60,000 ppm in boreholes near Nerja Cave, Spain (Benavente *et al*. 2010, 2015).

The $CO_2$ concentration in the individual karst reservoir (e.g., cave) is the result of balancing all the input and output $CO_2$ fluxes (Lang *et al*., 2017). The principal cave inputs include: (1) natural $CO_2$ fluxes associated with direct diffusive flux from soils/epikarst (e.g. Ek and Gewelt, 1985; Cuezva *et al*., 2011; Krajnc *et al*., 2017; Pla *et al*., 2017), (2) indirect $CO_2$ fluxes derived from dripwater/stream degassing (Baldini *et al*., 2006; Breitenbach *et al*., 2015), and (3) anthropogenic exhalation from visitors in some show caves (Faimon *et al*., 2006; Milanolo and Gabrovšek, 2009; Šebela and Turk, 2014; Lang *et al*.,
2015), (4) Other possible $CO_2$ fluxes derived from microbial decay of organic matter in cave sediments, animal respiration, or endogenous processes (Atkinson, 1977; Batiot-Guilhe *et al*., 2007; Mattey *et al*., 2016), (5) deep magmatic or metamorphic sources (Bergel *et al*., 2017) and (6) atmospheric air. The cave outputs are controlled by cave ventilation that is mainly driven by (1) cave geometry and (2) the temperature difference between the exterior and interior cave (Lang *et al*., 2017). Of these potential sources, soil respiration and atmospheric air are traditionally considered to be the most significant in most caves
(Ridley *et al*., 2015; Lang *et al*., 2017). The close relationships existing between the outdoor atmosphere, the soil/rock membrane and the underground atmosphere constitute a multicomponent system that works in concert (Pla *et al*., 2017). The production and transport of subterranean $CO_2$ within surface soils or on bedrock cavities have been widely studied, especially the surface-atmosphere $CO_2$ exchanges, with either manual (e.g. Davidson *et al*., 1998) or automated soil chamber systems (Lund *et al*., 2010), where soil $CO_2$ is generally of biological origin, though some studies have observed a geological source
(e.g. Rey *et al*., 2012; Mattey *et al*., 2016).

Variations in pressure and wind (Takle *et al*., 2004; Sanchez-Cañete *et al*., 2016) are factors that influence soil $CO_2$ transport. In the events, the magnitude and variability of soil $CO_2$ production is mainly driven by soil temperature (Pumpanen *et al*., 2003), soil water content (Vargas *et al*., 2012), or soil geochemistry (Roland *et al*., 2013). The importance of understanding specific cave ventilation mechanisms has been well highlighted in recent studies (Kowalczk and Froelich, 2010; Benavente *et*
*al*., 2015; Breitenbach *et al*., 2015), i.e. Mattey *et al*. (2010) revealed that variations in cave-air $pCO_2$ are in relation to unusual seasonal ventilation regimes and suggested that it should be very careful to link paired speleothem fabrics to specific seasons without knowledge of local processes operating in the cave. A growing number of studies focus on mechanisms of variations in soil $CO_2$ fluxes and sources of $CO_2$ in subterranean caves (Kuzyakov and Gavrichkova, 2010; Krajnc *et al*., 2017), where subterranean caves are considered as a temporary means of $CO_2$ storage—accumulating $CO_2$ from different sources (Pu *et al*.,
2014; Krajnc *et al*., 2017). The ventilation of subterranean $CO_2$ from the pores, fissures and cavities present in the karst environment accounts for variations of the total $CO_2$ concentration (Serrano-Ortiz *et al*., 2010; Breitenbach *et al*., 2015).



High-resolution datasets of $p\mathrm{CO_2}$ have been established, e.g. a linear relationship existed between the distance from the cave entrances and cave $p\mathrm{CO_2}$ (Baldini *et al.*, 2006) or whether it is correlated with outer temperature (Ek and Gewelt, 1985). According to their study, Carbon dioxide produced by the soil biomass is accumulated into underground voids due to
gravitational drainage from cracks and fissures. Especially, in descending caves where carbon dioxide is heavier than the other main atmospheric components, can accumulate during the hot season due to the "cold trap effect".

The stable carbon isotope is a useful tool to understand the mixing process existing inside a cave. An increasing number of authors use the Keeling plot (Keeling, 1958) to express cave air as a mix between two end-members (Spötl *et al.*, 2005; Kowalczk and Froelich, 2010; Frisia *et al.*, 2011; Tremaine *et al.*, 2011). The light end-member source should be located close
to the roots of C3 type vegetation, with carbon isotope ranging from −30‰ to −24‰ (Vogel, 1993). Both bulk and root-free soil respired $\delta^{13}\mathrm{CO_2}$ exhibited depleted values ranging between −29‰ and −26‰ (Unger *et al.*, 2010). Soil air $\mathrm{CO_2}$ is enriched by about 4.4‰ (Cerling *et al.*, 1991) due to a different diffusion coefficient for $^{12}\mathrm{C}$ and $^{13}\mathrm{C}$ compared to root respired and decomposed $\mathrm{CO_2}$. Thus, soil is commonly considered as the light end-member, and this assumption is correct as long as the $\mathrm{CO_2}$ concentration in the cave is lower than the soil (Peyraube *et al.*, 2013). The $\delta^{13}\mathrm{C_{CO2}}$ derived from geothermal sources (e.g.,
magmatic or metamorphic sources) typically ranges from 2‰ to 6‰ (Faure, 1986). Moreover, the R/Ra versus $\delta^{13}\mathrm{C_{CO2}}$ plot, traditionally used to estimate crustal versus mantle components of $\mathrm{CO_2}$ (Sano and Marty, 1995). $^{222}\mathrm{Rn}$ is a radioactive gas that is naturally produced from the decay of uranium and other radioactive atoms in the carbonate host-rock in caves, which is accumulated in the subterranean atmosphere and usually covaries with $\mathrm{CO_2}$ concentration (e.g. Gregorič *et al.*, 2013). Kowalczk and Froelich (2010) evaluated cave air ventilation and $\mathrm{CO_2}$ outgassing by $^{222}\mathrm{Rn}$ modelling in the Hollow Ridge
Cave (Florida, USA), finding the highest $\mathrm{CO_2}$ outgassing in late summer and early autumn (about 4 mol h$^{-1}$) and the lowest in winter (about 0.5 mol h$^{-1}$). The process that was suffering a sharp decrease as consequence of the ventilation has also been confirmed by Valladares *et al.* (2014) who used $^{222}\mathrm{Rn}$.

Though high-frequency data logging has produced reliable annual carbon balances around the world, few studies have been designed to monitor continuously the $\mathrm{CO_2}$ exchanges from the soil and cave. In this study, the authors have presented the soil
$\mathrm{CO_2}$, stream $p\mathrm{CO_2}$, and cave air $p\mathrm{CO_2}$ monitoring data in high frequency from Xueyu Cave, SW China during the period of 2015-2016. The aim of this paper is to 1) understand the quantitative relationship between cave air $\mathrm{CO_2}$, soil $\mathrm{CO_2}$ and the stream $p\mathrm{CO_2}$; 2) To reveal the sources and factors that control the variations in cave air $\mathrm{CO_2}$.

## 2 The study area

Xueyu Cave (29°47′00″ N, 107°47′13″ E) is located in Fengdu County, with the elevation of 233 m at the entrance, Chongqing,
China (Fig. 1). This region has a typical subtropical monsoon climate with a multiyear average precipitation of approximately 1072 mm. The geological formation and secondary speleothems, including soda straw, stalactites, stalagmites, cave flags, cave shields in the cave were explored (Zhu *et al.*, 2004). As described, the thickness of the roof rocks of Xueyu Cave is over 150





m, and the mean internal temperature is 17.2 °C. The cave develops in the northwestern wing of the anticline that is consisted of Lower Triassic Feixianguan formation ($T_1f$) limestone with argillaceous rock at the base and silt rock at the top, Lower

Triassic Jialingjiang Formation ($T_1j$) dolomitic limestone with salt dissolution breccias at the top, and Middle Triassic Leikoupo Formation ($T_2l$) argillaceous limestone embedded with silty shale. The systemic study of the links between the host rock, water and speleothems has been performed to explain the universal cementation of sparry low-magnesium carbonate (Wu *et al*., 2015).

**Figure 1** (a) Geographical location of the study area, (b) Monthly air and precipitation in Xueyu Cave, (c) The location of the Xueyu Cave, its surrounding strata and the soil sampling site, (d) Sketch map of the Xueyu Cave and locations of the monitoring sites: X1 and X5 for cave air and stream $p$CO$_2$ monitoring.

The relationships between specific conductance (Spc), $Ca^{2+}$ and $HCO_3^-$ have been established and variations of cave $CO_2$ concentrations in the cave atmosphere and cave stream showed different changes in wet and dry season due to the ventilation

(Pu *et al*., 2015, 2018), Seasonal variations of calcite growth rate are primarily controlled by variations of cave air $pCO_2$ and drip water rate, which are highly related to seasonal changes of overlying soil $CO_2$ content outside the Xueyu Cave (Wang *et al*., 2013), leading to seasonal variations of $\delta^{18}O_{V-PDB}$ and $\delta^{13}C_{V-PDB}$ in modern calcite precipitates (Pu *et al*., 2015, 2016). High $^{222}$Rn and $CO_2$ concentrations typically occur during the warm summer periods, and low concentrations are typical in cold winter (Yang *et al*., 2013; Pu *et al*., 2018).

## 3 Methods and materials

A set of system for continuous and automatic $CO_2$ measurement with a $CO_2$ sensor (0-20000 ppm) was fixed on the ceiling of the Xueyu Cave. Soil $CO_2$ was collected to analyze $\delta^{13}C_{CO2}$ in summer and winter, respectively. Meteorological data including precipitation (precision 0.01 mm) and temperature (precision 0.1 °C) were recorded every 15 min using a HOBO weather station.

Two sites inside the Xueyu Cave for $pCO_2$ monitoring of cave air and the subterranean stream have been selected at Longfeng (X1) and Manzi (X5), respectively (Fig. 1). The data were recorded each quarter based on a GMM221 sensor (within the range 0~20000ppm, precision ± 1%) connected with RR-1008 data receiving terminal.

The $\delta^{13}C$ of soil $CO_2$, cave air $CO_2$ and $\delta^{13}C_{DIC}$ of stream water were completed at the Geochemistry and Isotope Laboratory of the Southwest University. Analyses were performed using a Delta-V-Plus Mass Spectrometer connected to a Gas Bench

pretreatment apparatus. The results were reported using V-PDB as the reference and the analysis precision was better than 0.2% (1σ).

## 4 Results

### 4.1 Climatic records

High rainfall amounts corresponded with the high temperature except in July or August when the study area was always

influenced by summer drought that was controlled by Western Pacific Subtropical High. During 2015-2016, the mean annual rainfall amounts was 1149 mm, and the air temperature ranged from 3.3 °C to 39.5 °C with an average of 19.5 °C.

### 4.2 Soil CO2

Fig. 2 describes soil $CO_2$ changes that drive the cave $CO_2$ and $pCO_2$ in the stream. In the soil at 40 cm, the soil concentrations ranged from 6500 ppm in December to 17000 ppm in June. It is in accord with other studies which show the range of soil $CO_2$

concentrations in karst regions between 1000 ppm and 30000 ppm (Spötl *et al*., 2005; Wang *et al.*, 2013; Krajnc *et al*., 2017).



Other variables were seasonally dependent, the evolution of soil $CO_2$ were controlled by soil temperature and humidity. The soil temperature ranged from 8.0 °C in December to 24.0 °C in August and the soil humidity varied between 0.5% and 24.0%, the minima occurred in spring months (March 2015) and dry summer (July-August, 2015-2016). When the temperature is suitable in summer, soil moisture works as the main constraining factor for variations in soil $CO_2$ (Fig. 2).


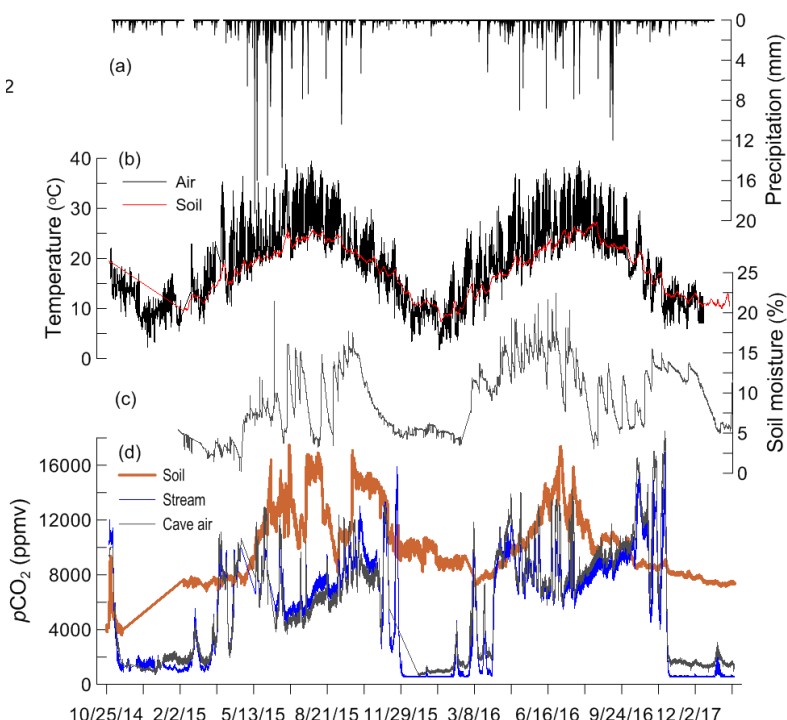

**Figure 2:** (a) Precipitation, (b) Air temperature and soil temperature, (c) Soil moisture, (d) $pCO_2$ values in the soil air, cave air and stream water of Xueyu system in the years 2015-2016.

### 4.3 Cave parameters

The cave air temperatures at X1 (the upper layer) ranges from 16.0 °C to 18.7 °C with a mean value of 16.2 °C, while at X5 (the lower layer) from 16.3 °C to 16.8 °C with a mean value of 16.6 °C. The complex cave geometry with three layers strongly influences airflow direction and velocity in different parts of the cave, resulting in changeable cave air temperature, but still less variable compared with the external air temperatures (3.3-39.5 °C). About 150 days (the 1st May to the 1st October) present continuous external temperatures above the cave temperatures and 90 days (the 1st December to the 1st March) present 145 continuous external temperatures below the cave temperatures (Fig. 2b).

The seasonality of cave $CO_2$ variations based on monthly monitoring has been reported by Pu *et al*. (2018) that seasonal $pCO_2$ in cave stream000000 and drip waters were higher in wet season from April to October than in dry season every year. Our data




further provide high-frequency records of temperatures and $CO_2$ variations that allow us to see the details of variations and the processes.

Monitoring data of cave $CO_2$ from 2015 to 2016 showed that cave $CO_2$ varied in a consistent way on a sub-annual scale, resulting in steady or fluctuated $CO_2$ periods (Fig. 2). In winter, cave $CO_2$ was relatively steady at its minimums (<1000 ppm); whereas, it fluctuated largely and increased to be relatively abundant in summer (>6000 ppm). However, the peaks of the cave $CO_2$ took place at the beginning of November 2015 and 2016. Specifically, abrupt changes of cave $CO_2$ occurred at the moments of alternate seasons (autumn to winter or spring to summer) with large variational magnitudes, e.g. cave $CO_2$

concentrations increased to 16000 ppm and decreased to 1000 ppm within several days in November 2015 & 2016. The sharpness of the transitions during the seasons demonstrates that it responded immediately to the changes of external environments. In winter, low cave $CO_2$ concentrations indicated limited contributions from sources. During rainfall events, cave $CO_2$ concentrations changed due to increased high-$CO_2$ flow that rainwater dissolved soil $CO_2$, which largely disturbed the periodical variations in cave $CO_2$ on annual/seasonal scale. The cave $CO_2$ values are comparable with the values in some

other karst caves reported by Sánchez-Moral *et al.* (1999) for the Altamira Cave (6000 ppm), lower than extreme 60000 ppm (Benavente *et al.,* 2015). A high-frequency monitoring in October 2014 and June 2015 showed the detailed changes of $pCO_2$ and carbon isotopes during rainfall events. Soil $CO_2$ concentrations have less variability than $pCO_2$ in the cave or stream (Fig. 3, Fig. 4).

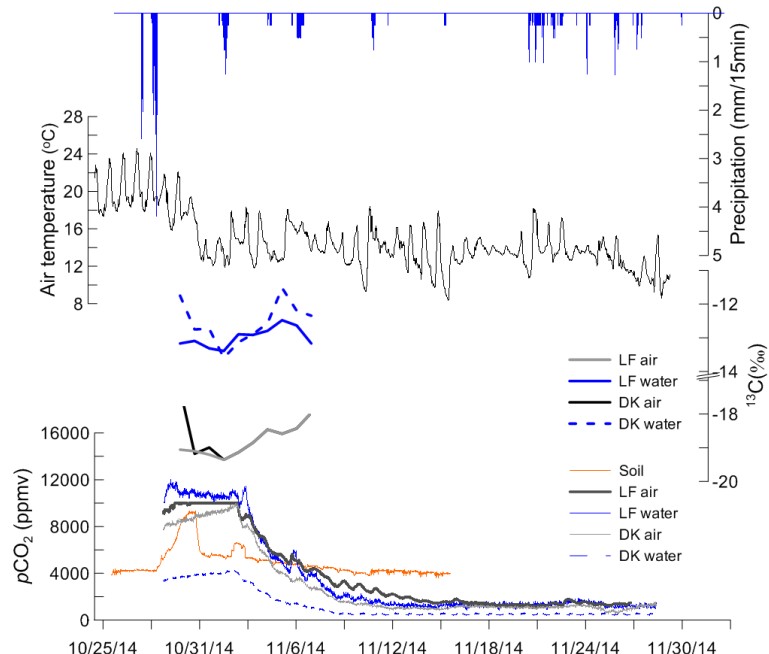

**Figure 3** Variations of monitoring items (precipitation, temperature, $\delta^{13}C$ and $pCO_2$) during rainfall events in October, 2014.



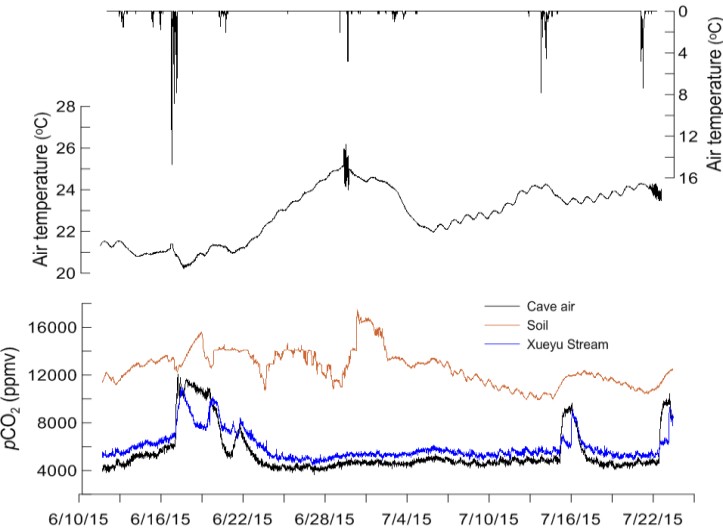

**Figure 4** Variations of monitoring items (precipitation, temperature, $\delta^{13}C$ and $pCO_2$) during rainfall events in June, 2015.

## 4.4 The carbon isotope $\delta^{13}C$ in cave air and stream water

The $\delta^{13}C$ value of background atmospheric $CO_2$ was -9.6 to 10.0‰, which is similar to the observation by Mattey et al. (2010)

that atmospheric $CO_2$ was −9.6‰. The $\delta^{13}C$ values ranged from -31.7‰ to -29.9‰ in plants and -18.0‰ in the overlying soil in winter but -23.9‰ in summer on average. In the subterranean stream, monthly-sampled $\delta^{13}C_{DIC}$ values ranged from -14.0‰ to -10.0‰ with a mean value of -12.2‰ (Fig. 5). $\delta^{13}C$ values of cave air ranged from -23.5‰ in summer to -18.5‰ in winter. High-frequency monitoring in November 2014 and June 2015 showed a decreasing trend and then an increasing trend of $\delta^{13}C$ values during rainfall events. Lower values corresponded to high flow periods, from April to September. Moreover, water

coming from the upper stream has lower values of $\delta^{13}C_{DIC}$ than that from the downstream.

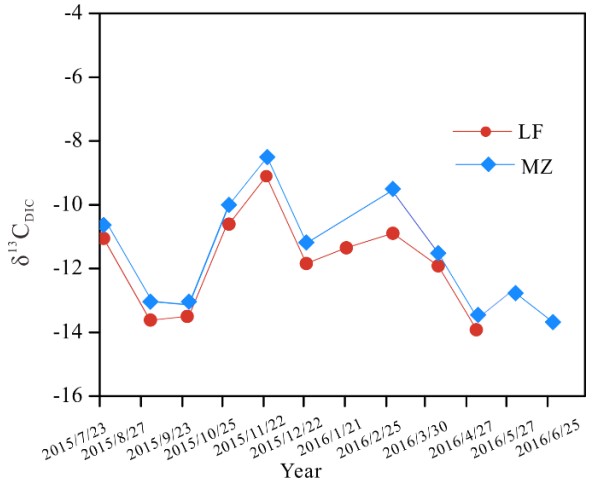

**Figure 5** The monthly variation in carbon isotopes of stream water at upstream and downstream in the Xueyu Cave



## 5 Discussion

### 5.1 $\delta^{13}C$ isotope tracing the sources of cave air $CO_2$

The $\delta^{13}C$ values of the cave stream generally showed seasonal fluctuations with the lowest values occurring in summer months and the highest values in winter months (Fig. 5). It was consistent with previous observation that winter samples with relatively low $pCO_2$ were isotopically heavy (Spötl *et al*., 2005). The interannual variability of carbon isotopes seems to be related to precipitation, resulting in lighter $\delta^{13}C$ values with more precipitation. In October, the monitoring results of rainfall events showed that $\delta^{13}C$ values of the stream DIC and cave air $CO_2$ decreased at the beginning of the rain and then increased during

the process at DK site (near the entrance of the cave). However, at LF site (upstream) those $\delta^{13}C$ values were in an increasing trend (Fig. 3). Moreover, the variability of $\delta^{13}C$ values was higher at DK than LF. In June, the $\delta^{13}C_{DIC}$ values of stream water at two sites decreased and then increased during the rainfall events. However, the $\delta^{13}C_{CO2}$ values of cave air were increasing at both site (Table 1).

The distribution of $\delta^{13}C_{CO2}$ of cave air is clearly similar to isotopic composition of the soil-respired $CO_2$ during rainfall events

(Fig. 6), which was more scattered in winter than in summer, indicating that there was no simple genetic link between soil gas $CO_2$ and the other $CO_2$ endmembers in the cave air.

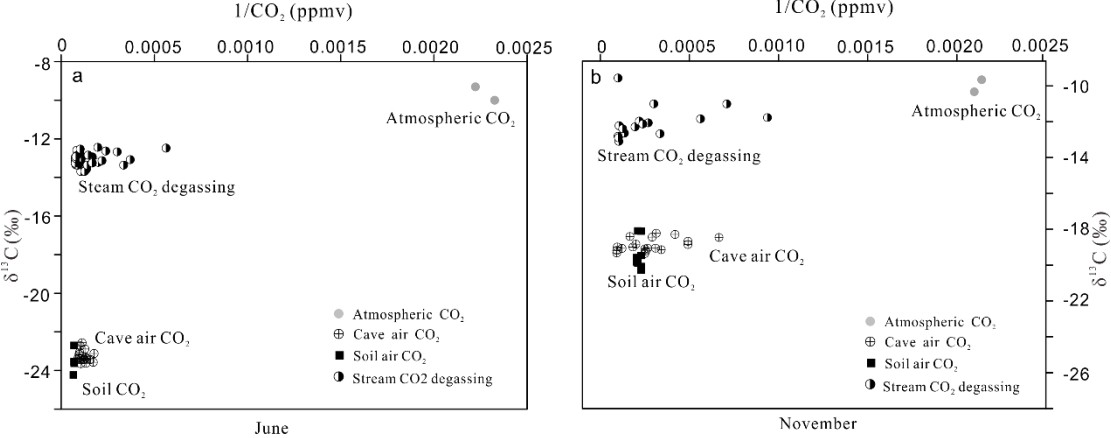

**Figure 6** The relationships between $\delta^{13}C_{V\text{-}PDB}$ and $1/CO_2$ during the occurring of rainfall events in June (a) and November (b).

There is a linear relationship between $\delta^{13}C_{CO2}$ versus inverse $[CO_2]$ concentration (Mattey *et al*. 2016). In order to understand

completely the underlying processes controlling the generation and dispersal of $CO_2$ in karst systems, the identification of the carbon isotopic compositions of the endmember components in each reservoir have been carried out. The source of the isotopically light endmember $CO_2$ in the standard model is soil respiration that is dominant in cave air (Baldini *et al*., 2006; Frisia *et al*., 2011; Breecker *et al*., 2012). Other potential sources include degassing from $CO_2$-riched groundwater, deep-sourced $CO_2$ and decomposition of organic matter within the cave itself (Breecker *et al*., 2012). Fractionation by diffusion in

the pore space results in heavier $\delta^{13}C$ (Cerling, 1984). The composition of root-respired $CO_2$ is likely to be in the same range





as $CO_2$ decomposed from organic matter, a fraction of 4.4 (‰) induced by diffusion suggest that pore-space $CO_2$ in Gibraltar soil might have a limiting composition of -23.6‰ to -21.6‰ (Mattey *et al.*, 2016).

In Xueyu Cave, the $\delta^{13}C_{CO2}$ of cave air showed seasonal variations and similar values to that in soils and regarding to the degassing phenomenon in the cave, we think an equation can be established:

$$\delta^{13}C_{CO2} = [\textstyle\sum_0^i(mCi)\,(\delta^{13}Ci)/\sum_0^i(mCi)]$$

where $m$ is the percentage of the $CO_2$ deriving from different sources, $i$ refers to different sources. The mixing model has two end members, one of which from the degassing of stream water is isotopically light and the other that represents $CO_2$ generated by oxidative decay of soil organic matter is relatively heavy in winter. However, the $CO_2$ from degassing was heavier and that
from soils was lighter in summer, indicating we can use a mixing model to distinguish the contributions from different sources.

The average values of $\delta^{13}C_{soil}$, $\delta^{13}C_{DIC}$ in June were -23.9‰ and -13.4‰, respectively (the $\delta^{13}C_{CO2}$ from degassing -21.4‰ due to isotopic fraction of 8‰). $\delta^{13}C_{CO2}$ of atmospheric air was -10.0‰ and $\delta^{13}C_{CO2}$ of cave air was -23.3‰. The contributions from soil $CO_2$ and stream degassing to cave air in June were 78.8% and 21.2% on average, respectively (Table 1). The average values of $\delta^{13}C_{soil}$, $\delta^{13}C_{DIC}$ in November were -18.0‰ and -12.2‰, respectively (the $\delta^{13}C_{CO2}$ from degassing -20.9‰ due to
isotopic fraction of 8‰). $\delta^{13}C_{CO2}$ of atmospheric air was -9.6‰ and $\delta^{13}C_{CO2}$ of cave air was -18.8‰. The contributions from soil $CO_2$ and stream degassing to cave air in November were 67.1% and 32.9% on average, respectively (Table 1). Moreover, it seemed that the contribution from soil $CO_2$ was increasing with the decreased $pCO_2$. It confirms cave air $CO_2$ was mainly from soil $CO_2$ no matter in June or November, but the contribution from soils was higher in June than in November. In other words, the contribution of C from the soil was larger in summer than in winter. The light $\delta^{13}C_{co2}$ in the Xueyu cave air are
close to −23.3 ‰ in summer, coherent with a biogenic origin of the $CO_2$ produced by tree root respiration and/or organic matter degradation by bacteria (Clark and Fritz, 1997), discarding the deep $CO_2$ and the human respired $CO_2$ as sources.

The monitored two rainfall events showed that the water filled up the rock porosity and fractures after a heavy rain event. Then the air contained in the surface layers of rock environment may have been pushed downward. The low value of light member $\delta^{13}C_{co2}$ during rainfall events could have been caused by this mass air movement, suggesting the presence of a super-light
member all along the year in some parts of the rock environment, with which depleted $CO_2$ can be pushed toward the cave after a heavy rain event (Peyraube *et al.*, 2017).

### 5.2 Controlling factors for variations in cave air $CO_2$

The stream coming through the cave undergoes degassing without significant calcite precipitation. The effect of human respiration in the cave is considered to be small based on the slight changes in $CO_2$ during tourism activity. Though some
studies observed that tree roots are penetrating through cave ceilings (Frisia *et al.*, 2011; Bergel *et al.*, 2017). It is possible that





the fractures are full of gas or water to prevent $CO_2$ diffusion from the cave in summer, resulting in slow exchange between the interior cave and atmosphere (Fig. 7).

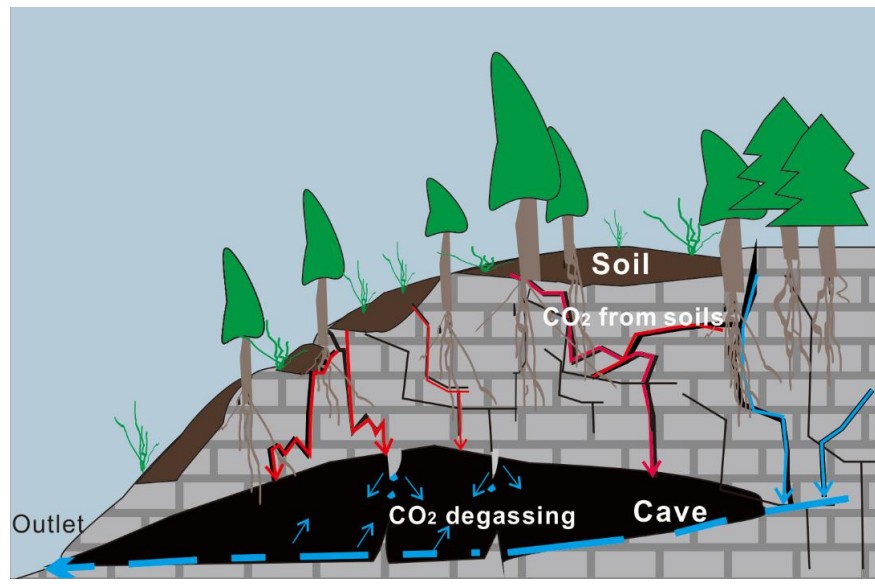

**Figure 7** Conceptual model for subsurface carbon cycling in Xueyu karst cave. $CO_2$ respired in soils is transported into caves by gaseous form or infiltrated in rainwater. Changes of ventilation patterns which might be correlated to soil moisture overlying can help to accumulate cave air $CO_2$ or make it dispersed in summer and winter. Sketch of the seasonally controlled airflow of the Xueyu Cave system and resulting in $pCO_2$ changes.

### 5.2.1 Soil $CO_2$

The soil temperature tracked air temperature very closely throughout the year 2015-2016 (Fig.2). From June to August, the study area always suffered summer drought due to the control of West Pacific Subtropical High (WPSH), which was coherent with high air temperature and low precipitation (Zhou *et al.*, 2009). Li and Li (2018) showed that both local temperature and precipitation significantly affected the monthly soil $pCO_2$ of the Furong Cave (a cave nearby, 100 km from Xueyu Cave in Chongqing).

The time series (Fig. 2) suggest that the seasonal pattern of soil $CO_2$ concentrations above Xueyu Cave were generally controlled by soil temperature and soil humidity. When soil moisture is not constrained, soil $CO_2$ production rates should rise with temperature, following an exponential or Arrhenius relationship (Fang and Moncrieff, 2001). Whereas soil humidity plays a key role in constraining soil $CO_2$ concentration, i.e. soil water content affects not only plant growth but also gas permeability and both diffusive and advective $CO_2$ transport (Mattey *et al.*, 2016), resulting in the maxima and minima of soil $CO_2$ concentrations in wet spring and dry summer months in 2016, respectively (Fig. 2). Lower soil gas $CO_2$ concentrations could be due to reduced $CO_2$ production rates in dry conditions, or to greater gas permeability in dry soil, which facilitates the escape



of $CO_2$ to the surface by advection induced by synoptic pressure changes, wind and diurnal heating and cooling (Mattey *et al.*, 2016). Soil $CO_2$ concentration decreased sharply with the onset of summer drought and rose back if soil water content recovered before October (i.e. August-September 2015). However, the increased soil humidity in later month could not promote the soil $CO_2$ concentration as soil temperature was too low (Fig. 2, October 2016). Thus, the seasonal variation in soil

$CO_2$ concentration controlled by soil temperature can be disturbed by variations in soil moisture.

The seasonal enrichment in soil $CO_2$, stimulated by higher summer temperatures, is transferred to the cave environment, resulting in seasonal changes of cave air composition and growth rate of speleothems (Spötl *et al.*, 2005; Baldini *et al.*, 2008). Carbon might be transmitted through different reservoirs that are physically or chemically linked by water/air transport or exchange processes. On one side, water moves downwards via fissures and fractures, some of it entering caves as drips and

seepage; some of it as shaft flow and stream water (Ford and Williams, 2007). On the other side, air has a more complex pattern of movement, which may circulate rapidly through high-permeability conduits provided by caves, natural fissures, artificial tunnels and boreholes or circulate more slowly as 'ground air' through the continuum of voids made up by natural fractures and pores in the bedrock (Weisbrod *et al.*, 2009; Mattey *et al.*, 2016). Normally, $CO_2$ concentrations in cave air are lower than overlying soils. Higher cave air $CO_2$ than in contemporaneous soil air might indicate an additional process that

generates such high levels (Benavente *et al.*, 2010, 2015). $pCO_2$ in the cave air changed more abruptly than in the soil air, which might be related to multiple $CO_2$ origins in Xueyu Cave and also the changes of different $CO_2$ transport pathways as this cave is connected to a permanent stream to allow carbon transmission by water and gas media. The soil $CO_2$ reaches the cavity via diffusion and causes gas recharge. Soil-produced $CO_2$ drifts slowly downwards into the cave, filling the pore space in the rock to hold the high $CO_2$ concentrations (Pla *et al.*, 2017). During rainfall events, the pore space acts as the source, to

push the $CO_2$ concentrations towards the cave. In October, when the cave air $pCO_2$ exceeds the soil $CO_2$ concentrations, the cave air $pCO_2$ climbs to a maximum.

### 5.2.2 The degassing from the subterranean stream

Cave air $pCO_2$ in Xueyu Cave showed a seasonal cycle with increased values from spring to autumn and fluctuated periodically at the transitions from winter to spring or from autumn to winter. Troester and White (1984) showed in a study of a cave

passage floored by a stream that cave-air $pCO_2$ was controlled by the degassing of the seasonally varied $pCO_2$ in the stream. In Xueyu Cave, stream flow may also play an important role for $pCO_2$ dynamics as $CO_2$ degassing and absorption by stream water, like in Ballynamintra Cave (e.g. Baldini *et al.*, 2006). The stream running through Xueyu Cave increased its discharge dramatically in the wet season, resulting in warm surface air into the cave companying with rainfall events. Increased $CO_2$ absorption from the slow-moving or stagnant cave air by carbonate weathering and potentially the stream might explain the

low cave-air $pCO_2$, which is similar to the observation in Mawmluh Cave of India (Riechelmann *et al.*, 2017). Strong stream flow can induce air velocities proportional to that of the stream via friction between water and air where the stream can adjust the system (Cigna, 1968; Fairchild and Baker, 2012; Breitenbach *et al.*, 2015).



Monitoring data from the stream at site X1 showed that variations of the stream $pCO_2$ (S-$pCO_2$) were consistent with the cave air $pCO_2$ (C-$pCO_2$) on the seasonal scale (Fig. 2). Specifically, S-$pCO_2$ variations at X1 were in accordance with C-$pCO_2$ in

dry season, whereas fluctuated largely in rainy season, up to 5000 ppm. The difference between S-$pCO_2$ and C-$pCO_2$ was relatively low, suggesting a good equilibrium between the stream and cave air unless during the periods of rainfall events and transitions. The $pCO_2$ variability $\theta(pCO_2)$ between S-$pCO_2$ and C-$pCO_2$ has been shown in Fig. 8.

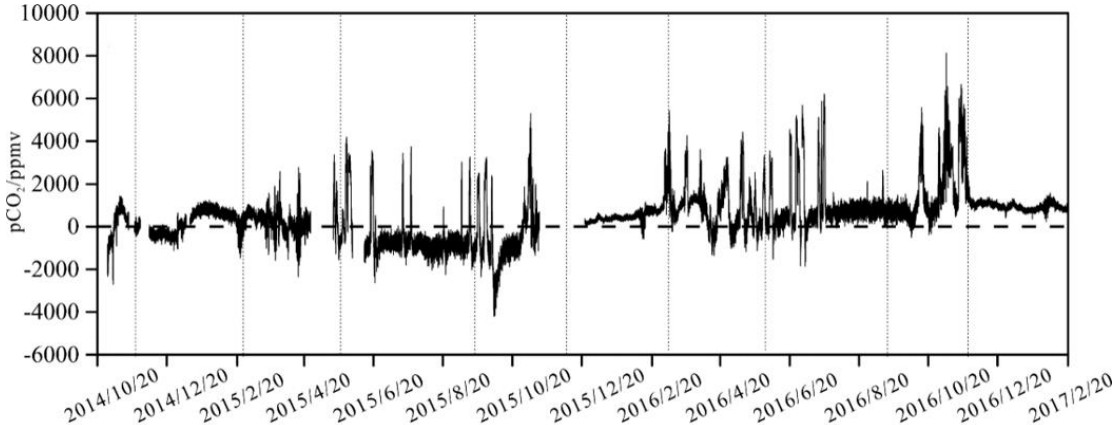

**Figure 8** The $pCO_2$ variability ($\theta(pCO_2)$=S-$pCO_2$ − C-$pCO_2$) in the Xueyu stream and cave air system

There is a close relationship between cave-air $pCO_2$ and stream $pCO_2$ (Fig. 2). The $pCO_2$ in the cave air and stream of Xueyu Cave nearly changed in the same pattern, showing higher and fluctuated values in summer but lower and steady values in winter. In general, it observed that S-$pCO_2$ and C-$pCO_2$ were in equilibrium throughout a year; while rainfall events and alternate moments of seasons could disturb abruptly the equilibrated state from the upstream, resulting in variations of C-$pCO_2$ and S-$pCO_2$ in the downstream. In dry season, the $pCO_2$ difference between LF and DK was nearly 0; whereas in rainy season,

the fluctuated range could reach up to 5000 ppm (S$_{LF}$-$pCO_2$>S$_{DK}$-$pCO_2$), indicating the degassing from the upstream to the downstream. The $\theta(pCO_2)$ was higher in 2016 than in 2015 due to fluctuated changes of rainfall amounts. The fluctuated values of cave air $pCO_2$ in Xueyu Cave were always corresponding to large variations in $\theta(pCO_2)$ with a time lag, indicating that $\theta(pCO_2)$ might drive cave air $pCO_2$ variations. In October, 2014, soil $CO_2$ concentrations increased gradually from 4000 ppm to 9000 ppm at the beginning of the rainfall event and then decreased sharply to 5000 ppm after the rainfall. On the contrary,

cave air $pCO_2$ at LF and DK were higher than soil $CO_2$, ranging from 8000 ppm to 12000 ppm, which were higher than that in soils (Fig. 3). The higher values of $pCO_2$ in the cave air might be originated to $CO_2$ in the vadose zone (Baldini *et al.*, 2006), which was brought by rainwater in the carbonate fissures or fractures. The stream water $pCO_2$ ranged from 8000 ppm to 9000 ppm at LF, but it was below than 4000 ppm at DK. It is obviously that the stream water was degassing from LF to DK.

### 5.2.3 Precipitation

Low $CO_2$ production in the overlying soils during the dry season aggravated the scarcity of $pCO_2$ in the cave. During rainfall processes, cave air $pCO_2$ responded quickly to heavy rains and strong storms, the variational magnitudes of which were highly





related to rainfall intensities (Table 2). There are negative correlations between cave air $p$CO$_2$ variability ($\triangle p$CO$_2$=$p$CO$_2$ after rain−$p$CO$_2$ before rain) and response time, i.e. R²=0.57 ($\triangle p$CO$_2$<4000 ppm) and R²=0.74 ($\triangle p$CO$_2$>4000 ppm). The accumulated rainfall amount in 2016 was higher than that in 2015, which was in consistent with the general increasing trend of cave air $p$CO$_2$ variations during the two years. More rainfall amounts resulted in high levels of cave air and stream $p$CO$_2$.

Two monitoring sites, X1 and X5 (LF and DK) for S-$p$CO$_2$ and C-$p$CO$_2$ variations during a rainfall event at the end of October 2014 were shown in Fig. 8. This period also belonged to the seasonal transition when the air temperature decreased from 20.0℃ to <9.0℃, S-$p$CO$_2$ and C-$p$CO$_2$ were decreasing with the variational magnitude of 10000 ppm. It is a pity that we did not fix the equipment to record the complete changes of cave $p$CO$_2$ before 28$^{th}$ October. However, it assumed that the infiltrated rainwater helped to push $p$CO$_2$ in the soil or fissures into the cave. Another observational period during a rainfall event was in June 2015, when soil CO$_2$ concentrations were higher than S-$p$CO$_2$ and C-$p$CO$_2$.

Furthermore, regarding to the geometry and structure of the Xueyu Cave, during the warm and rainy season, the system of epi-karstic fissures is almost permanently saturated with water, making the host rock membrane impermeable to prevent the CO$_2$ diffusion from cave. In contrast, during the cold and dry season the epi-karstic porous system is not water saturated, opening paths for CO$_2$ movement. Similar pattern but in reverse seasons has been observed in other soil-cave system (Mattey *et al*., 2016). The mechanism is depending on water that seals the pores or condenses the porous system, where gas transport through the overlying soil is determined by the pore size distribution, inter-particle porosity and water content (Cuezva *et al*., 2011). In this case the soil acts as an impermeable/permeable membrane due to different conditions, the space of porosity gradually changes and results in changed cavity ventilation, preventing/promoting communication between the cave and exterior. This mechanism is able to explain the abrupt changes of cave air $p$CO$_2$ during season transitions.

## 6 Conclusions

1) Two-year monitoring study of soil CO$_2$, subterranean stream and cave air $p$CO$_2$ indicates that cave air $p$CO$_2$ in Xueyu Cave was mainly controlled by soil CO$_2$ via gaseous diffusion and the degassing of the subterranean stream, whose variability is influenced by source endmembers and transport way.

2) High-resolution monitoring of $p$CO$_2$ in the soil and cave system may allow us to estimate the potential processes that drives variations of $p$CO$_2$ in cave air. Throughout the year, $^{13}$C$_{DIC}$ showed higher values in winter but lower values in summer. $^{13}$C of different endmembers showed that soil CO$_2$ made more contribution of C to the cave air CO$_2$ in June (75.6％) than in November (65.9％), and the second source was the degassing of stream.

3) The seasonal variation of cave air $p$CO$_2$ was very similar to that in stream $p$CO$_2$, showing high but fluctuated values in summer and steady but low values in winter. Moreover, there were abrupted change between by the end of October and March due to changes of recharged CO$_2$ inputs and variation in climatic parameters (temperature and precipitation). The



period of abruption occurs in November because of the delayed transport of soil $CO_2$.

## 7 Author contribution

340 MC wrote the manuscript and prepared the figures with contributions of all authors. JJ designed the whole research for monitoring, sampling and data analysis. JL, ZZ and JF did the sampling and laboratory work. All authors contributed to the data interpretation and discussion of the manuscript.

## 8 Competing interests

The authors declare that the research was conducted in the absence of any commercial or financial relationships that could be construed as a potential conflict of interest.

## 345 9 Acknowledgements

This research was supported by the National key research and developmental program of China (2016YFC0502306), the National Natural Science Foundation of China (NSF Grant no. 41472321) and the open project from Chongqing Key Laboratory of Karst Environment (Cqk201701). Thanks to Ze Zeng, XianFu Lv, Ge Hu and Sibo Zeng who helped with the field work.

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





## Figures

**Figure 1: (a) Geographical location of the study area, (b) Monthly air and precipitation in Xueyu Cave, (c) The location of the Xueyu Cave, its surrounding strata and the soil sampling site, (d) Sketch map of the Xueyu Cave and locations of the monitoring sites: X1**
**and X5 for cave air and stream $p$CO$_2$ monitoring.**

**Figure 2: (a) Precipitation, (b) Air temperature and soil temperature, (c) Soil moisture, (d) $p$CO$_2$ values in the soil air, cave air and stream water of Xueyu system in the years 2015-2016.**

**Figure 3: Variations of monitoring items (precipitation, temperature, $\delta^{13}$C and $p$CO$_2$) during rainfall events in October, 2014.**

**Figure 4: Variations of monitoring items (precipitation, temperature, $\delta^{13}$C and $p$CO$_2$) during rainfall events in June, 2015.**

**Figure 5: The monthly variation in carbon isotopes of stream water at upstream and downstream in the Xueyu Cave.**

**Figure 6: The relationships between $\delta^{13}$C$_{V\text{-PDB}}$ and 1/CO$_2$ during the occurring of rainfall events in June (a) and November (b).**

**Figure 7: Conceptual model for subsurface carbon cycling in Xueyu karst cave. CO$_2$ respired in soils is transported into caves by gaseous form or infiltrated in rainwater. Changes of ventilation patterns which might be correlated to soil moisture overlying can help to accumulate cave air CO$_2$ or make it dispersed in summer and winter. Sketch of the seasonally controlled airflow of the Xueyu**
**Cave system and resulting in $p$CO$_2$ changes.**

**Figure 8: The $p$CO$_2$ variability ($\theta$($p$CO$_2$)=S-$p$CO$_2$ − C-$p$CO$_2$) in the Xueyu stream and cave air system.**

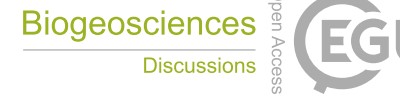

## Tables

**Table 1 The δ¹³C values from cave air and stream and the contribution of cave CO₂ from soils**

| Time | Cave air (‰) | | Stream (‰) | | The proportion from soils (%) | |
|------|------|------|------|------|------|------|
| | MZ | LF | MZ | LF | MZ | LF |
| 2014/10/30-09:00 | -18.2 | -19.1 | -10.6 | -12.9 | 59.2 | 63.9 |
| 2014/10/31-09:00 | -19.2 | -19.1 | -12.2 | -12.8 | 48.2 | 61.3 |
| 2014/11/1-09:00 | -19.0 | -19.2 | -12.2 | -13.0 | 56.2 | 60.7 |
| 2014/11/2-09:00 | -19.3 | -19.4 | -12.1 | -13.1 | 57.9 | 56.7 |
| 2014/11/3-09:00 | -19.1 | -19.1 | -12.6 | -12.6 | 60.1 | 56.9 |
| 2014/11/4-09:00 | -19.0 | -18.9 | -12.3 | -12.6 | 58.1 | 68.3 |
| 2014/11/5-09:00 | -18.3 | -18.5 | -12.1 | -12.5 | 84.4 | 82.6 |
| 2014/11/6-09:00 | -18.4 | -18.6 | -11.0 | -12.2 | 61.8 | 74.2 |
| 2014/11/7-09:00 | -18.4 | -18.4 | -11.7 | -12.3 | 75.5 | 82.8 |
| 2014/11/8-09:00 | -18.3 | -18.4 | -11.8 | -12.9 | 85.8 | 88.2 |
| Mean values | -18.7 | -18.9 | -11.9 | -12.7 | 64.7 | 69.5 |
| 2015/6/15-09:00 | -23.4 | -23.6 | -13.2 | -13.3 | 82.4 | 89.7 |
| 2015/6/16-09:00 | -23.3 | -23.2 | -13.4 | -13.9 | 77.9 | 68.6 |
| 2015/6/17-09:00 | -23.4 | -23.6 | -13.5 | -13.6 | 81.6 | 90.6 |
| 2015/6/18-09:00 | -23.4 | -23.4 | -13.9 | -13.8 | 79.3 | 79.4 |
| 2015/6/19-09:00 | -23.4 | -23.6 | -13.5 | -13.6 | 81.2 | 88.1 |
| 2015/6/20-09:00 | -23.4 | -23.3 | -13.0 | -13.2 | 85.2 | 81.7 |
| 2015/6/21-09:00 | -23.3 | -23.1 | -13.4 | -13.7 | 80.3 | 64.5 |
| 2015/6/22-09:00 | -22.7 | -23.2 | -12.8 | -13.5 | 63.7 | 71.9 |
| 2015/6/23-09:00 | -22.9 | -23.3 | -13.1 | -13.3 | 65.1 | 81.1 |
| 2015/6/24-09:00 | -23.4 | -23.3 | -12.9 | -13.3 | 86.0 | 77.0 |
| Mean values | -23.3 | -23.4 | -13.3 | -13.5 | 78.3 | 79.3 |







**Table 2 The $p\mathrm{CO_2}$ variability and response/equilibrium time responding to different rainfall intensity during rainy season in 2015**

| No. | Time | Intensity | $p\mathrm{CO_2}$ range /ppm | Response time /h | Equilibrium time /h |
|-----|------|-----------|------------------|---------------|----------------|
| 1 | 5/17 | Heavy rain | 1800 (8200-10000) | 9 | 44 |
| 2 | 5/23 | Moderate rain | 2500 (6500-9000) | 8.4 | not |
| 3 | 5/25 | Moderate rain | 5700 (7300-13000) | 17.5 | not |
| 4 | 5/28 | Moderate rain | 3400 (9400-12500) | 2.5 | 32.4 |
| 5 | 6/17 | Strong storm | 5840 (6160-12000) | 2 | 69.4 |
| 6 | 7/14 | Heavy rain | 4300 (4900-9200) | 25 | 134 |
| 7 | 7/22 | Heavy rain | 5500 (4500-10000) | 10.5 | 27.5 |
| 8 | 8/19 | Moderate rain | 2600 (6400-9000) | 9 | 53.5 |
| 9 | 9/11 | Storm | 4800 (7200-12000) | 11.4 | 83.5 |
| 10 | 9/17 | Heavy rain | 4600 (7000-11600) | 27.5 | 70 |