# Peer review of "Constraining the soil carbon source to cave-air CO2: evidence from the high-time resolution monitoring soil CO2, cave-air CO2 and its $\delta^{13}$ C in Xueyudong, Southwest China"

_Biogeosciences, 2019_

## Referee Comment (RC1) · Anonymous Referee #1 · 8 Apr 2019

The review of the article "Constraining the soil carbon source to cave-air CO2: evidence from the high-time resolution monitoring soil CO2, cave-air CO2 and its $\delta$13C in Xueyudong, Southwest China" by Min Cao, Yongjun Jiang, Jiaqi Lei, Qiufang He, Jiaxin Fan, Ze Zeng. The authors present the data on CO2 in the soil, cave stream, and cave atmosphere (Xueyu Cave, China) and its surrounding. The data were gathered during the period of 2015-2016. The aim of the article is (1) to understand the quantitative relationship between all the forms of CO2, (2) to reveal their sources, and (3) to understand the factors that control the cave air CO2 variations. The topic

of the article is important and is worthy of publication. In the article, however, there are some aspects that require revision and other ones that could be substantially improved before publishing. My main reservation is that the conclusions should be better proved by a data analysis (e.g., Cross-correlation Analysis). The results of the data analysis should be presented and discussed in detail. The data sets are nice, but they could be much better presented. The x-axis should be more extended in order to be better distinguishable individual fluctuations in the variables. Other comments: • Throughout the text, it is important to distinguish $CO_2$ itself from $CO_2$ concentration and $PCO_2$ (e.g., the lines/paragraph 85). • The expression "$PCO_2$ in the water" (stream $PCO_2$) is acceptable only as an abbreviation in the text. Furthermore, it is important to explain that it means $PCO_2$ of gaseous $CO_2$ that would be in equilibrium with aqueous carbonates. • In principle, $PCO_2$ is dimensionless variables (or it has units of pressure). If the $CO_2$ quantity is given in ppmv units, it means "$CO_2$ concentration". • Some soil characteristics should be given in the paragraph Study Area. • More detail information should be given in monitoring/calculating of the stream $PCO_2$ in the paragraph Methods and Materials. • The x-axes in the plots (Fig. 2, 3, 4, 5) should be better divided (e.g., by one month, three months, etc.). • The secondary y-axis in Fig. 4 should represent "Precipitation". • I do not understand what the conceptual model in Fig. 7 brings new/beneficial. • In the text, there are missing the citation: Liu and Zhao 2000, and Baker et al., 1998 and 2014, referenced in the Reference list.

Please also note the supplement to this comment:
https://www.biogeosciences-discuss.net/bg-2019-66/bg-2019-66-RC1-supplement.pdf

---

## Referee Comment (RC2) · Anonymous Referee #2 · 9 Apr 2019

Comments for Cao et al. "Constraining the soil carbon source to cave-air $CO_2$: evidence from the high-time resolution monitoring soil $CO_2$, cave-air $CO_2$ and its $\delta^{13}C$ in Xueyudong, Southwest China"

**Major Comments**
This article uses environmental and isotope evidence from soil, stream water, and cave air to characterize the dynamics of carbon distribution in the Xueyu Cave system (China) and identify the contributions of potential reservoirs to overall cave air $CO_2$. The work is important because it builds on a growing set of literature describing how and why cave air $CO_2$ changes and has implications for interpretations of speleothem records used to reconstruct past climate. However, the paper is also missing key sections of the methodology, not all data is reported, and the discussion and data analysis are incomplete. The following areas require the authors' attention before publication:

1. **Manuscript grammar:** I appreciate that the authors may not be native English speakers, but sections of the manuscript are difficult to read. In particular, this hampered my understanding of the arguments the authors made in the discussion. I noted several sections that were unclear and need revision.
2. **Methodology:** Sections 2 and 3 are missing important information about sampling locations, sample collection (methodology, frequency, storage), and analysis methodology (instrumentation, standards). Measurements of d13C-atmosphere and d13C-plant material are reported but no methodology is provided.
3. **Data tables:** Not all data is reported in the tables, making it difficult to reproduce the authors' graphs and calculations. If there is not space in the main paper, data should be placed in a supplemental section or data repository.
4. **Discussion section:**
   a. The mixing model employed for interpretation is not appropriate. Based on the authors' data, a model identifying CO2 sources must, at minimum, (1) include atmospheric $CO_2$ and (2) consider the close relationship between stream- and cave air-$CO_2$ concentration. The authors must also explain why they do not consider other potential sources listed in the introduction.
   b. It is not clear to me that the November data really describe 'winter' conditions as cave air pCO2 does not drop to its 'winter baseline' until a week or two after the collection date. Do your isotope data represent baseline summer/winter cave conditions or only those during rain events?
   c. The Discussion repeats information from the Results. I suggest a restructuring of the Discussion. In addition to your interpretation at this cave site the Discussion should focus on (1) comparison to previous studies of this nature and (2) the broader implications of the research for the cave community (e.g., studies of modern dripwater-calcite formation relationships and speleothem-based climate reconstructions).

**Specific Comments**
**Title**
A more informative title is "Constraining the source and dynamics of cave air $CO_2$ in a cave system in Xueyudong, Southwest China through $CO_2$ and $\delta^{13}C$ measurements"

**Abstract**
Line 20
Your abstract suggests that we need studies like this one to interpret stalagmite records, but does not tell the reader how this study contributes to our understanding of how to interpret speleothem records.

**1 Introduction**

The introduction could focus more attention on why we care about CO2 concentrations. I gather that you are interested in caves as a source of proxy records – spend more time explaining the connection between cave CO2 and speleothem records (as well as the current gaps in knowledge). The introduction should lead the author logically to the final sentence of the section (line 85) where you state the aims of the paper.

Line 69
Is this region dominated by C3 plants? Cite a reference for this if so.

Line 85
This section needs to be clearer. I suggest:
"The aim of this paper is to (1) identify the dynamics of carbon distribution and transfer between cave air $CO_2$, soil air $CO_2$, and stream $CO_2$, and to (2) identify the contributions of major reservoirs to overall cave air $CO_2$."

Line 88
Rephrase "The study area" to "Study area"

Line 89
More information is needed on the stream. Does it flow in/out of the cave? Or is it entirely underground? Pieces of information are available in the manuscript, but it should all be collected and put up front in this section.

**Figure 1**
- Make all figure subsection labels (a, b, c, d) more obvious
- Legends on subsections b and c are too small
- 1C
    - Is this figure after another paper? Needs to be cited if so
    - Why is 'location of measured geological section' in here? You did not measure any sections
    - Rephrase 'River/stream and its name' as 'River'
    - Rephrase 'The curves that frame the Xueyu Cave' as 'Xueyu Cave outline'
- 1D
    - The pictures of equipment are too small. Include them as separate sections of the figure or put them in the supplemental material
    - The map needs a north arrow and scale bar
    - The location of the stream needs to be better defined. Where does it enter/exit the cave?
- Caption
    - Describe the inset in part a (the small map of China)
    - Where are monitoring sites DK, LF, and MZ? They must be labeled

**3 Methods and Materials**
All measurement types require more information so readers can assess the methodology.
For CO2 concentration measurements:
- Automated measurements (CO2-cave air, CO2-soil, and CO2-stream)

- o Were all measurements made with the GMM221 sensor?
  - o How was the sensor modified for measurement of CO2-stream? List part numbers if direct from manufacturer.
  - o Who is the sensor made by? Vaisala?
  - o How frequently were measurements made? What time periods were measured?
  - o How were the sensors calibrated? How often were they calibrated?
  - o What was the depth of measurement for soil CO2?
- Precipitation and temperature
  - o List the part number(s) for the HOBO weather station
- Discrete samples
  - o All discrete samples
    - When were measurements made (list months, not summer/winter)?
    - What was the time period of sampling (two 10-day periods)
    - What were the frequency of measurements (1/day)?
    - How were samples stored and transported? How much time elapsed between collection and measurement?
    - How was CO2 concentration determined for the discrete samples of cave air, soil CO2, and DIC (i.e., data in Figure 6)
  - o D13C-cave air CO2
    - What precautions were taken to avoid sampling your own breath?
  - o D13C-soil CO2
    - What were the depths of collection?
    - How much soil air was collected? Sampling soil air has the potential to introduce gas advection and destroy the d13C signal that you are attempting to measure
  - o D13C-DIC stream water
    - How were samples collected and what preservation techniques were used? Were samples filtered, was air headspace eliminated, were samples refrigerated/stored in the dark, were microbes poisoned with a substance like MgCl2? This is critical for the d13C-DIC measurements as improper storage can substantially alter the d13C-DIC and render interpretation unviable.
  - o D13C-atmosphere
    - This needs to be included in the methodology
    - What precautions were taken to avoid sampling your own breath?
  - o D13C-plants
    - This needs to be included in the methodology
    - What plants and parts of plants were sampled? Why were these particular samples chosen?
- Analysis
  - o What is the methodology for d13C analyses?
  - o What standards were used for d13C measurement?

Line 112
Be clear that samples collected for d13C-CO2 analyses are not the same samples as those from the continuous collection regime.

Be more precise than "in summer and winter, respectively." The samples were collected once a day during two 10 day periods in November 2014 and June 2015. Also:
- Note that these are the same collection periods for d13C-cave air and –stream DIC

- Why were these time periods chosen?
- Why are there data gaps in the d13C data (e.g., DK air of Figure 3)?

**4 Results**
Line 127
"Soil CO2" needs to be "Soil $CO_2$"

Line 129
Soil CO2 concentrations bottom out around 4000 ppm in November 2014

Line 130
Why do you compare soil CO2 concentration at your site to these other studies? Do they have similar climate and vegetation regimes?

Line 131
Be consistent in using "soil moisture" instead of "humidity."

Line 134
If soil moisture controls respiration when temperature is suitable, what is occurring in summer 2015? It looks like there are time periods when pCO2 is high but soil moisture is low (July-Aigust).

**Figure 2**
- Make the data gaps more obvious. Note in the text where these are and why they occurred
- The x-axis is difficult to read. Label it by month instead?
- Mark the d13C sampling intervals on here so it is obvious where to look for the 'zoomed-in' sections presented in Figures 3 and 4
- 2A
    o I find the inverted y-axis confusing - precipitation should logically increase upwards
- 2B
    o Include cave temperature on here as well (or at least the average)

Line 139
Rephrase "Cave parameters" to "Environmental measurements"
Line 140-141
- What are the "upper layer" and "lower layer?"
- The average cave temperatures are different from the average presented on line 93.
- Include cave temperature in Figure 2B
- What are the "three layers" – this is the first time this is mentioned in the text.

Line 147
Typo "stream000000"

Line 151
Does cave CO2 decrease to atmospheric levels? It looks like it does from Figure 2

Line 157
- Could low cave CO2 concentrations be related to effective transport of cave air to the outside?
- In any event, this kind of interpretation should be left to the Discussion section

Line 157-159
I'm not clear on the meaning of this sentence.

Line 162
What is "less variability?" Define this.

**Figure 3**
Figures 3 and 4 should be combined for ease of reference
- Precipitation should increase upwards
- Precipitation should be black, as in the other diagrams
- The same materials (e.g., CO2-cave air and d13C-cave air) should be the same line color and type
- Include error bars on d13C measurements
- "LF" and "DK" are not defined before Figure 3. Where are these sites?
- Caption
  o Rephrase to "during rainfall events in *October-November* 2014"

**Figure 4**
- The precipitation plot is labeled as air temperature
- Precipitation should increase upwards
- What are the high-frequency oscillations (6/29 and 7/22) in the cave temperature record? Were sensors repaced at this time?
- Where are the d13C measurements?
- The same materials (e.g., CO2-cave air and d13C-cave air) should be the same line color and type
- Include error bars on d13C measurements
- Caption
  o Rephrase to "during rainfall events in *June-July* 2014"

Line 168
Rephrase "4.4 The carbon isotope d13C in cave air and stream water" to "4.4 Carbon isotopes in cave air, stream water, and soil"

Line 169
Why cite Mattey et al. (2010) for atmospheric d13C measurements at the Rock of Gibraltar? There are long-term records of atmospheric CO2 that would be more directly relevant to your site

Line 170
- This is the first time that measurements of plant d13C are mentioned. Information about plant collection and measurement should go in the methodology
- What is the range of d13C-soil CO2?
- Remind readers of the depth of soil CO2 collection as this is a critical parameter for interpretation

Line 173
- Change "High-frequency" to "Daily"

- A decreasing then increasing trend is potentially seen in the 'DK water' data, but I do not see this trend in any of the other samples

Line 174
- Where is data for low/high streamflow? It is not mentioned before this point

**Figure 5**
- Plot needs error bars
- Why are the high resolution measurement periods not shown?
- Where are sites LF and MZ? Specify the 'upstream' and 'downstream' locations
- This data needs to be reported in a table (or in the supplemental information)

**5 Discussion**
Line 183
Rephrase 'lighter $\delta^{13}$C to 'more negative $\delta^{13}$C.' Values cannot be lighter or heaver. See, for example, table 2.1 in Sharp's Stable Isotope Geochemistry (https://digitalrepository.unm.edu/unm_oer/1/). Fix throughout the manuscript.

Line 184
- The values for d13C-cave air need to be reported in a table and the collection+analysis method need to be described in the Methodology
- 'cave air CO2 decreased at the beginning of the rain and then increased during the process at DK site.' There does not appear to be a strong initial decrease in the 'LF air' data and the 'DK air' data do not cover the entire time period. I suggest incorporating these observations into your interpretation
- When does the rain event start? This could be stated clearly here and be shown more clearly (vertical dotted lines?) in the graphs

Line 185
As noted above, it is not clear where the DK and LF sites are. I will not note further instances, but this needs to be addressed for the whole paper.

Line 186
Define 'the variability of d13C values'

Line 187
'the d13C-DIC values of stream water at two sites decreased and then increased during the rainfall events.' Depending on exactly when the rainfall event occurred, this may be true for site MZ. However, I see no overall change in the values for site LF.

Line 189-191
This sentence is unclear and appears to contradict itself. Please clarify how you are interpreting the relationship between soil gas and cave air.

**Figure 6**
- The y-axes on both plots should be the same to allow easy comparison
- The left plot has 'Steam $CO_2$ degassing,' which should be 'St**r**eam'

- The 'Stream CO2 degassing' data reported in this figure appear to be d13C-DIC values. Reporting these data as the d13C of CO2 in equilibrium with stream DIC requires calculation of the fractionation factor between DIC and CO2
- Keep the order the same for all graphs. Show November and then June (June is shown first in Figure 6)

Line 200
'heavier d13C' should be 'higher d13C'

Line 200-202
- This sentence is unclear – is your intent to relate the d13C of respired organic matter to d13C of soil air CO2?
- Why are soil air measurements in Gibraltar relevant to your field site in SE China? Why not use your own measurements to make an estimate?

Line 205
I have the following issues with the discussion section:
- Why is a 2-endmember mixing model appropriate for your conceptual model? Several of your citations suggest a simple 2-endmember mixing model is inappropriate for understanding changes in cave air.
  - o You consider CO2 contributions from soil, stream, and human breath
  - o However, your introduction considers these additional sources important: atmospheric CO2, organic matter decay in the cave, magmatic/metamorphic sources
- Atmospheric air appears to be a particularly important endmember that this model does not address. The authors need to revise their data analysis to incorporate all of the information available from the dataset conceptual model of how/why cave air CO2 changes
- If >75 % of cave air CO2 is from the soil, why is there much better seasonal correlation between CO2-cave air and CO2-stream? Do your results apply only to rain events or year round?
- What causes the overall U-shape in the cave air and stream CO2 data every summer? Again, if soil CO2 is controlling cave air CO2, why is this signal not visible in the soil CO2 data?
- It is unclear to me from the discussion whether you think the soil, stream, or both are controls on cave air CO2. However, in the conclusions you definitively identify soil contributions as most important. Your position should be made clearer and should be supported by the isotope and CO2 concentration data.
- You briefly describe that d13C-DIC of the stream is controlled by flow rate (Line 174). Is there a relationship between stream flow rate and cave air CO2 or d13C?
- The discussion repeats results and repeats itself in sections. It should be edited for clarity and structure. I suggest the following general structure:
  - o Interpretation of what is occurring at Xueyu Cave
  - o Comparison to other studies of this nature
  - o Implications for developing paleoclimate records from speleothems (here and elsewhere)

Line 211
Is d13C-soil referring to soil organic matter or soil air CO2? If it refers to soil air CO2, keep in mind that d13C-soil air CO2 changes with depth. Justify using a single value.

Line 211-212

- A citation and explanation are needed for the 'd13C-CO2 from degassing -21.4 per mil due to isotopic fraction of 8 per mil.' Converting from DIC to the CO2 in equilibrium with it is not a straightforward connection for unfamiliar readers
- 'fraction' should be 'fractionation'

Line 213
I do not get the same output from your model using the values in Table 1

Line 214
Same as line 211-212 – a citation and explanation are needed for the fractionation between DIC-CO2

Line 219
'light d13Cco2' should be 'more negative d13C'

Line 228
How does water degassing CO2 not precipitate calcite?

**Figure 7**
- How is this model different from those proposed/used by other you cite? Might be better just to cite/describe the model.
- I did not understand that the river flowed from inside the cave to outside the cave until this figure – this information should be up front in the study area description

Line 277-278
This sentence is unclear: what does 'resulting in warm surface air into the cave companying with rainfall events' refer to?

Line 283
The terms 'S-pCO2' and 'C-pCO2' are confusing. I recommend not using them

Line 287
Delete the final sentence of this paragraph

Figure 8
- Mark months of the year on the x-axis, not the 20th of each month
- Mark when the cave switches between summer and winter modes

Line 290
This section largely repeats what has been already said

Line 303
Where is the CO2 data for stream water at the two LF and DK sites? We are only presented with one dataset

Line 305
- This section is difficult to understand. I'm not sure what I am supposed to get out of it.
- Define the metrics 'before rain' and 'after rain,' response time, intensity, and equilibrium time

- Lines 311-316 do not seem to add to the section. If you are reporting results, they should be in the Results section

Line 309
'in consistent' should be 'inconsistent'

**6 Conclusions**
Line 331
Measurements were made two times in the year, not 'throughout the year'

Line 322
'$^{13}C$' should be '$\delta^{13}C$'

Line 333
The stated percentage contributions do not match Table 1

**Table 1**
- The time transgressive values do not give the reader the idea that a single value of d13C-soil CO2 is assumed for the whole time period
- The table should include the d13C values for cave air, stream DIC, and calculated CO2 in equilibrium with stream DIC

**Table 2**
- This table does not mean much to the reader as the parameters are not defined (intensity, response time, equilibrium time)
- This table can be moved to the supplemental information

---

## Referee Comment (RC3) · Anonymous Referee #3 · 12 Apr 2019

about a cave research study entitled 'Constraining the soil carbon source to cave-air CO2: evidence from the high-time resolution monitoring soil CO2, cave-air CO2 and its $\delta$13C in Xueyudong, Southwest China'

In this contribution, the authors measured partial pressure of cave and soil gas CO2 and DIC concentrations in a cave stream. Those measurements were complemented by temperature (cave, soil and atmosphere) and precipitation observations. The log-

ging was performed for more than two years. Furthermore, they provide the stable C isotopic composition of various components (stream water, soil and cave air) for two shorter periods of time in lower resolution (10 days, one sample per day). The aim of this study is to investigate the major sources of $CO_2$ in the cave.

Unfortunately, the manuscript is relatively difficult to read and I am not always sure, I understood what the authors wanted to say. In a possible revision, one focus should be on improvements with respect to the readability. Apart from this, I think, the study is not in a shape to be published. To my opinion, the text should be considerably improved before it could be considered for publication. There are many details, which should be improved and some sections should be reworked. However, I want to emphasize, that I think that a lot of the findings appear to not to be valid but that there are some potential within the data, to make it a quite nice contribution to the scientific community, which is worth to publish once the problems are solved, the text reworked and new ideas implemented.

Major points

Throughout the text (e.g., line 9) when the authors write of 'pCO2 of cave water'. To me, this sounds very sloppy and should be prevented. I mean, gaseous $CO_2$ (which can be well expressed as pCO2) is dissolved in water and then forms the different carbon species often referred to as dissolved inorganic carbon (DIC). The DIC cannot be expressed as pCO2. It is all dissolved and should be expressed in terms of concentrations. To my opinion the authors most likely want to express with their wording something similar to: 'the pCO2 the cave water is in equilibrium with'. This wording is much more precise and should be applied throughout the whole text.

The abstract should be reworked. In line 11 you are arguing that pCO2 of the cave air depend on wet and dry periods. But as I understood your data, the main influence is on the temperature. Only if temperature is warm enough, the wet/dry relationship is important. The last sentence of the abstract, appears weird for me as well after having

read the whole paper. In this sentence, you are referring to stalagmite records and what your findings imply for those. However, in the manuscript you did not discuss the influence of the important cave processes on the stalagmite stable carbon proxy. So either you should delete this sentence or add some discussion about this in the text. I guess the latter would be the more valid approach, as this way your story will have more impact.

As the stream seems to play an important role in the cave, I think it is important to let the reader know some more details. E.g., for which distance the stream is in contact with the cave atmosphere, how much water is transported, how fast is its velocity and something like that. This could be well included in Section 2. This is important, as this way other researchers investigating other caves can set their observations in better context and comparison with your results is easier.

I recognized that the cave is ascending into the host rock. However, with respect to cave $CO_2$ and the derived ventilation regime, it behaves, according to your data, like those caves which are going downwards into the rock. This is quite interesting and could be much more emphasized. Maybe the stream is the key for this behavior? You should put some efforts here and elaborate more on this.

Section three should be completely reworked, as some information are given twice (e.g., about the $CO_2$-sensor; Line 111 and 116) and others incomplete (How was soil $CO_2$ measured? How was the sampling performed for soil gas d13C analysis? How was the $pCO_2$, the stream water is in equilibrium with, determined?) or no information at all (e.g., about measurements of cave and soil temperature, soil moisture, d13C values of plants [according to line 170 they have been measured]).

Within the manuscript you are claiming: 'When the temperature is suitable in summer, soil moisture works as the main constraining factor for variations in soil $CO_2$.' (e.g., Line 134). First, please don't discuss this in the results, but put it to the discussion and please be more detailed here. By eye, it is easily visible, that soil gas $CO_2$ might

be changed by soil moisture during the warm season. However, there seem to be a clear time delay between soil moisture and pCO2. I feel, you have to discuss this. Where is this delay coming from? Please, do some statistics, e.g., calculate correlation coefficient, determine time lag, .... You might be even able to make some statements about the sensitivity of soil CO2 variations with respect to soil humidity under various temperature ranges, as soil pCO2 seems also to react on soil humidity under lower temperatures. However, the sensitivity appears to be different when the variations in the cold and warm seasons are compared. This all, will have some influence on section 5.2.

Fig. 6: What exactly is plotted for the stream water? The measured d13C, ok. But how have you determined the x-axis value? Is this the pCO2 value, the stream water is in equilibrium with? But not all of this C can degas from the stream (due to the chemical limits of degassing) and thus cannot contribute to the cave air CO2. So the pCO2 what comes from the stream is for sure lower than the pCO2 value the water is in equilibrium with. Thus, those values should be put more to the right (the question is how much?). Due to all this, I think, providing this stream C source in the Keeling plot this way is quite bold without arguing why this can be done. This is then changing the available CO2 which can degas and is thus also changing the according mCi in your equation (line 205). As those plots contain the basis of your argumentation according to your present discussion, a change here, might have major influences on your final findings.

Technical comments:

Line 26: 'always with higher values in summer and lower values in winter' I want to point out that this is not true. This is only true for caves, which lead downwards with increasing distance from the entrance. For those caves, which leads upwards, the opposite will be observed. But you have indicated this correctly, e.g., in line 43. So please, be consistent and more precise here.

Line 27-28: Unfortunately, I do not understand this sentence. Can you reword this?

[Figure]

Line 32: It seems to me, that 'although' is not fitting here.

Line 64: 'carbon' with a small first letter.

Line 65: 'Especially, in descending caves where carbon dioxide is heavier than the other main atmospheric components . . .' I am pretty sure you are meaning this differently than stated here. $CO_2$ is always heavier than the main atmospheric components ($O_2$ and $N_2$), not only in descending caves. Maybe reword this to something like that: 'As carbon dioxide is heavier than the other main atmospheric components, $CO_2$ accumulates in descending caves during the hot season due to the "cold trap effect".

Line 73: The 'Thus' does not seem to fit here, as the previous sentence does not provide a reason for what you are stating in the following sentence, which you begun with 'Thus'.

Line75: Please define R/Ra. What is this?

Line 76-77: Please be more precise and reword this sentence as $^{222}Rn$ is not produced from the decay of U and other radioactive atoms as you have stated here. It would be more precise to write something similar to: '$^{222}Rn$ is a radioactive isotope that is naturally produced within the $^{238}U$ decay chain. As it is heavier than air it is accumulating in the . . .'.

Line 90: 'multiyear average precipitation' sounds a bit strange. Do you mean 'average annual precipitation'?

Line 91: Is the term 'secondary speleothems' correct? It seems to me, you might mean 'secondary carbonates' instead?

Line 93: Please provide information, of how constant cave air T is. Give its variation throughout the year and plot it maybe even in Fig. 2

Fig. 1: It is not clear, why the monitored drip sites are shown in d), as these are not discussed in the text. In addition, I am very confused about the naming of your river

drip-sites. Here in Fig. 1 they are labelled as X1 and X5. In Tab. 1 they are labeled MZ and LF while throughout the text and in the figures they are labeled DK and LF. Please stay consistent.

Line 103: 'cave' appears to be unnecessary here.

Line 105: Please, replace ',' by '.' after ')'.

Line 106: 'drip rate' instead of 'drip water rate'.

Line 126: 'amount' instead of 'amounts'

Line 128 and 131: Here you quite often make some statements, which should be carefully discussed before. To my opinion those statements do not belong in the results section, but should be shifted towards the discussion section.

Line 133-134: This belongs also to the discussion and it is not quite clear, what 'suitable' is meaning here. Please, be more precise. I also wonder if this statement is justified (see above). Fig. 2: Please indicate the cave air temperature.

Line 137: Capital letters of 'Air' and 'Soil' should be small ones.

Line 146-147: Sorry, but I do not understand this sentence. Could you rewrite this under the correction of the 'stream00000' typo.

Line 156-163: This should be shifted to the discussion part. And I do not understand the sentence in Line 157. Also the following sentence (line 157-159) is not clear to me.

Line 169: Typo. Should be '-10 permil', shouldn't it?

Line 170: Please include here, that you are talking about soil gas, which has values of -18permill to -23.9.

Line 183-185: I am sorry. From my side, this does not look as contemporaneously as you described it. Please provide some shaded rectangles in the appropriate figures to allow to better follow your argumentation. For me it even seems, that there is no rain

event in October, which was also covered with DIC and pCO2 measurements at DK. In November are some rain events, but there, I could not see the described relationship. DK-d13C is already decreasing before the rain event. Thus, it appears the decrease in d13C has nothing to do with the single rain event. And to be honest, the observation of such behavior (if there would be indeed one) during only one single event is not very convincing.

Line 186-188: Please also show this in a figure. With only seeing the numbers in the table, I am not able to follow.

Line 205: Please subscript the 'i' in the equation.

Line 207-209: Why has the mixing model only two endmembers? What is with the atmospheric input? Why can this be neglected. Please explain. But then, an end-member modelling with three sources is much more difficult. (even with the assumed two end-members, the equation in line 205 needs an additional equation to order to be solved. [Sum of all mCi = 1])

Line 211-212: Please, cite the work you are taking the fractionation factor from.

Line 212-216: Even with your two end-member modelling, I am somewhat confused, about your numbers for the contribution of the C sources to the cave air. There is quite some scatter in the d13C of soil gas, cave air and DIC (or degassed CO2) of the stream. The problem then is that the difference between stream degassed CO2 and soil air is quite small compared to the scatter in your values. This makes the calculation quite difficult. At least, I would require to give some error estimates in your values here.

Line 230: Why are you mentioning roots here. Are there roots in your cave? I assume not, otherwise I would have expected some description of them earlier in Section 2. So it is not clear to me, why you are mentioning them here with respect to other caves. This sentence is also completely out of context with the sentence before and after.

Line 280: Mawmluh cave was not investigated by Riechelmann et al., 2017. They

investigate Bunker Cave. Please change the citation accordingly.

Line 292: Why do you say that cave CO2 and stream DIC are in equilibrium with each other? For me this looks quite different in Fig. 8. There is barely a time, when the difference between both is 0. Even if this would be the case, all your argumentation from above that the stream is a significant CO2 source is destroyed by this sentence. If both would be in equilibrium, no CO2 can be degassed from the stream and could not provide CO2 to the cave atmosphere. And if all the CO2 of the stream would have been degassed earlier, you could also not do any end-member modelling as you do not have the initial conditions of the water.

Line 308: Please define and explain, what the response time is. How have you calculated this? Can you show a plot of the observed relationship?

---

## Author Comment (AC1) · 24 May 2019

**Response to Referee #1**

The review of the article "Constraining the soil carbon source to cave-air $CO_2$: evidence from the high-time resolution monitoring soil $CO_2$, cave-air $CO_2$ and its $\delta^{13}C$ in Xueyudong, Southwest China"by Min Cao, Yongjun Jiang, Jiaqi Lei, Qiufang He, JiaxinFan, Ze Zeng. The authors present the data on $CO_2$ in the soil, cave stream, and cave atmosphere (Xueyu Cave,China) and its surrounding. The data weregathered during the period of 2015-2016. The aim of the article is (1) to understand the quantitative relationship between all the forms of $CO_2$, (2) to reveal their sources, and (3) to understand the factors that control the cave air $CO_2$ variations. The topic of the article is important and is worthy of publication. In the article, however, there are some aspects that require revision and other ones that could be substantially improved before publishing. My main reservation is that the conclusions should be better proved by a data analysis (e.g., Cross-correlation Analysis). The results of the data analysis should be presented and discussed in detail. The data sets are nice, but they could be much better presented. The x-axis should be more extended in order to be better distinguishable individual fluctuations in the variables.

*Answer to general comments:*
*We would like to thank the referee for his generally positive comments. We will pay more attention in presenting and explaining the our data in the final version.*

Other comments: Throughout the text, it is important to distinguish $CO_2$ itself from $CO_2$ concentration and $pCO_2$ (e.g., the lines/paragraph 85). The expression "$PCO_2$ in the water" (stream $pCO_2$ is acceptable only as an abbreviation in the text. Furthermore, it is important to explain that it means $pCO_2$ of gaseous $CO_2$ that would be in equilibrium with aqueous carbonates. In principle, $pCO_2$ is dimensionless variables (or it has units of pressure). If the $CO_2$ quantity is given in ppmv units, it means "$CO_2$ concentration". Some soil characteristics should be given in the paragraph Study Area. More detail information should be given in monitoring/calculating of the stream $CO_2$ in the paragraph Methods and Materials. The x-axes in the plots (Fig. 2, 3, 4, 5) should be better divided (e.g., by one month, three months, etc.). The secondary y-axis in Fig. 4 should represent "Precipitation". I do not understand what the conceptual model in Fig. 7 brings new/beneficial. In the text, there are missing the citation: Liu and Zhao 2000, and Baker et al., 1998 and 2014, referenced in the Reference list.

*Answer to other comments:*
1. *We put more details in the study area part, the method part and discussion part, and refined the conclusions too.*
2. *$pCO_2$ only refers $CO_2$ in aqueous form. We checked the use of $CO_2$ itself and $pCO_2$ to make sure that they are expressed in the correct form in the revised text. In most cases, it is not in equilibrium with aqueous carbonates, which can be seen in Fig. 8.*
3. *We updated the division of x axis in some figures and the mistakes in the Fig.4. But the range of x axis only several days, e.g. in Fig.3.4.*
4. *Regarding to the references, we checked all the manuscript to make sure that the citations in the maintext are all consistent with the ones in the reference list. References by Liu and Zhao 2000, and Baker et al., 1998 and 2014 were cited in the previous manuscript but then cancelled*

*in the maintext without removing from the reference list.*

---

## Author Comment (AC2) · 24 May 2019

**Response to Referee #2**

**Major Comments**

This article uses environmental and isotope evidence from soil, stream water, and cave air to characterize the dynamics of carbon distribution in the Xueyu Cave system (China) and identify the contributions of potential reservoirs to overall cave air $CO_2$. The work is important because it builds on a growing set of literature describing how and why cave air $CO_2$ changes and has implications for interpretations of speleothem records used to reconstruct past climate. However, the paper is also missing key sections of the methodology, not all data is reported, and the discussion and data analysis are incomplete. The following areas require the authors' attention before publication:

1. **Manuscript grammar:** I appreciate that the authors may not be native English speakers, but sections of the manuscript are difficult to read. In particular, this hampered my understanding of the arguments the authors made in the discussion. I noted several sections that were unclear and need revision.

2. **Methodology:** Sections 2 and 3 are missing important information about sampling locations, sample collection (methodology, frequency, storage), and analysis methodology (instrumentation, standards). Measurements of d13C-atmosphere and d13C-plant material are reported but no methodology is provided.

3. **Data tables:** Not all data is reported in the tables, making it difficult to reproduce the authors' graphs and calculations. If there is not space in the main paper, data should be placed in a supplemental section or data repository.

4. **Discussion section:**

a. The mixing model employed for interpretation is not appropriate. Based on the authors' data, a model identifying CO2 sources must, at minimum, (1) include atmospheric $CO_2$ and (2) consider the close relationship between stream- and cave air-$CO_2$ concentration. The authors must also explain why they do not consider other potential sources listed in the introduction.

b. It is not clear to me that the November data really describe 'winter' conditions as cave air pCO2 does not drop to its 'winter baseline' until a week or two after the collection date. Do your isotope data represent baseline summer/winter cave conditions or only those during rain events?

c. The Discussion repeats information from the Results. I suggest a restructuring of the Discussion. In addition to your interpretation at this cave site the Discussion should focus on (1) comparison to previous studies of this nature and (2) the broader implications of the research for the cave community (e.g., studies of modern dripwater-calcite formation relationships and speleothem-based climate reconstructions).

**Answer to the major comments:**

We would like to say thanks to the referee for all his valuable comments for this paper, we added more background information in the methodology part.

1.We should say sorry that some parts of the manuscript are unreadable. We would check all the grammar problems in the revised version.

2. We added a new figure and also in the text part to present the sampling locations, sample collection (methodology, frequency, storage), and analysis methodology

(instrumentation, standards).

3. We would like to provide all data in supplementary table.

4. a. From the introduction part, we know that there are several potential sources of cave air $CO_2$ including degassing from $CO_2$-rich groundwater, advection and diffusion of vadose air, human respiration, the decomposition of organic matter, deep geogenic sources etc. However, according to the seasonal variations of cave air $CO_2$ concentration and related stable carbon isotope, we can neglect the atmospheric $CO_2$. Because if there atmospheric $CO_2$ outside the cave takes part in the mixing model, the carbon isotope should become positive with more inputs from the external air. However, in Xueyu Cave, we never found this phenomenon. Besides, the human respiration and decomposition of organic matter are excluded according to the previous study (Pu et al., 2016). Though we could not exclude the carbon source from ground air, because we did not have samples from the boreholes. Actually, the soil and vadose air $CO_2$ show similar range of stable carbon isotopes. In general, the $CO_2$ from soil and vadose may hold similar values of stable carbon isotopes though ground air $CO_2$ shows more negative values of stable carbon isotopes according to the review from Baldini et al. (2018).

b. Actually, the November data were collected at the transitional period when the cave air $CO_2$ concentration was decreasing during rain events. So it is not representative the 'winter baseline'.

c. We have written the Discussion where we added more information and widened the range of research implications.

**Specific Comments**
**Title**
A more informative title is "Constraining the source and dynamics of cave air $CO_2$ in a cave system in Xueyudong, Southwest China through $CO_2$ and $\delta^{13}C$ measurements"
**Answer**: Thanks for your comments, we accepted it as "Constraining the sources and dynamics of cave air $CO_2$ in Xueyu Cave system, Southwest China through $CO_2$ and $\delta^{13}C$ measurements"

**Abstract**
Line 20
Your abstract suggests that we need studies like this one to interpret stalagmite records, but does not tell the reader how this study contributes to our understanding of how to interpret speleothem records.
**Answer**: We think that the monitoring of modern cave air $CO_2$ can help to interpret the carbon isotope proxy in speleothems. However, we never analyzed the speleothems in this manuscript. We cancelled this sentence finally.

**1 Introduction**

The introduction could focus more attention on why we care about $CO_2$ concentrations. I gather that you are interested in caves as a source of proxy records – spend more time explaining the connection between cave $CO_2$ and speleothem records (as well as the current gaps in knowledge). The introduction should lead the author logically to the final sentence of the section (line 85) where you state the aims of the paper.

**Answer**: We arranged the introduction to make it more logical. We focused on explaining the sources and influencing factors of cave air $CO_2$ in previous version.

Line 69

Is this region dominated by C3 plants? Cite a reference for this if so.

Line 85

This section needs to be clearer. I suggest:

"The aim of this paper is to (1) identify the dynamics of carbon distribution and transfer between cave air $CO_2$, soil air $CO_2$, and stream $CO_2$, and to (2) identify the contributions of major reservoirs to overall cave air $CO_2$."

Line 88

Rephrase "The study area" to "Study area"

Line 89

More information is needed on the stream. Does it flow in/out of the cave? Or is it entirely underground? Pieces of information are available in the manuscript, but it should all be collected and put up front in this section.

**Answer**:

Line 69: The study area is dominated by C3 plants (evergreen broadleaf woods), the reference has been added.

Line 85: Thanks, we accepted it.

Line 88: Updated.

Line 89: More information about the stream has been updated, as "Most parts of the cave are narrow, deep passages (canyon passages), which are developed along strata and can be divided into three broad levels at 233–236 m (Level I), 249–262 m (Level II) and 281–283 m. (Level III) above sea level. A cave stream flows only in the bottom level (Pu et al., 2016). There is no allogenic stream sinking underground at the head of Xueyu Cave (Pu et al., 2015). The cave stream catchment is about 8–9 km2. Previous investigations by Zhu et al. (2004) and Pu et al. (2016) have described the hydrogeological and hydrochemical functioning of the Xueyu Cave stream. The cave stream is the only entrance of Xueyu Cave with an explored length of 1644 m. The discharge of the underground river ranges from 4.1 L/s in dry period to 26.6 L/s in wet period."

**Figure 1**

- Make all figure subsection labels (a, b, c, d) more obvious

- Legends on subsections b and c are too small

- 1C

o Is this figure after another paper? Needs to be cited if so

o Why is 'location of measured geological section' in here? You did not measure any sections

o Rephrase 'River/stream and its name' as 'River'

o Rephrase 'The curves that frame the Xueyu Cave' as 'Xueyu Cave outline'

- 1D

o The pictures of equipment are too small. Include them as separate sections of the figure or put them in the supplemental material

o The map needs a north arrow and scale bar

o The location of the stream needs to be better defined. Where does it enter/exit the cave?

- Caption

o Describe the inset in part a (the small map of China)

o Where are monitoring sites DK, LF, and MZ? They must be labeled

**Answer**:

We have updated the figure 1, using large-scale labels.

Fig.1C was modified from Wu et al. (2015), we put the citation notes. The 'location of measured geological section' should be related to another geological cross section from A to B. Here we did not use it and cancelled this part. 'River/stream and its name' and 'The curves that frame the Xueyu Cave' have been rephrased as 'River''Xueyu Cave outline'.

Fig.1D we would like to put the photos as supplementary material. A north arrow and scale bar have been posted. We labeled the entrance of the cave which is also the outlet of the cave stream.

Fig.1A include the maps of China and the province where Xueyu Cave is located. The sites LF and MZ correspond to X1 and X5 respectively. DK represents the MZ too. We make it consistent now.

**3 Methods and Materials**

All measurement types require more information so readers can assess the methodology.

For $CO_2$ concentration measurements:

- Automated measurements ($CO_2$-cave air, $CO_2$-soil, and $CO_2$-stream)

  Were all measurements made with the GMM221 sensor?

o How was the sensor modified for measurement of $CO_2$-stream? List part numbers if direct from manufacturer.

o Who is the sensor made by? Vaisala?

o How frequently were measurements made? What time periods were measured?

o How were the sensors calibrated? How often were they calibrated?

o What was the depth of measurement for soil $CO_2$?

**Answer:**

Yes, all measurements were made with the GMM221 sensor. The original sensor is made by Vaisala. But the equipment has been optimized when used in the stream.

-The measurements were performed every 15min, the reliability of sensor in the soil was calibrated on a monthly basis by the portable equipment that can insert into the soil for $CO_2$ measurement (portable pump-suction infrared $CO_2$ gas detector, measuring range $20×10^{-6}$~$20,000×10^{-6}$ pp with the precision$\leq±2\%$). Cave air $CO_2$ was determined with a calibrated CDU 440 $CO_2$ meter (measuring range $10×10^{-6}$~$20,000×10^{-6}$

ppm with the resolution of 10 ppm, made by Industrial Scientific, Pittsburgh, PA, USA). Besides, the sensor in the stream was difficult to be calibrated by another equipment, but logging data have been compared with $pCO_2$ that was calculated by

$$PCO_2 = \frac{(HCO_3^-)(H^+)}{K_1 Kco_2}$$ .

-The depth for measurement in the soil is 40cm.

- Precipitation and temperature
o List the part number(s) for the HOBO weather station
**Answer:**
Regarding to "Precipitation and temperature", I do not understand the "part number(s) for the HOBO weather station", do you mean the H21-SYS-A

- Discrete samples
o All discrete samples
   When were measurements made (list months, not summer/winter)?
   What was the time period of sampling (two 10-day periods)
   What were the frequency of measurements (1/day)?
   How were samples stored and transported? How much time elapsed between collection and measurement?
   How was CO2 concentration determined for the discrete samples of cave air, soil CO2, and DIC (i.e., data in Figure 6)
**Answer:**
o The regular measurements of hydrochemistry, samples for $d^{13}C_{DIC}$ analysis took place every month but samples for $d^{13}C_{soil}$ and $d^{13}C_{cave\ air}$ were collected only at the two periods (October-November 2014 and June-July 2015) on a daily basis. DIC Samples

were stored at 4℃ in the portable refrigerator and in the refrigerator in the lab. The

gas samples were kept at the room temperature (10-25℃).

o The measurement of gas isotope took place after the two concentrated sampling periods. Whereas the $d^{13}C_{DIC}$ was analyzed every 6 months. $CO_2$ concentrations of discrete samples for cave air, soil $CO_2$ and DIC analyses were recorded by portable equipment that we mentioned above (in the method part).
o The air gas was absorbed into a trace gas bag by a pump from open air to avoid the influence of human respiration. While the measurement of soil was more complex. A steel tube with holes at the bottom end was inserted into the soil at 40cm, the top end was sealed with a plastic cap. The gas was pumped into a trace gas bag (100ml) next day.
o The depth was at 40cm from surface and 100ml gas was collected. The quantity of soil gas is actually too small and the advection can be neglected.
o Water samples for $d^{13}C_{DIC}$ analysis were filtered and injected in 15ml brown bottles. Two drops of HgCl were added in order to prevent microbial activities. Then the bottle was sealed to make sure there were no bubbles inside. The samples were stored at

4℃ in the refrigerator until analysis. All processes followed the standards from the lab.

o The methods for d$^{13}$C-cave air and d$^{13}$C-plants have been updated. The leaves of plants (Pinus massoniana Lamb., Ficus virens, Bauhinia championii) were sampled for analysis as they are dominant in the catchment.

- Analysis
o What is the methodology for d13C analyses?
o What standards were used for d13C measurement?

**Answer:**

The measurement was performed at the Environmental Stable Isotope Lab, CAAS. The δ$^{13}$C of $CO_2$ in the trace gas bags was introduced to Delta V Plus. Internal laboratory $CO_2$-in-air standards were calibrated against calcium carbonate standards. The samples for DIC were stored at 4 degrees Celsius in the fridge until analyses which were performed using a Delta plus XL. The results were reported using V-PDB as the reference and the analysis precision was better than 0.15‰ (1σ). Plant leaves were collected in summer and winter 2014. The measurement of δ$^{13}$C in plants was based on vario PYRO cube elemental analyzer combined with ISOPRIME-100. The samples were combusted in a flow-type combustion flask under a continuous oxygen flow after being ground and passed through 100-mesh sieve. The oxygen gas containing the combustion products was carried by helium into successive magnesium perchlorate. $CO_2$ was separated through absorption column and injected into the IRMS. Lab standards were injected every 12 samples for calibration with the long standard deviation of 0.2‰.

Line 112
Be clear that samples collected for d13C- $CO_2$ analyses are not the same samples as those from the continuous collection regime.
Be more precise than "in summer and winter, respectively." The samples were collected once a day during two 10 day periods in November 2014 and June 2015. Also:
- Note that these are the same collection periods for d13C-cave air and –stream DIC
- Why were these time periods chosen?
- Why are there data gaps in the d13C data (e.g., DK air of Figure 3)?

**Answer:**

We make the periods more specific. The main reasons to choose the periods in October 2014 are due to the rainfall events and the transitional time for cave $CO_2$ decreasing. In Figure 3, there are no data gaps for DK. The reason should be explained that in the excel some samples in DK lack of "-", resulting in a no continuous line for DK. We update the figure.

**4 Results**
Line 127
"Soil CO2" needs to be "Soil $CO_2$"
Line 129

Soil CO2 concentrations bottom out around 4000 ppm in November 2014

Line 130

Why do you compare soil CO2 concentration at your site to these other studies? Do they have similar climate and vegetation regimes?

Line 131

Be consistent in using "soil moisture" instead of "humidity."

Line 134

If soil moisture controls respiration when temperature is suitable, what is occurring in summer 2015? It looks like there are time periods when pCO2 is high but soil moisture is low (July-Aigust).

**Answer:**

Line 127, corrected.

Line 129, corrected

Line 130, we just want to show the range of soil concentration.

Line 131, corrected

Line 134, we think that soil moisture is very important. Regarding to periods when $pCO_2$ is high but soil moisture is low, we would like to explain it because of time lag.

**Figure 2**

- Make the data gaps more obvious. Note in the text where these are and why they occurred

- The x-axis is difficult to read. Label it by month instead?

- Mark the d13C sampling intervals on here so it is obvious where to look for the 'zoomed-in' sections presented in Figures 3 and 4

- 2A

o I find the inverted y-axis confusing - precipitation should logically increase upwards

- 2B

o Include cave temperature on here as well (or at least the average)

**Answer:**

We will improve the figure to make x axis much clearer and also to mark the intervals in Fig.3 and Fig.4

Normally, precipitation should logically increase upwards, we used inverted y axis just to save space. But this kind of layout can be found in climatic figures.

Cave temperature is very stable, nearly horizontal, but we can put it in the new version.

Line 139

Rephrase "Cave parameters" to "Environmental measurements"

Line 140-141

- What are the "upper layer" and "lower layer?"

- The average cave temperatures are different from the average presented on line 93.

- Include cave temperature in Figure 2B

- What are the "three layers" – this is the first time this is mentioned in the text.

**Answer:**

Line 139, updated.

Line 140-141, they are not actually "upper layer" and "lower layer?", two sites are

located in the deepest cave and the entrance of the cave, respectively.

It's true that the average cave temperatures are different from the one presented in line 93. Because the previous one uses the general average value while the latter averages are based on the new records from 2014-2016. We will make it consistent in the new version.

"three layers" –we explain in "Study area", it is as following figure

[Figure]

The three-lay structure of Xueyu cave

Line 147

Typo "stream000000"

Line 151

Does cave CO2 decrease to atmospheric levels? It looks like it does from Figure 2

Line 157

- Could low cave CO2 concentrations be related to effective transport of cave air to the outside?

- In any event, this kind of interpretation should be left to the Discussion section

Line 157-159

I'm not clear on the meaning of this sentence.

Line 162

What is "less variability?" Define this.

**Answer:**

Line 147, cancelled as 'stream'.

Line 151, cave $CO_2$ decreased in winter, but still three times higher than atmospheric levels.

Line 157, the transport of cave air to the outside could be a reason to explain the low concentrations of cave air $CO_2$. We will move this part to the discussion section.

Line 157-159, the meaning of this sentence is that there is seasonal variation of cave air $CO_2$, but the rainfall events could disturb the seasonality and also bring variations in cave air $CO_2$.

Line 162, "less variability?" means that variational magnitudes in soil $CO_2$ concentration are less than that in cave air and stream $pCO_2$.

**Figure 3**

Figures 3 and 4 should be combined for ease of reference

- Precipitation should increase upwards

- Precipitation should be black, as in the other diagrams

- The same materials (e.g., CO2-cave air and d13C-cave air) should be the same line color and type

- Include error bars on d13C measurements

- "LF" and "DK" are not defined before Figure 3. Where are these sites?

- Caption

o Rephrase to "during rainfall events in October-November 2014"

**Answer:**

We will combine the two figures and adjusted the colors, direction and redraw it with error bars. LF and DK labels have been explained in the method section.

**Figure 4**

- The precipitation plot is labeled as air temperature

- Precipitation should increase upwards

- What are the high-frequency oscillations (6/29 and 7/22) in the cave temperature record? Were sensors replaced at this time?

- Where are the d13C measurements?

- The same materials (e.g., CO2-cave air and d13C-cave air) should be the same line color and type

- Include error bars on d13C measurements

- Caption

o Rephrase to "during rainfall events in June-July 2014"

**Answer:**

The precipitation label has been changed and about the y axis, we will adjust it. The high temperatures during the period 6/29-7/22 are normal because the air temperatures here are not the inside cave temperatures. $d^{13}C$ measurements in details will be added. We also accept other detailed comments, including to update the colors, caption and add error bars.

Line 168

Rephrase "4.4 The carbon isotope d13C in cave air and stream water" to "4.4 Carbon isotopes in cave air, stream water, and soil"

Line 169

Why cite Mattey et al. (2010) for atmospheric d13C measurements at the Rock of Gibraltar? There are long-term records of atmospheric CO2 that would be more directly relevant to your site

Line 170

- This is the first time that measurements of plant d13C are mentioned. Information about plant collection and measurement should go in the methodology

- What is the range of d13C-soil CO2?

- Remind readers of the depth of soil CO2 collection as this is a critical parameter for interpretation

**Answer:**

Line 168, rephrased.

Line 169, to cite Mattey et al. (2010) for atmospheric d$^{13}$C measurements at the Rock of Gibraltar is just to say that our results are very similar the value observed in the Rock of Gibraltar. Anyway, it is not so much matter.

Line 170, in the updated version, we put more information about the measurement in the method section.

The range of d$^{13}$C-soil $CO_2$ is from -18.0‰ to -23.9‰ at 40cm depth.

Line 173, changed.

- A decreasing then increasing trend is significant in the 'DK water' and 'DK air' data, but in 'LF water' and 'LF air', the increasing trend is still significant though the decreasing trend is not obvious.

Line 174, the stream information has been added in the 'Study area' section. The high flow is related to periods with more rainfall events, which result in large discharge.

**Figure 5**

- Plot needs error bars

- Why are the high resolution measurement periods not shown?

- Where are sites LF and MZ? Specify the 'upstream' and 'downstream' locations

- This data needs to be reported in a table (or in the supplemental information)

**Answer:**

We will update the figure with error bars and a new table in the revised version. The high-resolutions periods are shown in Fig. 3 and 4, so we just showed the monthly data in Fig.5.

Locations of LF and MZ(DK) can be found in Fig.1.

**5 Discussion**

Line 183

Rephrase 'lighter δ$_{13}$C to 'more negative δ$_{13}$C.' Values cannot be lighter or heaver. See, for example, table 2.1 in Sharp's Stable Isotope Geochemistry (https://digitalrepository.unm.edu/unm_oer/1/). Fix throughout the manuscript.

Line 184

- The values for d13C-cave air need to be reported in a table and the collection+analysis method need to be described in the Methodology

- 'cave air CO2 decreased at the beginning of the rain and then increased during the process at DK site.' There does not appear to be a strong initial decrease in the 'LF air' data and the 'DK air' data do not cover the entire time period. I suggest incorporating these observations into your interpretation

- When does the rain event start? This could be stated clearly here and be shown more clearly (vertical dotted lines?) in the graphs

Line 185

As noted above, it is not clear where the DK and LF sites are. I will not note further instances, but this needs to be addressed for the whole paper.

Line 186

Define 'the variability of d13C values'

Line 187

'the d13C-DIC values of stream water at two sites decreased and then increased during the rainfall events.' Depending on exactly when the rainfall event occurred, this may be true for site MZ. However, I see no overall change in the values for site LF.

Line 189-191

This sentence is unclear and appears to contradict itself. Please clarify how you are interpreting the relationship between soil gas and cave air.

**Answer:**

Line 183, thanks for your recommendation. The sentence has been rephrased.

Line 184, the background information has been put in the method section. 'cave air $CO_2$ decreased at the beginning of the rain and then increased during the process at DK site.' There does not appear to be a strong initial decrease in the 'LF air' data and the 'LF water' data could be explained by the fractures that transport $CO_2$ in gas forms. In the above part, we explain why 'DK air' data do not cover the entire time period. Actually, the trends in DK water and air are very similar. The rainfall events will be added in the new Fig.5.

Line 185, sorry to bring so much troubles, we make it clear in the Figure 1. DK and LF sites are X1 and X5, we will check through all the text in the revised version.

Line 186, 'the variability of d$^{13}$C values' means the variational magnitude.

Line 187, the answer is similar to the one in Line 184.

Line 189-191, we would like to say that stable carbon isotopes in cave air are very similar to that in soils. During the rainfall events, the variations can reflect the movement of soil $CO_2$.

**Figure 6**

- The y-axes on both plots should be the same to allow easy comparison
- The left plot has 'Steam $CO_2$ degassing,' which should be 'St**r**eam'
- The 'Stream CO2 degassing' data reported in this figure appear to be d13C-DIC values. Reporting these data as the d13C of CO2 in equilibrium with stream DIC requires calculation of the fractionation factor between DIC and CO2
- Keep the order the same for all graphs. Show November and then June (June is shown first in Figure 6)

**Answer:**

We made the consistent y axes in both plots. The 'St**r**eam' has replaced the 'stream $CO_2$ degassing'. Other comments are all accepted for revised Figure 6.

Line 200

'heavier d13C' should be 'higher d13C'

Line 200-202

- This sentence is unclear – is your intent to relate the d13C of respired organic matter to d13C of soil air CO2?

- Why are soil air measurements in Gibraltar relevant to your field site in SE China? Why not use your own measurements to make an estimate?

**Answer:**

Line 200, corrected.

Line 200-202, we use our monitoring data for estimation. We cite the results from Gibraltar just to compare with our data.

Line 205

I have the following issues with the discussion section:

- Why is a 2-endmember mixing model appropriate for your conceptual model? Several of your citations suggest a simple 2-endmember mixing model is inappropriate for understanding changes in cave air.

o You consider CO2 contributions from soil, stream, and human breath

o However, your introduction considers these additional sources important: atmospheric CO2, organic matter decay in the cave, magmatic/metamorphic sources

- Atmospheric air appears to be a particularly important endmember that this model does not address. The authors need to revise their data analysis to incorporate all of the information available from the dataset conceptual model of how/why cave air CO2 changes

- If >75 % of cave air CO2 is from the soil, why is there much better seasonal correlation between CO2-cave air and CO2-stream? Do your results apply only to rain events or year round?

- What causes the overall U-shape in the cave air and stream CO2 data every summer? Again, if soil CO2 is controlling cave air CO2, why is this signal not visible in the soil CO2 data?

- It is unclear to me from the discussion whether you think the soil, stream, or both are controls on cave air CO2. However, in the conclusions you definitively identify soil contributions as most important. Your position should be made clearer and should be supported by the isotope and CO2 concentration data.

- You briefly describe that d13C-DIC of the stream is controlled by flow rate (Line 174). Is there a relationship between stream flow rate and cave air CO2 or d13C?

- The discussion repeats results and repeats itself in sections. It should be edited for clarity and structure. I suggest the following general structure:

o Interpretation of what is occurring at Xueyu Cave

o Comparison to other studies of this nature

o Implications for developing paleoclimate records from speleothems (here and elsewhere)

**Answer:**

Line 205, two endmembers are simple. Though we introduced more sources, but from the filed monitoring, the magmatic/metamorphic sources and human breath can be excluded. We do not include the atmospheric $CO_2$ though it seems more reasonable to assume atmospheric $CO_2$ brings low $CO_2$ concentration in many caves. The reason is that if atmospheric $CO_2$ makes more contribution, the cave air $CO_2$ should show higher values of $d^{13}C$ and low $CO_2$ concentration in November (which can be seen in the Figure 6).

75 % of cave air $CO_2$ is from the soil, which makes the background of cave air $CO_2$, the close relationship between $CO_2$-cave air and $CO_2$-stream can be consider as the equilibrium between the air and water. The contribution calculation is based on rainfall events.

Soil $CO_2$ shows high correlations to temperature and soil moisture. The overall U-

shape in the cave air and stream $CO_2$ data every summer can be considered as accumulation of $CO_2$, not only related to soil $CO_2$ source, but also the transport way and the cave geometry. We have a conclusion that soil and stream are controls on cave air $CO_2$ based on isotope similarity and the consistent change of stream-cave air $CO_2$.

$d^{13}C_{DIC}$ variations are mainly controlled by sources, the interaction between water and rock.

Thanks for your suggestion. The discussion not just repeats results, because we wanted to make the result part and the discussion part to be consistent. We will adjust the structure to separate discussion well from the results.

Line 211

Is d13C-soil referring to soil organic matter or soil air CO2? If it refers to soil air CO2, keep in mind that d13C-soil air CO2 changes with depth. Justify using a single value.

**Answer:**

$d^{13}C$-soil refers to soil air, we always collected samples at the same depth.

Line 211-212

- A citation and explanation are needed for the 'd13C-CO2 from degassing -21.4 per mil due to isotopic fraction of 8 per mil.' Converting from DIC to the CO2 in equilibrium with it is not a straightforward connection for unfamiliar readers

- 'fraction' should be 'fractionation'

**Answer:**

Line 211-212, more explanation about background of stream degassing has been added.

'fraction' was changed to 'fractionation'

Line 213

I do not get the same output from your model using the values in Table 1

Line 214

Same as line 211-212 – a citation and explanation are needed for the fractionation between DIC-CO2

Line 219

'light d13Cco2' should be 'more negative d13C'

Line 228

How does water degassing CO2 not precipitate calcite?

**Answer:**

Line 213, we checked that there was mismatch between the Table and the text. We should have put the contribution from stream contribution not the $d^{13}C_{DIC}$ (considering the fractionation).

Line 214, 219, corrected.

Line 228, normally, the degassing companies with precipitating calcite. We just said that the precipitation is not significant.

**Figure 7**

- How is this model different from those proposed/used by other you cite? Might be better just to cite/describe the model.

- I did not understand that the river flowed from inside the cave to outside the cave until this figure – this information should be up front in the study area description

**Answers:**

This figure was abstracted from our study area. There are other models in previous studies, however, few figures with streams. So we think this figure can help readers to understand the main text better. More information about the stream we also explained in the 'Study area' section.

Line 277-278

This sentence is unclear: what does 'resulting in warm surface air into the cave companying with rainfall events' refer to?

Line 283

The terms 'S-pCO2' and 'C-pCO2' are confusing. I recommend not using them

Line 287

Delete the final sentence of this paragraph

**Answer:**

Line 277-278, we should have described more better, I mean that high-temperature water infiltration in summer always accompany with the rainfall events.

Line 283, we accept the comments to avoid using the 'S-pCO2' and 'C-pCO2'. Instead, we use $pCO_{2(cave\ air)}$ and $pCO_{2(stream)}$.

Figure 8

- Mark months of the year on the x-axis, not the 20th of each month

- Mark when the cave switches between summer and winter modes

**Answer:**

We accept the suggestions to mark the months and transitions.

Line 290

This section largely repeats what has been already said

Line 303

Where is the CO2 data for stream water at the two LF and DK sites? We are only presented with one dataset

Line 305

- This section is difficult to understand. I'm not sure what I am supposed to get out of it.

- Define the metrics 'before rain' and 'after rain,' response time, intensity, and equilibrium time

- Lines 311-316 do not seem to add to the section. If you are reporting results, they should be in the Results section

Line 309

'in consistent' should be 'inconsistent'

**Answer:**

Line 290, we will cancel the repeated part.

Line 303, the pattern between the air and water is similar in LF and DK, that is why we only put one to present the trend.

Line 305, we want to find if there is relationship between cave air $pCO_2$ the intensity and amount of rainfall events. 'before rain' and 'after rain' just the time that we collected the data. Response time and intensity refer to the lasting time of the rainfall events and their intensity, equilibrium time refers to the time it takes to make the balance between the stream and cave air $CO_2$.

Line 309, we still think it should be 'in consistent', because here we want to express that high frequency and high amount of rainfall events help to maintain the high concentration of cave air $CO_2$.

**6 Conclusions**

Line 331

Measurements were made two times in the year, not 'throughout the year'

Line 322

'$_{13}C$' should be '$\delta_{13}C$'

Line 333

The stated percentage contributions do not match Table 1

**Answer:**

Line 331, ok, we will accept this expression, that in two intervals not throughout the year.

Line 322, corrected.

Line 333, the table and the text are in consistent in the new one

**Table 1**

- The time transgressive values do not give the reader the idea that a single value of d13C-soil CO2 is assumed for the whole time period

- The table should include the d13C values for cave air, stream DIC, and calculated CO2 in equilibrium with stream DIC

**Answer:**

Yes, we will add the mentioned one to this part.

**Table 2**

- This table does not mean much to the reader as the parameters are not defined (intensity, response time, equilibrium time)

- This table can be moved to the supplemental information

**Answer:**

We will move this part to the supplemental material.

**References:**

Baldini, J. U. L., Bertram, R. A., & Ridley, H. E., (2018). Ground air: a first approximation of the earth's second largest reservoir of carbon dioxide gas. Science of The Total Environment, 616-617, 1007-1013.

Pu, J., Wang, A., Shen, L., Yin, J., Yuan, D., & Zhao, H., (2016). Factors controlling the

growth rate, carbon and oxygen isotope variation in modern calcite precipitation in a subtropical cave, southwest china. Journal of Asian Earth Sciences, 119(2), 167-178.

---

## Author Comment (AC3) · 24 May 2019

**Response to Referee #3**

In this contribution, the authors measured partial pressure of cave and soil gas CO2 and DIC concentrations in a cave stream. Those measurements were complemented by temperature (cave, soil and atmosphere) and precipitation observations. The logging was performed for more than two years. Furthermore, they provide the stable C isotopic composition of various components (stream water, soil and cave air) for two shorter periods of time in lower resolution (10 days, one sample per day). The aim of this study is to investigate the major sources of CO2 in the cave. Unfortunately, the manuscript is relatively difficult to read and I am not always sure, I understood what the authors wanted to say. In a possible revision, one focus should be on improvements with respect to the readability. Apart from this, I think, the study is not in a shape to be published. To my opinion, the text should be considerably improved before it could be considered for publication. There are many details, which should be improved and some sections should be reworked. However, I want to emphasize, that I think that a lot of the findings appear to not to be valid but that there are some potential within the data, to make it a quite nice contribution to the scientific community, which is worth to publish once the problems are solved, the text reworked and new ideas implemented.

**Answer:**

We would like to say thanks to the referee for all his valuable comments for this paper. We will pay more attention to solve the problems.

**Major points**

Throughout the text (e.g., line 9) when the authors write of 'pCO2 of cave water'. To me, this sounds very sloppy and should be prevented. I mean, gaseous CO2 (which can be well expressed as pCO2) is dissolved in water and then forms the different carbon species often referred to as dissolved inorganic carbon (DIC). The DIC cannot be expressed as pCO2. It is all dissolved and should be expressed in terms of concentrations. To my opinion the authors most likely want to express with their wording something similar to: 'the pCO2 the cave water is in equilibrium with'. This wording is much more precise and should be applied throughout the whole text.

**Answer:**

Yes, $pCO_2$ is assumed to be in equilibrium with the sampled waters by the equation

$$PCO_2 = \frac{(HCO_3^-)(H^+)}{K_1 K_{CO_2}}$$

The abstract should be reworked. In line 11 you are arguing that pCO2 of the cave air depend on wet and dry periods. But as I understood your data, the main influence is on the temperature. Only if temperature is warm enough, the wet/dry relationship is important. The last sentence of the abstract, appears weird for me as well after having read the whole paper. In this sentence, you are referring to stalagmite records and what your findings imply for those. However, in the manuscript you did not discuss the influence of the important cave processes on the stalagmite stable carbon proxy. So either you should delete this sentence or add some discussion about this in the text. I guess the latter would be the more valid approach, as this way your story will have more impact.

**Answer:**

Ok, we have to rewrite the abstract. The $CO_2$ production is mainly depending on temperature, but in hot summer, water is a very important factor to control it and the transport is also dominated by rainfall events. We wanted to make it more meaningful with the last sentence but it seems that it lacks some correlation. In the revised version, we will add some discussion in the text.

As the stream seems to play an important role in the cave, I think it is important to let the reader know some more details. E.g., for which distance the stream is in contact with the cave atmosphere, how much water is transported, how fast is its velocity and something like that. This could be well included in Section 2. This is important, as this way other researchers investigating other caves can set their observations in better context and comparison with your results is easier. I recognized that the cave is ascending into the host rock. However, with respect to cave CO2 and the derived ventilation regime, it behaves, according to your data, like those caves which are going downwards into the rock. This is quite interesting and could be much more emphasized. Maybe the stream is the key for this behavior? You should put some efforts here and elaborate more on this.

**Answer:**

Thanks for your suggestions, we added more information about the stream in the 'Study area' section. There are three lays in the cave space, which can be highly related to stream flow in different geological periods. In modern time, the stream flows through in the bottom layer. In the revised version, we will put more information about the geology and topography.

Section three should be completely reworked, as some information are given twice (e.g., about the CO2-sensor; Line 111 and 116) and others incomplete (How was soil CO2 measured? How was the sampling performed for soil gas d13C analysis? How was the pCO2, the stream water is in equilibrium with, determined?) or no information at all (e.g., about measurements of cave and soil temperature, soil moisture, d13C values of plants [according to line 170 they have been measured]).

**Answer:**

We were planning to introduce the equipment in Line 111 and present the monitoring frequency in Line116. The soil temperature and soil $CO_2$ concentrations were obtained since May 2013 by a composite measurement system, including a $CO_2$ sensor (GMM221, made by VAISALA in Finland with the resolution of 1 ppm and the range of 0-20000 ppm) and temperature and humidity sensor (AV-10T and AV-EC5 produced by AVALON, U.S.A with a resolution of 0.1 °C and 0.1%). All sensors were imbedded into the soil at the depth of 40 cm by drilling in the soil sampling site which is located about 40 m on the top with an elevation of about 300 m a.s.l. and 400 m a.s.l. in horizontal distance from the entrance of the cave (Fig.1). Above the cave, soils range from 0 to 50 cm in thickness. These soils are stony clays-rich and yellow soils that support evergreen forest and grainland (Field survey). Two sites inside the Xueyu Cave for $p$CO$_2$ monitoring of cave air and the subterranean stream have been selected at LF (X1) and MZ (X5), respectively (Fig. 1). The data were recorded each quarter

based on a GMM221 sensor (within the range 0~20000 ppm, precision ± 1%) connected with RR-1008 data receiving terminal, which were installed in the cave and stream to measure $CO_2$ concentrations. The sensor in the soil was calibrated on a monthly basis by the portable equipment that can insert into the soil for $CO_2$ measurement (portable pump-suction infrared $CO_2$ gas detector, measuring range $20 \times 10^{-6}$~$20,000 \times 10^{-6}$ ppm with the precision$\leq \pm 2$ %).Cave air $CO_2$ was determined with a calibrated CDU 440 $CO_2$ meter (measuring range $10 \times 10^{-6}$~$20,000 \times 10^{-6}$ ppm with the resolution of 10 ppm, made by Industrial Scientific, Pittsburgh, PA, USA). The sensor in the stream was difficult to be calibrated by another equipment, but logging data have been compared with $p$CO_2 that was calculated by hydrochemical parameters. To obtain the detailed hydrochemical variations, a CDTP300 multi-parameter water quality meter (made in Greenspan Corporation in Australia) was installed to record water temperature, water level, Ec and pH with resolutions of 0.01 °C, 0.01cm, 0.01 µS/cm and 0.01 pH units. '

Within the manuscript you are claiming: 'When the temperature is suitable in summer, soil moisture works as the main constraining factor for variations in soil CO2.' (e.g., Line 134). First, please don't discuss this in the results, but put it to the discussion and please be more detailed here. By eye, it is easily visible, that soil gas CO2 might be changed by soil moisture during the warm season. However, there seem to be a clear time delay between soil moisture and pCO2. I feel, you have to discuss this. Where is this delay coming from? Please, do some statistics, e.g., calculate correlation coefficient, determine time lag, : : : . You might be even able to make some statements about the sensitivity of soil CO2 variations with respect to soil humidity under various temperature ranges, as soil pCO2 seems also to react on soil humidity under lower temperatures. However, the sensitivity appears to be different when the variations in the cold and warm seasons are compared. This all, will have some influence on section

**Answer:**

Ok, we will separate results and discussion. Yes, there should be a time lag we did not pay more attention to. We have monitored but not put in this paper, the relationship between soil temperature, soil moisture and soil $CO_2$ on daily or seasonal scales.

Fig. 6: What exactly is plotted for the stream water? The measured d13C, ok. But how have you determined the x-axis value? Is this the pCO2 value, the stream water is in equilibrium with? But not all of this C can degas from the stream (due to the chemical limits of degassing) and thus cannot contribute to the cave air CO2. So the pCO2 what comes from the stream is for sure lower than the pCO2 value the water is in equilibrium with. Thus, those values should be put more to the right (the question is how much?). Due to all this, I think, providing this stream C source in the Keeling plot this way is quite bold without arguing why this can be done. This is then changing the available CO2 which can degas and is thus also changing the according mCi in your equation (line 205). As those plots contain the basis of your argumentation according to your present discussion, a change here, might have major influences on your final findings. Technical comments:

**Answer:**

In Fig. 6, we just wanted to show the difference of stable carbon isotope between two sites in the stream (upstream and downstream). The more positive values in the downstream can indicate that degassing had occurred. Because more d$^{13}$C was going to precipitate, and more negative d$^{13}$C will degas into air. We use the Keeling plot to show that the contributions from soil and stream. Why these two sources? We excluded human respiration, deep source and atmospheric air according to their concentrations or carbon isotopes.

**Technical comments:**

Line 26: 'always with higher values in summer and lower values in winter' I want to point out that this is not true. This is only true for caves, which lead downwards with increasing distance from the entrance. For those caves, which leads upwards, the opposite will be observed. But you have indicated this correctly, e.g., in line 43. So please, be consistent and more precise here.

**Answer:**

Thanks for your comments, that is true, what we said is just correct for some caves, especially the caves from the cited study. However, different cave modes show different circulation/ventilation characteristics, resulting in variations of cave $CO_2$.

Line 27-28: Unfortunately, I do not understand this sentence. Can you reword this?

Line 32: It seems to me, that 'although' is not fitting here.

Line 64: 'carbon' with a small first letter.

Line 65: 'Especially, in descending caves where carbon dioxide is heavier than the other main atmospheric components : : :' I am pretty sure you are meaning this differently than stated here. CO2 is always heavier than the main atmospheric components (O2 and N2), not only in descending caves. Maybe reword this to something like that: 'As carbon dioxide is heavier than the other main atmospheric components, CO2 accumulates in descending caves during the hot season due to the "cold trap effect".

**Answer:**

Line 27-28, this sentence was going to explain how soil $CO_2$ goes into the cave system. 'The $CO_2$ inputs that penetrate caves and become part of the karstic atmosphere via directly in gaseous form, dissolved $CO_2$ in infiltrated waters from the soil matter' was changed into 'Soil $CO_2$ can enter the cave soil gaseous form directly or be dissolved in the water and move with the water'.

Line 32, 'Although' was cancelled.

Line 64, 'Carbon' has been changed to 'carbon'.

Line 65, Ok, we accepted your expression, which is much clearer.

Line 73: The 'Thus' does not seem to fit here, as the previous sentence does not provide a reason for what you are stating in the following sentence, which you begun with 'Thus'.

Line75: Please define R/Ra. What is this?

Line 76-77: Please be more precise and reword this sentence as 222Rn is not produced from the decay of U and other radioactive atoms as you have stated here. It would be more precise

to write something similar to: '222Rn is a radioactive isotope that is naturally produced within the 238U decay chain. As it is heavier than air it is accumulating in the : : :'.

**Answer:**

Line 73, cancelled.

Line 75, R/Ra should be '$^{226}Ra/^{228}Ra$', which is the isotopic ratios of Ra.

Line 76-77, new sentence is like this: $^{222}Rn$ is a radioactive isotope that is naturally produced within the $^{238}U$ decay chain. As it is heavier than air it is accumulating in the $^{222}Rn$ is a radioactive gas that is naturally produced from the decay of uranium and other radioactive atoms in the carbonate host-rock in caves, which is accumulated in the subterranean atmosphere and usually covaries with $CO_2$ concentration.

Line 90: 'multiyear average precipitation' sounds a bit strange. Do you mean 'average annual precipitation'?

Line 91: Is the term 'secondary speleothems' correct? It seems to me, you might mean 'secondary carbonates' instead?

Line 93: Please provide information, of how constant cave air T is. Give its variation throughout the year and plot it maybe even in Fig. 2

**Answer:**

Line 90, Yes, it should be expressed as 'average annual precipitation'.

Line 91, Ok, normally, speleothems as secondary carbonates.

Line 93: We update in the Fig.2

Fig. 1: It is not clear, why the monitored drip sites are shown in d), as these are not discussed in the text. In addition, I am very confused about the naming of your river drip-sites. Here in Fig. 1 they are labelled as X1 and X5. In Tab. 1 they are labeled MZ and LF while throughout the text and in the figures they are labeled DK and LF. Please stay consistent.

**Answer:**

We corrected in the revised version. X1 and X5 are LF and MZ (DK). We did not discuss drip water, so we have to remove the drip water sites in the Fig.1d.

Line 103: 'cave' appears to be unnecessary here.

Line 105: Please, replace ',' by '.' after ')'.

Line 106: 'drip rate' instead of 'drip water rate'.

Line 126: 'amount' instead of 'amounts'

Line 128 and 131: Here you quite often make some statements, which should be carefully discussed before. To my opinion those statements do not belong in the results section, but should be shifted towards the discussion section.

Line 133-134: This belongs also to the discussion and it is not quite clear, what 'suitable' is meaning here. Please, be more precise. I also wonder if this statement is justified (see above).

**Answer:**

Line 103, cancelled

Line 105, 106, 126, corrected.

Line 128-131, Line 133-134, we will separate better the results without comments.

Fig. 2: Please indicate the cave air temperature.

**Answer:** we will do.

Line 137: Capital letters of 'Air' and 'Soil' should be small ones.

Line 146-147: Sorry, but I do not understand this sentence. Could you rewrite this under the correction of the 'stream00000' typo.

Line 156-163: This should be shifted to the discussion part. And I do not understand the sentence in Line 157. Also the following sentence (line 157-159) is not clear to me.

Line 169: Typo. Should be '-10 permil', shouldn't it?

Line 170: Please include here, that you are talking about soil gas, which has values of -18permill to -23.9.

**Answer:**

Line 137, corrected.

Line 146-147, According to Pu et al. (2018), the seasonality of cave $CO_2$ variations occurred based on monthly monitoring with high values in summer and low values in winter.

Line 156-163, we moved this part to the discussion part and improved like this 'During the transitional periods between different seasons, $CO_2$ concentration varied sharply. Especially with rainfall events, cave $CO_2$ concentrations changed largely due to increased high-$CO_2$ flow. A high-frequency monitoring in October-November 2014 and June-July 2015 showed the detailed changes of $pCO_2$ and carbon isotopes during rainfall events.'

Line 169, corrected.

Line 170, yes, we were talking about soil gas.

Line 183-185: I am sorry. From my side, this does not look as contemporaneously as you described it. Please provide some shaded rectangles in the appropriate figures to allow to better follow your argumentation. For me it even seems, that there is no rain event in October, which was also covered with DIC and $pCO_2$ measurements at DK. In November are some rain events, but there, I could not see the described relationship. DK-d13C is already decreasing before the rain event. Thus, it appears the decrease in d13C has nothing to do with the single rain event. And to be honest, the observation of such behavior (if there would be indeed one) during only one single event is not very convincing.

**Answer:**

Line 183-185, there were a little rain in October, but we mainly monitored the rainfall event at the end of October. We will mark it in Fig. 2. DK-d$^{13}$C changed during the rainfall event. Of course, it is not so convincing with one/two rainfall events. However, it needs a lot of work to monitor many rainfall events. At present, we could not afford to do such things.

Line 186-188: Please also show this in a figure. With only seeing the numbers in the table, I am not able to follow.

Line 205: Please subscript the 'i' in the equation.

Line 207-209: Why has the mixing model only two endmembers? What is with the atmospheric input? Why can this be neglected. Please explain. But then, an endmember modelling with three

sources is much more difficult. (even with the assumed two end-members, the equation in line 205 needs an additional equation to order to be solved. [Sum of all mCi = 1])

Line 211-212: Please, cite the work you are taking the fractionation factor from.

**Answer:**

Line 186-188, we will present in a figure.

Line 205, 'i' in the equation refers to different sources, in this paper, soil $CO_2$ and stream degassing.

Line 207-209, We excluded other endmembers. Atmospheric input is not considered as a direct input due to the carbon isotope. If atmospheric input lowers the winter $CO_2$ concentration, the cave $CO_2$ concentration and isotope should have closed to atmospheric end.

Line 211-212, cited.

Line 212-216: Even with your two end-member modelling, I am somewhat confused, about your numbers for the contribution of the C sources to the cave air. There is quite some scatter in the d13C of soil gas, cave air and DIC (or degassed CO2) of the stream. The problem then is that the difference between stream degassed CO2 and soil air is quite small compared to the scatter in your values. This makes the calculation quite difficult. At least, I would require to give some error estimates in your values here.

Line 230: Why are you mentioning roots here. Are there roots in your cave? I assume not, otherwise I would have expected some description of them earlier in Section 2. So it is not clear to me, why you are mentioning them here with respect to other caves. This sentence is also completely out of context with the sentence before and after.

**Answer:**

Line 212-216, We will put the error bars in the figures.

Line 230, yes, no roots in the cave, we removed this part.

Line 280: Mawmluh cave was not investigated by Riechelmann et al., 2017. They investigate Bunker Cave. Please change the citation accordingly.

Line 292: Why do you say that cave CO2 and stream DIC are in equilibrium with each other? For me this looks quite different in Fig. 8. There is barely a time, when the difference between both is 0. Even if this would be the case, all your argumentation from above that the stream is a significant CO2 source is destroyed by this sentence. If both would be in equilibrium, no CO2 can be degassed from the stream and could not provide CO2 to the cave atmosphere. And if all the CO2 of the stream would have been degassed earlier, you could also not do any end-member modelling as you do not have the initial conditions of the water.

Line 308: Please define and explain, what the response time is. How have you calculated this? Can you show a plot of the observed relationship?

**Answer:**

Line 280, corrected.

Line 292, we said they are in equilibrium because the difference between aquatic and air $pCO_2$ are not so large. Actually, they are not in equilibrium, we can see that they fluctuated within 1000 ppm in most of time. We can say that degassing occurred during the two intervals, but we do not know if it is true throughout the year. However, we can

make sure that there is exchange between air and water during the monitoring intervals. Line 308, the response time means that the time it takes to be in equilibrium. We can see rainfall events make the large difference of $pCO_2$ between air and water, degassing does not always take place until its concentration become stable.

---

## Author Comment (AC4) · 17 Jun 2019

**Response to Referee #1**

**General comments**

The review of the article "Constraining the soil carbon source to cave-air $CO_2$: evidence from the high-time resolution monitoring soil $CO_2$, cave-air $CO_2$ and its $\delta^{13}C$ in Xueyudong, Southwest China"by Min Cao, Yongjun Jiang, Jiaqi Lei, Qiufang He, JiaxinFan, Ze Zeng. The authors present the data on $CO_2$ in the soil, cave stream, and cave atmosphere (Xueyu Cave,China) and its surrounding. The data weregathered during the period of 2015-2016. The aim of the article is (1) to understand the quantitative relationship between all the forms of $CO_2$, (2) to reveal their sources, and (3) to understand the factors that control the cave air $CO_2$ variations. The topic of the article is important and is worthy of publication. In the article, however, there are some aspects that require revision and other ones that could be substantially improved before publishing. My main reservation is that the conclusions should be better proved by a data analysis (e.g., Cross-correlation Analysis). The results of the data analysis should be presented and discussed in detail. The data sets are nice, but they could be much better presented. The x-axis should be more extended in order to be better distinguishable individual fluctuations in the variables.

**Answer to general comments:**

We would like to thank the referee for his generally positive comments. We will pay more attention in presenting and explaining the our data in the final version.

We posted a table of the correlation analysis:

Table 1 The correlation matrix of environmental parameters in Xueyu system

| | Soil M | Soil T | Prep | Cave T | Soil CO$_2$ | Discharge | pH | MZ stream CO$_2$ | MZ air CO$_2$ | LF stream CO$_2$ | LF air CO$_2$ | Spc | TOC |
|---|---|---|---|---|---|---|---|---|---|---|---|---|---|
| Soil M | 1.00 | | | | | | | | | | | | |
| Soil T | .285** | 1.00 | | | | | | | | | | | |
| Prep | -.023** | -.013** | 1.00 | | | | | | | | | | |
| Cave T | .326** | .367** | -.040** | 1.00 | | | | | | | | | |
| Soil CO$_2$ | .263** | .639** | -.027** | .116** | 1.00 | | | | | | | | |
| Discharge | -.062** | .011* | .217** | -.122** | -.027** | 1.00 | | | | | | | |
| pH | .044** | 0.00 | -.094** | .296** | -.052** | -.278** | 1.00 | | | | | | |
| MZ stream CO$_2$ | .189** | .416** | .073** | -.192** | .294** | .224** | -.735** | 1.00 | | | | | |
| MZ air CO$_2$ | -.589** | .518** | .052** | -.795** | .683** | .222** | -.989** | .868** | 1.00 | | | | |
| LF stream CO$_2$ | .030** | .402** | .054** | -.237** | .263** | .304** | -.926** | .655** | .877** | 1.00 | | | |
| LF air CO$_2$ | -.030** | .423** | .059** | -.210** | .237** | .253** | -.904** | .768** | .963** | .952** | 1.00 | | |
| Spc | .134** | .227** | .077** | -.305** | .062** | .253** | -.740** | .610** | .957** | .749** | .710** | 1.00 | |
| TOC | .190** | -.540** | -.023** | -.447** | -.176** | -.046** | -.194** | -.596** | -.570** | -.080** | -.209** | .111* | 1.00 |

**. P<0.01; *. P<0.05; Soil M=Soil moisture, Soil T=Soil temperature, Prep=Precipitation, Cave T=Cave temperature

We updated the text in the dicussion part '4.3 Environmental parameters and their correlation':

"There are significant correltations between stream $pCO_2$ and cave air $CO_2$, especially at LF site ($R^2$=0.95, p<0.01). The correlation between soil $CO_2$ and soil temperature is significant too ($R^2$=0.64, p<0.01)."

We put more details in the study area part, the method part and discussion part, and refined the conclusions too:

" 1) Two-year monitoring study of soil $CO_2$, subterranean stream and cave air $CO_2$ concentration reveals that a dynamic equilibrium between $CO_2$ sources and sinks.   Seasonal dynamics took place with the minimum cave air $CO_2$ concentrations during winter and peaks in November.

2)High-resolution monitoring of $CO_2$ concentrations in the soil and cave system may allow us to estimate the potential cave air $CO_2$ sources in Xueyu Cave (subterranean stream, air from vadose/soil zone). Throughout the year, $\delta^{13}C_{DIC}$ showed higher values in winter but lower values in summer. $\delta^{13}C$ of different endmembers showed that soil $CO_2$ made more contribution of C to the cave air $CO_2$ in June (75.6%) than in November (65.9%), and the second source was the degassing of stream. The accumulation of cave air $CO_2$ concentration maintains the high values in summer due to the confined space.

3)The seasonal variations in cave air $CO_2$ concentration were very similar to that in stream $pCO_2$, showing which shows high but fluctuated values in summer and steady but low values in winter. Stream water seems to be a constant source of $CO_2$ as an increase of up to 5800 ppm in 2 hours was observed and $CO_2$ degassing occurred after strong rain events. In winter, stream water is the carbon sink of cave $CO_2$.

4)Cave air $CO_2$ concentrations are similar at different sites in Xueyu Cave. The anthropogenic impact of visitors to cave air $CO_2$ concentrations is evident from the hourly fluctuations, but not significant on daily or longer time scales."

**Other comments:**   Throughout the text, it is important to distinguish $CO_2$ itself from $CO_2$ concentration and $pCO_2$ (e.g., the lines/paragraph 85). The expression "$PCO_2$ in the water" (stream $pCO_2$ is acceptable only as an abbreviation in the text. Furthermore, it is important to explain that it means $pCO_2$ of gaseous $CO_2$ that would be in equilibrium with aqueous carbonates. In   principle, $pCO_2$   is   dimensionless   variables   (or   it   has   units   of   pressure).   If   the   $CO_2$ quantity is given in ppmv units, it means "$CO_2$ concentration".

Some soil characteristics should be given in the paragraph Study Area. More detail information should be given in monitoring/calculating of the stream $CO_2$ in the paragraph Methods and Materials.

The x-axes in the plots (Fig. 2, 3, 4, 5) should be better divided (e.g., by one month, three months, etc.). The secondary y-axis in Fig. 4 should represent "Precipitation". I do not understand what the conceptual model in Fig. 7 brings new/beneficial. In the text, there are missing the citation: Liu and Zhao 2000, and Baker et al., 1998 and 2014, referenced in the Reference list.

**Answer to other comments:**

1. We checked the use of $CO_2$ itself and $pCO_2$, $CO_2$ concentration to make sure that they are expressed in the correct form in the revised text. Actually, $pCO_2$ has unit, such as Pa, but we use ppm in $CO_2$ quantity to make it simple and comparable between air and water.

2. In most cases, it is not in equilibrium with aqueous carbonates, which can be seen in Fig. 8.

3. The x-axis had been adjusted in order to be better distinguishable individual fluctuations in the variables. The figures 2 and 3 can be seen:

4. Regarding to the references, we checked all the manuscript to make sure that the citations in the maintext are all consistent with the ones in the reference list. References by Liu and Zhao 2000, and Baker et al., 1998 and 2014 were cited in the previous manuscript but then cancelled in the maintext without removing from the reference list.

**Corrected:**

The changed x-axis in new Fig.3 and 4

[Figure]

[Figure]

So in the revised version, we cancelled the following part:

Baker, A., Genty, D., Dreybrodt, W., Barnes, W. L., Mockler, N. J., and Grapes, J.: Testing theoretically predicted stalagmite growth rate with recent annually laminated samples: implications for past stalagmite deposition, Geochim. Cosmochim. Ac., 62(3), 393-404, https://doi.org/10.1016/S0016-7037(97)00343-8, 1998.

Baker, A. J., Mattey, D. P., and Baldini, J. U. L.: Reconstructing modern stalagmite growth from cave monitoring, local meteorology, and experimental measurements of dripwater films, Earth Planet. Sc. Lett., 392(392), 239-249, https://doi.org/10.1016/j.epsl.2014.02.036, 2014.

Liu, Z., and Zhao, J.: Contribution of carbonate rock weathering to the atmospheric CO₂ sink, Environ. Geol., 39(9), 1053-1058, https://doi.org/10.1007/s002549900072, 2000.

---

## Author Comment (AC5) · 17 Jun 2019

**Response to Referee #2**

**Major Comments**

This article uses environmental and isotope evidence from soil, stream water, and cave air to characterize the dynamics of carbon distribution in the Xueyu Cave system (China) and identify the contributions of potential reservoirs to overall cave air CO2. The work is important because it builds on a growing set of literature describing how and why cave air CO2 changes and has implications for interpretations of speleothem records used to reconstruct past climate. However, the paper is also missing key sections of the methodology, not all data is reported, and the discussion and data analysis are incomplete. The following areas require the authors' attention before publication:

1. **Manuscript grammar:** I appreciate that the authors may not be native English speakers, but sections of the manuscript are difficult to read. In particular, this hampered my understanding of the arguments the authors made in the discussion. I noted several sections that were unclear and need revision.

2. **Methodology:** Sections 2 and 3 are missing important information about sampling locations, sample collection (methodology, frequency, storage), and analysis methodology (instrumentation, standards). Measurements of $d^{13}C$ -atmosphere and $d^{13}C$ -plant material are reported but no methodology is provided.

3. **Data tables:** Not all data is reported in the tables, making it difficult to reproduce the authors' graphs and calculations. If there is not space in the main paper, data should be placed in a supplemental section or data repository.

4. **Discussion section:**

a. The mixing model employed for interpretation is not appropriate. Based on the authors' data, a model identifying CO2 sources must, at minimum, (1) include atmospheric CO2 and (2) consider the close relationship between stream- and cave air-CO2 concentration. The authors must also explain why they do not consider other potential sources listed in the introduction.

b. It is not clear to me that the November data really describe 'winter' conditions as cave air pCO2 does not drop to its 'winter baseline' until a week or two after the collection date. Do your isotope data represent baseline summer/winter cave conditions or only those during rain events?

c. The Discussion repeats information from the Results. I suggest a restructuring of the Discussion. In addition to your interpretation at this cave site the Discussion should focus on (1) comparison to previous studies of this nature and (2) the broader implications of the research for the cave community (e.g., studies of modern dripwater-calcite formation relationships and speleothem-based climate reconstructions).

**Answer to the major comments:**

We would like to say thanks to the referee for all his valuable comments for this paper, we added more background information in the methodology part.

1.We should say sorry that some parts of the manuscript are unreadable. We checked all the grammar problems in the revised version.

2. We added a new figure 2 in the text part to present the sampling locations, sample collection (methodology, frequency, storage), and analysis methodology (instrumentation, standards).

3. We would like to provide all the raw data in a supplementary table, and the data also exceed the excel limit, but it's difficult to list all.

4. a. From the introduction part, we know that there are several potential sources of cave air $CO_2$ including degassing from $CO_2$-rich groundwater, advection and diffusion of vadose air, human respiration, the decomposition of organic matter, deep geogenic sources etc. However, according to the seasonal variations of cave air $CO_2$ concentration and related stable carbon isotope, we can neglect the atmospheric $CO_2$. Because if atmospheric $CO_2$ outside the cave takes part in the mixing model, the carbon isotope should become positive with more inputs from the external air. However, this phenomenon in Xueyu Cave never happened. Besides, the human respiration and decomposition of organic matter are excluded according to the previous study (Pu et al., 2016). Though we could not exclude the carbon source from ground air, because we did not have samples from the boreholes. Actually, the soil and vadose air $CO_2$ show similar range of stable carbon isotopes. In general, the $CO_2$ from soil and vadose may hold similar values of stable carbon isotopes though ground air $CO_2$ shows more negative values of stable carbon isotopes according to the review from Baldini et al. (2018). We think that both are the endmembers with more negative $\delta^{13}C$.

b. Actually, the November data were collected at the transitional period when the cave air $CO_2$ concentration was decreasing during rain events. So it is not representative the 'winter baseline'. In

the text, we use the specific months, instead of winter and summer.

c. We have written the Discussion where we added more information and widened the range of research implications.

**Specific Comments**

**Title**

A more informative title is "Constraining the source and dynamics of cave air CO2 in a cave system in Xueyudong, Southwest China through CO2 and $\delta^{13}C$ measurements"

**Answer**:

Thanks for your comments, we accepted it as "Constraining the sources and dynamics of cave air $CO_2$ in Xueyu Cave system, Southwest China through $CO_2$ and $\delta^{13}C$ measurements"

**Abstract**

Line 20

Your abstract suggests that we need studies like this one to interpret stalagmite records, but does not tell the reader how this study contributes to our understanding of how to interpret speleothem records.

**Answer**: We think that the monitoring of modern cave air $CO_2$ can help to interpret the carbon isotope proxy in speleothems. However, we never analyzed the speleothems in this manuscript. We cancelled this sentence finally.

**1 Introduction**

The introduction could focus more attention on why we care about $CO_2$ concentrations. I gather that you are interested in caves as a source of proxy records – spend more time explaining the connection between cave $CO_2$ and speleothem records (as well as the current gaps in knowledge). The introduction should lead the author logically to the final sentence of the section (line 85) where you state the aims of the paper.

**Answer**:

We arranged the introduction to make it more logical. Our logical structure: the current cave air $CO_2$, their sources and dynamics and carbon isotopes. We start the introduction part like this:

"In karst regions, carbon dioxide ($CO_2$) concentrations in epikarst (especially from soils) largely affect karst landscapes (Ford and Williams, 2007). Shallow caves are widely distributed in the terrestrial environment and contain a significant volume of underground air with high concentrations of $CO_2$ (Wood, 1985; Faimon et al., 2006; Bourges et al., 2014). According to Ek and Gewelt (1985) and Baldini (2010)'s reviews, the earliest measurements of $CO_2$ in cave air date from 1859. Modern sensors and logging techniques have been deployed to provide detailed records of $CO_2$, temperature and humidity in cave atmospheres (Spötl et al., 2005; Frisia et al., 2011; Bourges et al., 2014). In all cases the cave air $CO_2$ concentration is greater than in the open atmosphere, a proper understanding of the causes and dynamics of seasonality in cave air $CO_2$ is fundamental for speleothem palaeoclimatology (Fairchild and Baker, 2012)."

Line 69

Is this region dominated by C3 plants? Cite a reference for this if so.

Line 85

This section needs to be clearer. I suggest:

"The aim of this paper is to (1) identify the dynamics of carbon distribution and transfer between cave air CO2, soil air CO2, and stream CO2, and to (2) identify the contributions of major reservoirs to overall cave air CO2."

Line 88

Rephrase "The study area" to "Study area"

Line 89

More information is needed on the stream. Does it flow in/out of the cave? Or is it entirely underground? Pieces of information are available in the manuscript, but it should all be collected and put up front in this section.

**Answers**:

Line 69: The study area is dominated by C3 plants (evergreen broadleaf woods), the reference has been added. "The light end-member source should be located close to the roots of C3 type vegetation that is representative of evergreen broadleaf woods (Pu et al., 2016)."

Line 85: Thanks, we accepted it: "The aim of this paper is to (1) identify the dynamics of carbon distribution and transfer between cave air $CO_2$, soil air $CO_2$, and stream $CO_2$ and (2) to identify the contributions of major reservoirs to overall cave air $CO_2$."

Line 88: Updated. "The study area" has been changed into "Study area".

Line 89: More information about the stream has been updated, as "Most parts of the cave are narrow, deep passages (canyon passages), which are developed along strata and is composed of three levels of passages: at 233–236 m (Level I), 249–262 m (Level II) and 281–283 m (Level III), separately. A cave stream flows at the bottom level (Pu et al., 2016). There is no allogenic stream sinking underground at the head of Xueyu Cave (Pu et al., 2015). The cave stream catchment is about 8–9 km$^2$. Previous investigations by Zhu et al. (2004) and Pu et al. (2016) have described the hydrogeological and hydrochemical functioning of the Xueyu Cave stream. The stream is the only entrance of Xueyu Cave with an explored length of 1644 m. The discharge of the underground river ranges from 4.1 L/s in dry period to 26.6 L/s in wet period."

**Figure 1**

- Make all figure subsection labels (a, b, c, d) more obvious

- Legends on subsections b and c are too small

- 1C

o Is this figure after another paper? Needs to be cited if so

o Why is 'location of measured geological section' in here? You did not measure any sections

o Rephrase 'River/stream and its name' as 'River'

o Rephrase 'The curves that frame the Xueyu Cave' as 'Xueyu Cave outline'

- 1D

o The pictures of equipment are too small. Include them as separate sections of the figure or put

them in the supplemental material

o The map needs a north arrow and scale bar

o The location of the stream needs to be better defined. Where does it enter/exit the cave?

- Caption

o Describe the inset in part a (the small map of China)

o Where are monitoring sites DK, LF, and MZ? They must be labeled

**Answer**:

We have updated the figure 1, using large-scale labels.

[Figure]

Figure 1

Fig.1C was modified from Wu et al. (2015), we put the citation notes. The 'location of measured

geological section' should be related to another geological cross section from A to B. Here we did

not use it and cancelled this part. 'River/stream and its name' and 'The curves that frame the Xueyu

Cave' have been rephrased as 'River' and 'Xueyu Cave outline', respectively.

Fig.1D we would like to put the photos as supplementary material. A north arrow and scale bar have

been posted. We labeled the entrance of the cave which is also the outlet of the cave stream (see the

Figure 1 above).

Fig.1A includes the maps of China and the province where Xueyu Cave is located. The sites LF and MZ correspond to X1 and X5, respectively. DK represents the MZ too. We make it consistent now.

**3 Methods and Materials**

All measurement types require more information so readers can assess the methodology.

For $CO_2$ concentration measurements:

- Automated measurements ($CO_2$-cave air, $CO_2$-soil, and $CO_2$-stream)

  Were all measurements made with the GMM221 sensor?

o How was the sensor modified for measurement of $CO_2$-stream? List part numbers if direct from manufacturer.

o Who is the sensor made by? Vaisala?

o How frequently were measurements made? What time periods were measured?

o How were the sensors calibrated? How often were they calibrated?

o What was the depth of measurement for soil $CO_2$?

**Answer:**

Yes, all measurements were made with the GMM221 sensors. The original sensor is made by Vaisala.

-The measurements were performed every 15min, the reliability of sensor in the soil was calibrated on a monthly basis by the portable equipment that can insert into the soil for $CO_2$ measurement (portable pump-suction infrared $CO_2$ gas detector, measuring range $20 \times 10^{-6} \sim 20,000 \times 10^{-6}$ pp with the precision$\leq \pm 2\%$). Cave air $CO_2$ was determined with a calibrated CDU 440 $CO_2$ meter (measuring range $10 \times 10^{-6} \sim 20,000 \times 10^{-6}$ ppm with the resolution of 10 ppm, made by Industrial Scientific, Pittsburgh, PA, USA). Besides, the sensor in the stream was difficult to be calibrated by another equipment, but logging data have been compared with $pCO_2$ that was calculated through

equation: $P_{CO_2} = \dfrac{(HCO_3^-)(H^+)}{K_1 K_{CO_2}}$ .

-The depth for measurement in the soil is 40cm.

"A set of system for continuous and automatic soil $CO_2$ measurement with a $CO_2$ sensor was fixed in the soils above Xueyu Cave (Fig.2). The soil temperature and soil $CO_2$ concentrations were

obtained from October 2014 by a composite measurement system, including a $CO_2$ sensor (GMM221, made by VAISALA in Finland with the resolution of 1 ppm and the range of 0-20000 ppm) and temperature and humidity sensor (AV-10T and AV-EC5 produced by AVALON, U.S.A with a resolution of 0.1 °C and 0.1%). All sensors were imbedded into the soil at the depth of 40 cm by drilling in the soil sampling site which is located about 40 m on the top with an elevation of about 300 m a.s.l. and 400 m a.s.l. in horizontal distance from the entrance of the cave (Fig.2). Above the cave, soils range from 0 to 50 cm in thickness. These soils are stony clays-rich and yellow soils that support evergreen forest and grainland (Field survey)."

[Figure]

Figure 2 Cross section of Xueyu Cave passages and the sampling locations

"Two sites inside the Xueyu Cave for monitoring of cave air and the subterranean stream were located at LF (X1) and MZ (X5) (Fig. 2). GMM221 sensors with RR-1008 data logging were installed in the cave and stream to measure $CO_2$ concentrations. The sensor in the soil was calibrated on a monthly basis by the portable equipment that can insert into the soil for $CO_2$ measurement (portable pump-suction infrared $CO_2$ gas detector, measuring range $20\times10^{-6}\sim20,000\times10^{-6}$ pp with the precision$\leq\pm 2$ ％ ).Cave air $CO_2$ was determined with a calibrated CDU 440 $CO_2$ meter (measuring range $10\times10^{-6}\sim20,000\times10^{-6}$ ppm with the resolution of 10 ppm, made by Industrial Scientific, Pittsburgh, PA, USA). The sensor in the stream was difficult to be calibrated by another equipment, but logging data have been compared with $p$CO_2 that was calculated by hydrochemical parameters. To obtain the detailed hydrochemical variations, a CDTP300 multi-parameter water quality meter (made in Greenspan Corporation in Australia) was installed to record water

temperature, water level, Ec and pH with resolutions of 0.01 °C, 0.01cm, 0.01 µS/cm and 0.01 pH units. Both the $CO_2$ measurement system and the water quality data logger were set at the same time-interval of 15 min."

- Precipitation and temperature

o List the part number(s) for the HOBO weather station

**Answer:**

We added in the text:

"Meteorological data including precipitation (with the precision of 0.01 mm) and temperature (with the precision of 0.1 °C) were recorded every 15 min using a HOBO weather station (H21-SYS-A)."

- Discrete samples

o All discrete samples

When were measurements made (list months, not summer/winter)?

What was the time period of sampling (two 10-day periods)

What were the frequency of measurements (1/day)?

How were samples stored and transported? How much time elapsed between collection and measurement?

How was CO2 concentration determined for the discrete samples of cave air, soil CO2, and DIC (i.e., data in Figure 6)

**Answer:**

o The regular sampling took place every month, but the samples for $\delta^{13}C_{CO2}$ analysis were collected in several days of November 2014 and June 2015. The measurement of gas isotope was carried out after the two concentrated sampling periods. $CO_2$ concentrations of discrete samples for cave air, soil $CO_2$ and DIC analysis were recorded by portable equipment that we mentioned above (in the method part). The air gas was absorbed into a trace gas bag by a pump from open air to avoid the influence of human respiration. The depth for soil $CO_2$ was at 40cm from surface and 100ml gas was collected. The quantity of soil gas is actually too small and the advection can be neglected. The

methods for $d^{13}C$-cave air and $d^{13}C$-plants have been updated. The leaves of plants (Pinus massoniana Lamb., Ficus virens, Bauhinia championii) were sampled for analysis as they are dominant in the catchment.

"A steel tube with holes at the bottom end was inserted into the soil at 40cm, the top end was sealed with a plastic cap. The gas was pumped into a trace gas bag (100ml) next day. Soil and cave air/stream samples for $\delta^{13}C_{CO2}$ analyses were collected using a pump and carefully sealed in the 100ml trace gas bags, and shipped to the Southwest University in during two periods (10[th]-20[th], June and 30[th] October-8[th] November) on a daily basis. All samples were stored at 4℃ before pretreatments for analysis. The measurement was performed at the Environmental Stable Isotope Lab, CAAS. The $\delta^{13}C$ of $CO_2$ in the bags was introduced to Delta V Plus. Internal laboratory $CO_2$-in-air standards were calibrated against calcium carbonate standards. DIC samples were filtered and injected in 15ml brown bottle without bubbles and 2 drops of $HgCl_2$ were added in order to prevent microbial activities, the samples then were stored at 4℃ in the portable refrigerator and in the refrigerator in the lab.

Analyses were performed using a Delta plus XL. The results were reported using V-PDB as the reference and the analysis precision was better than 0.15‰ (1σ). Plant leaves were collected in summer and winter 2014. The measurement of $\delta^{13}C$ in plants was based on vario PYRO cube elemental analyzer combined with ISOPRIME-100. The samples were combusted in a flow-type combustion flask under a continuous oxygen flow after being ground and passed through 100-mesh sieve. The oxygen gas containing the combustion products was carried by helium into successive magnesium perchlorate. $CO_2$ was separated through absorption column and injected into the IRMS. Lab standards were injected every 12 samples for calibration with the long standard deviation of 0.2‰."

- Analysis

o What is the methodology for $d^{13}C$ analyses?

o What standards were used for $d^{13}C$ measurement?

**Answer:**

The $\delta^{13}C$ of $CO_2$ in the trace gas bags was introduced to Delta V Plus. Internal laboratory $CO_2$-inair standards were calibrated against calcium carbonate standards. The measurement of $\delta^{13}C$ in plants was based on vario PYRO cube elemental analyzer combined with ISOPRIME-100. Lab standards were injected every 12 samples for calibration with the long standard deviation of 0.2‰. The detailed information added in the text can be seen from the last answer.

Line 112

Be clear that samples collected for $d^{13}C$- $CO_2$ analyses are not the same samples as those from the continuous collection regime.

Be more precise than "in summer and winter, respectively." The samples were collected once a day during two 10 day periods in November 2014 and June 2015. Also:

- Note that these are the same collection periods for $d^{13}C$-cave air and –stream DIC

- Why were these time periods chosen?

- Why are there data gaps in the $d^{13}C$ data (e.g., DK air of Figure 3)?

**Answer:**

Yes, samples collected for $d^{13}C$-$CO_2$ analyses are not the same samples as those from the continuous collection regime.

We make the periods more specific. The main reasons to choose the periods in October 2014 are due to the rainfall events and the transitional time for cave $CO_2$ decreasing. In Figure 3, there are no data gaps for DK (we use MZ to refer it now), the problem in the figure is a mistake that the raw data are in wrong form, which results in no recognition. The figures can be seen in the following answers.

**4 Results**

Line 127

"Soil CO2" needs to be "Soil $CO_2$"

Line 129

Soil CO2 concentrations bottom out around 4000 ppm in November 2014

Line 130

Why do you compare soil CO2 concentration at your site to these other studies? Do they have similar climate and vegetation regimes?

Line 131

Be consistent in using "soil moisture" instead of "humidity."

Line 134

If soil moisture controls respiration when temperature is suitable, what is occurring in summer 2015?

It looks like there are time periods when $pCO_2$ is high but soil moisture is low (July-August).

**Answer:**

Line 127, corrected. "Soil CO2" needs to be "Soil $CO_2$".

Line 129, corrected, "The soil concentrations ranged from 4000 ppm in December to 17000 ppm in June with the mean soil $CO_2$ being 8890±4576 ppm."

Line 130, we just want to show the range of soil concentration in other places in the world.

Line 131, corrected, "soil moisture varied between 0.5% and 24.0% with the minima occurring in spring months (March 2015) and dry summer (July-August, 2015-2016)".

Line 134, we think that soil moisture is very important. Regarding to periods when $pCO_2$ is high but soil moisture is low, we would like to explain it because of time lag.

**Figure 2**

- Make the data gaps more obvious. Note in the text where these are and why they occurred

- The x-axis is difficult to read. Label it by month instead?

- Mark the $d^{13}C$ sampling intervals on here so it is obvious where to look for the 'zoomed-in' sections presented in Figures 3 and 4

- 2A

o I find the inverted y-axis confusing - precipitation should logically increase upwards

- 2B

o Include cave temperature on here as well (or at least the average)

**Answer:**

We updated the figure to make x axis much clearer and also to mark the intervals in Fig.3 and Fig.4. Normally, precipitation should logically increase upwards, we used inverted y axis just to save space. But this kind of layout can be found in climatic figures. Cave temperature is very stable, nearly horizontal.

[Figure]

Line 139

Rephrase "Cave parameters" to "Environmental measurements"

Line 140-141

- What are the "upper layer" and "lower layer?"

- The average cave temperatures are different from the average presented on line 93.

- Include cave temperature in Figure 2B

- What are the "three layers" – this is the first time this is mentioned in the text.

**Answer:**

Line 139, updated. We use "Environmental measurements".

Line 140-141, they are not actually "upper layer" and "lower layer?", two sites are located in the

deepest cave and the entrance of the cave, respectively. I want to say the upper stream and down

stream.

It's true that the average temperatures are different from the one presented in line 93. Because the previous one refers to the cave temperature and the latter refer to temperature of cave stream water. "The cave stream temperatures at LF (the innermost part in the cave) ranges from 16.0 °C to 18.7 °C with a mean value of 16.2±0.2 °C, while from 16.3 °C to 16.8 °C at MZ (the entrance part) with a mean value of 16.6±0.1 °C."

"three layers" –we explain in "Study area", "Most parts of the cave are narrow, deep passages (canyon passages), which are developed along strata and is composed of three levels of passages: at 233–236 m (Level I), 249–262 m (Level II) and 281–283 m (Level III), separately."

Line 147

Typo "stream000000"

Line 151

Does cave CO2 decrease to atmospheric levels? It looks like it does from Figure 2

Line 157

- Could low cave CO2 concentrations be related to effective transport of cave air to the outside?

- In any event, this kind of interpretation should be left to the Discussion section

Line 157-159

I'm not clear on the meaning of this sentence.

Line 162

What is "less variability?" Define this.

**Answer:**

Line 147, we cancelled 000000, which is additional inputs (a mistake), now as 'stream' in the textgg.

Line 151, cave $CO_2$ decreased in winter, but still three times higher than atmospheric levels.

Line 157, the transport of cave air to the outside could be a reason to explain the low concentrations of cave air $CO_2$. We will move this part to the discussion section.

Line 157-159, the meaning of this sentence is that there is seasonal variation of cave air $CO_2$, but the rainfall events could disturb the seasonality and also bring variations in cave air $CO_2$.

Line 162, "less variability?" means that variational magnitudes in soil $CO_2$ concentration are less than that in cave air and stream $pCO_2$.

**Figure 3**

Figures 3 and 4 should be combined for ease of reference

- Precipitation should increase upwards

- Precipitation should be black, as in the other diagrams

- The same materials (e.g., CO2-cave air and d$^{13}$C-cave air) should be the same line color and type

- Include error bars on d$^{13}$C measurements

- "LF" and "DK" are not defined before Figure 3. Where are these sites?

- Caption

o Rephrase to "during rainfall events in October-November 2014"

**Answer:**

We combined the Figure 3 and Figure 4, we also adjusted the colors. LF and DK labels have been explained in the method part and they are also labeled in Figure 1. The Caption was: "Variations of monitoring items (precipitation, temperature, $\delta^{13}$C and $p$CO$_2$) during rainfall events in October-November, 2014 and June 2015."

[Figure]

**Figure 4**

- The precipitation plot is labeled as air temperature

- Precipitation should increase upwards

- What are the high-frequency oscillations (6/29 and 7/22) in the cave temperature record? Were sensors replaced at this time?

- Where are the $d^{13}C$ measurements?

- The same materials (e.g., CO2-cave air and $d^{13}C$-cave air) should be the same line color and type

- Include error bars on $d^{13}C$ measurements

- Caption

o Rephrase to "during rainfall events in June-July 2014"

**Answer:**

The precipitation label has been changed in the above figure. The increasing temperatures (external air temperatures) during the period 6/29-7/22 are normal. The temperature in the cave is still stable (the discontinuous line). In the new figure, we only showed the period from 11$^{th}$, June to 24$^{th}$, June. Error bars of carbon isotopes were added. The caption was shown in the last Answer.

Line 168

Rephrase "4.4 The carbon isotope $d^{13}C$ in cave air and stream water" to "4.4 Carbon isotopes in cave air, stream water, and soil"

Line 169

Why cite Mattey et al. (2010) for atmospheric $d^{13}C$ measurements at the Rock of Gibraltar? There are long-term records of atmospheric CO2 that would be more directly relevant to your site

Line 170

- This is the first time that measurements of plant $d^{13}C$ are mentioned. Information about plant collection and measurement should go in the methodology

- What is the range of $d^{13}C$-soil CO2?

- Remind readers of the depth of soil CO2 collection as this is a critical parameter for interpretation.

Line 173

- Change "High-frequency" to "Daily".

- A decreasing then increasing trend is potentially seen in the 'DK water' data, but I do not see this

trend in any of the other samples

Line 174

- Where is data for low/high streamflow? It is not mentioned before this point

**Answer:**

Line 168, rephrased.

"4.4 Carbon isotopes in cave air, stream water, and soil"

Line 169, we cited the results of atmospheric $d^{13}C$ measurements at the Rock of Gibraltar from Mattey et al. (2010) for comparison. However, as you pointed out that it is not suitable to cite in the results part. We cancelled in the revised version.

Line 170, in the updated version, we put more information about the measurement in the method section.

The range of $d^{13}C$-soil $CO_2$ is from -18.0‰ to -23.9‰ at 40cm depth.

"Plant leaves were collected in summer and winter 2014. The measurement of $\delta^{13}C$ in plants was based on vario PYRO cube elemental analyzer combined with ISOPRIME-100."

Line 173, changed.

"Daily monitoring in November 2014 showed a decreasing trend and then an increasing trend of $\delta^{13}C$ values during rainfall events. The trend during the rainfall events in June 2015 was not significant."

- In November, a decreasing then increasing trend is significant in the 'DK water' and 'DK air' data (now we use MZ), but in 'LF water' and 'LF air', the increasing trend is still significant though the decreasing trend is not obvious at the beginning (See Figure 3).

Line 174, the stream information has been added in the 'Study area' section. The high flow is related to periods with more rainfall events, which result in large discharge.

"A cave stream flows at the bottom level (Pu et al., 2016). There is no allogenic stream sinking underground at the head of Xueyu Cave (Pu et al., 2015). The cave stream catchment is about 8–9 $km^2$. Previous investigations by Zhu et al. (2004) and Pu et al. (2016) have described the hydrogeological and hydrochemical functioning of the Xueyu Cave stream. The stream is the only entrance of Xueyu Cave with an explored length of 1644 m. The discharge of the underground river ranges from 4.1 L/s in dry period to 26.6 L/s in wet period, the velocity of stream flow is 0.27 m/s

on average."

**Figure 5**

- Plot needs error bars

- Why are the high-resolution measurement periods not shown?

- Where are sites LF and MZ? Specify the 'upstream' and 'downstream' locations

- This data needs to be reported in a table (or in the supplemental information)

**Answer:**

We cancelled this figure as we have discussed the daily details in the two intervals and it is not necessary to put monthly data.

**5 Discussion**

Line 183

Rephrase 'lighter $\delta^{13}$C to 'more negative $\delta^{13}$C.' Values cannot be lighter or heaver. See, for example, table 2.1 in Sharp's Stable Isotope Geochemistry (https://digitalrepository.unm.edu/unm_oer/1/). Fix throughout the manuscript.

Line 184

- The values for d$^{13}$C-cave air need to be reported in a table and the collection+analysis method need to be described in the Methodology

- 'cave air $CO_2$ decreased at the beginning of the rain and then increased during the process at DK site.' There does not appear to be a strong initial decrease in the 'LF air' data and the 'DK air' data do not cover the entire time period. I suggest incorporating these observations into your interpretation

- When does the rain event start? This could be stated clearly here and be shown more clearly (vertical dotted lines?) in the graphs

Line 185

As noted above, it is not clear where the DK and LF sites are. I will not note further instances, but this needs to be addressed for the whole paper.

Line 186

Define 'the variability of d$^{13}$C values'

Line 187

'the $d^{13}C$-DIC values of stream water at two sites decreased and then increased during the rainfall events.' Depending on exactly when the rainfall event occurred, this may be true for site MZ. However, I see no overall change in the values for site LF.

Line 189-191

This sentence is unclear and appears to contradict itself. Please clarify how you are interpreting the relationship between soil gas and cave air.

**Answer:**

Line 183, thanks for your recommendation. The sentence has been rephrased.

The sentence is "The interannual variability of carbon isotopes seems to be related to precipitation, resulting in more negative $\delta^{13}C$ values with more precipitation."

Line 184

-the background information of the collection and analysis of carbon isotopes has been put in the method section, which can be seen in answer for methodology.

-'cave air $CO_2$ decreased at the beginning of the rain and then increased during the process at DK (MZ) site.' There does not appear to be a strong initial decrease in the 'LF air' data and the 'LF water' data could be explained by the fractures that transport $CO_2$ in gas forms.

-In the above part, we explain why 'DK air' data do not cover the entire time period, just a mistake, the new Figure 3 presents the clear trend.

Line 185, sorry to bring so much troubles, we make it clear in the new Figure 1. LF and MZ sites are located in the in the deepest and the entrance of the Xueyu cave.

Line 186, 'the variability of $d^{13}C$ values' means the variational magnitude.

"Moreover, the variational magnitude of $\delta^{13}C$ values was larger at MZ than LF."

Line 187, we still support the idea that 'the $d^{13}C$-DIC values of stream water at two sites decreased and then increased during the rainfall events.' Because the $d^{13}C$-DIC decreased from -13.2‰ or -13.2‰ to -13.9‰, then increased back to 12.9 or 13.3‰. However, the trend is not so clear in the Figure 3(G) due to the large-range y axis. Actually, the variational changes of $d^{13}C$-DIC in LF and MZ are very similar.

Line 189-191, we think that stable carbon isotopes in cave air are very similar to that in soils. We

think that soil gas can enter the cave space directly to show similarity of carbon isotopic values between them.

**Figure 6**

- The y-axes on both plots should be the same to allow easy comparison

- The left plot has 'Steam CO2 degassing,' which should be 'St**r**eam'

- The 'Stream CO2 degassing' data reported in this figure appear to be $d^{13}$C-DIC values. Reporting these data as the $d^{13}$C of CO2 in equilibrium with stream DIC requires calculation of the fractionation factor between DIC and CO2

- Keep the order the same for all graphs. Show November and then June (June is shown first in Figure 6)

**Answer:**

We made the consistent y axes in both plots. The 'St**r**eam' has replaced the 'steam $CO_2$ degassing'. Other comments are all accepted for revised Figure 6, e.g. the values of $d^{13}$C-DIC in the figure contain the fractionation.

[Figure]

Line 200

'heavier $d^{13}$C' should be 'higher $d^{13}$C'

Line 200-202

- This sentence is unclear – is your intent to relate the $d^{13}$C of respired organic matter to $d^{13}$C of soil air CO2?

- Why are soil air measurements in Gibraltar relevant to your field site in SE China? Why not use your own measurements to make an estimate?

**Answer:**

Line 200, corrected. "Fractionation by diffusion in the pore space results in higher $\delta^{13}C$."

Line 200-202, We cited the results from Gibraltar just to compare with our data. But now we cancelled the sentence as we think it is not necessary to cite any other data.

Line 205

I have the following issues with the discussion section:

- Why is a 2-endmember mixing model appropriate for your conceptual model? Several of your citations suggest a simple 2-endmember mixing model is inappropriate for understanding changes in cave air.

o You consider CO2 contributions from soil, stream, and human breath

o However, your introduction considers these additional sources important: atmospheric CO2, organic matter decay in the cave, magmatic/metamorphic sources

- Atmospheric air appears to be a particularly important endmember that this model does not address. The authors need to revise their data analysis to incorporate all of the information available from the dataset conceptual model of how/why cave air CO2 changes

- If >75 % of cave air CO2 is from the soil, why is there much better seasonal correlation between CO2-cave air and CO2-stream? Do your results apply only to rain events or year round?

- What causes the overall U-shape in the cave air and stream CO2 data every summer? Again, if soil CO2 is controlling cave air CO2, why is this signal not visible in the soil CO2 data?

- It is unclear to me from the discussion whether you think the soil, stream, or both are controls on cave air CO2. However, in the conclusions you definitively identify soil contributions as most important. Your position should be made clearer and should be supported by the isotope and CO2 concentration data.

- You briefly describe that $d^{13}C$-DIC of the stream is controlled by flow rate (Line 174). Is there a relationship between stream flow rate and cave air CO2 or $d^{13}C$?

- The discussion repeats results and repeats itself in sections. It should be edited for clarity and structure. I suggest the following general structure:

o Interpretation of what is occurring at Xueyu Cave

o Comparison to other studies of this nature

o Implications for developing paleoclimate records from speleothems (here and elsewhere)

**Answer:**

Line 205

-Two endmembers are simple. Though we introduced more sources, but from the filed monitoring, the magmatic/metamorphic sources and human breath can be excluded. The magmatic/metamorphic sources have very positive isotopic values. We do not include the atmospheric $CO_2$ though it seems more reasonable to assume atmospheric $CO_2$ brings low $CO_2$ concentration in many caves. The reason is that if atmospheric $CO_2$ makes more contribution, the cave air $CO_2$ would show higher values of $d^{13}C$ and low $CO_2$ concentration in November (which can be seen in the Figure 6A). The human respiration has been excluded as the previous monitoring from Xu et al. found that there is no linear relationship between the number of tourists and the carbon concentrations in the cave air. At last, the proportion from organic matter decay in the cave can be neglected too. In the Xuyue Cave environment, the organic matters are not abundant, even if they work as a source, the contribution will be very limited.

- Normally, atmospheric air appears to be a particularly important endmember in other caves. However, the sharply decreasing trend did not go along with increased carbon isotopic values, showing no marked influence from atmospheric $CO_2$ that has low $CO_2$ concentration but higher isotopic values.

-75 % of cave air $CO_2$ is from the soil, which makes the background of cave air $CO_2$, the close relationship between $CO_2$-cave air and $CO_2$-stream can be consider as the equilibrium between the air and water. The contribution calculation is based on two rainfall events, which confirms the contributions from both stream degassing and the diffusion of overlying soil or fractures. Though our results are observed from rainfall processes, the implication can be extended. Rainfall processes can reveal the changes in short period, which allow us to explore what happens inside the system.

- The overall U-shape in the cave air every summer (April-November) can be considered as accumulation of $CO_2$, not only related to soil $CO_2$ source, but also the transport way and the cave geometry. We think that the abundant soil concentrations can be dissolved in the infiltration water and finally to increase $CO_2$ in the stream. The degassing of stream $CO_2$ as well as the gaseous soil $CO_2$ entering the cave through fissures and fractures accumulate in the cave. The U-shape in summer

can be also explained that most of karst spaces are filled with water, providing a relatively sealed environment. This environment maintains the stable $CO_2$ concentrations except during rainfall events. Soil $CO_2$ shows high correlations to soil temperature and soil moisture, no U-shape concentration.

- It is clear that the soil, stream, both are controlling on cave air $CO_2$ variations, which can be seen from the consistent variations of cave $CO_2$ and stream $CO_2$. And their contributions can be calculated from carbon isotopes.

-d$^{13}$C$_{DIC}$ variations are mainly controlled by sources, the interaction between water and rock. We do not know whether there is linear relationship between stream flow rate and cave air $CO_2$ because we did not measure corresponding flow rate.

-Thanks for your suggestion. The discussion not just repeats results, because we wanted to make the result part and the discussion part to be consistent. But we will adjust the structure to separate discussion well from the results:

5.1 Variations in cave air $CO_2$ in Xueyu Cave

We started with "The sharpness of the transitions during the seasons demonstrates that it responded immediately to the changes of external environments. The cave $CO_2$ values are comparable with the values in some other karst caves reported by Sánchez-Moral *et al.* (1999) for the Altamira Cave (6000 ppm), lower than extreme 60000 ppm (Benavente *et al.,* 2015). A high-frequency monitoring in November 2014 and June 2015 showed the detailed $p$CO$_2$ changes in the cave air and stream, and carbon isotopes during rainfall events. In November, low cave air $CO_2$ concentrations indicated the more open system for gas exchange or low-concentration recharge was predominant. During rainfall events in June, cave air $CO_2$ concentrations increased due to the increased high-$CO_2$ stream flow. Soil $CO_2$ concentrations did not show abrupt variations (Fig. 3). The similarity of variations in soil, cave air and stream $CO_2$ concentrations indicated that there are correlations between them. There are significant correltations between stream $p$CO$_2$ and cave air $CO_2$, especially at LF site (R$^2$=0.95, p<0.01). The correlation between soil $CO_2$ and soil temperature is significant too…"

"5.2 $\delta^{13}$C isotope tracing the sources and their contributions to cave air $CO_2$

The $\delta^{13}$C values of the cave stream generally showed seasonal fluctuations with the lowest values occurring in June and the highest values in November months (Fig. 5). It was consistent with

previous observation that winter samples with relatively low $p$CO$_2$ were isotopically heavy (Spötl *et al*., 2005). In October, the monitoring results of rainfall events showed that $\delta^{13}$C values of the stream DIC and cave air CO$_2$ decreased at the beginning of the rain and then increased during the process at MZ site (near the entrance of the cave). However, at LF site (upstream) those $\delta^{13}$C values were in an increasing trend (Fig. 3). Moreover, the variational magnitude of $\delta^{13}$C values was larger at MZ than at LF."

Line 211

Is d$^{13}$C -soil referring to soil organic matter or soil air CO2? If it refers to soil air CO2, keep in mind that d$^{13}$C -soil air CO2 changes with depth. Justify using a single value.

**Answer:**

d$^{13}$C-soil refers to soil air, we always collected samples at the same depth at 40cm.

Line 211-212

- A citation and explanation are needed for the 'd$^{13}$C-CO2 from degassing -21.4 per mil due to isotopic fraction of 8 per mil.' Converting from DIC to the CO2 in equilibrium with it is not a straightforward connection for unfamiliar readers

- 'fraction' should be 'fractionation'

**Answer:**

Line 211-212, more explanation about background of stream degassing has been added.

"The average values of $\delta^{13}$C$_{soil,}$ $\delta^{13}$C$_{DIC}$ in June were -23.9‰ and -13.4‰, respectively, the $\delta^{13}$C$_{CO2}$ from stream degassing -is 21.4‰ due to the fact that equilibrium isotopic fractionation of 8‰ is between liquid water and vapour at the air–water interfaces (DIC and CO$_2$) (Zhang et al., 1995)."

'fraction' was changed to 'fractionation'.

Line 213

I do not get the same output from your model using the values in Table 1

Line 214

Same as line 211-212 – a citation and explanation are needed for the fractionation between DIC-CO2

Line 219

'light d$^{13}$C$_{CO2}$' should be 'more negative d$^{13}$C'

Line 228

How does water degassing $CO_2$ not precipitate calcite?

**Answer:**

Line 213, we checked that there was mismatch between the Table and the text. We should have put the isotopic values of gas from stream not the d$^{13}$C$_{DIC}$ in aquatic form (considering the fractionation).

Line 214, changed, "The average values of $\delta^{13}$C$_{soil}$, $\delta^{13}$C$_{DIC}$ in November were -18.3‰ and -12.2‰, respectively (the $\delta^{13}$C$_{CO2}$ from stream degassing was -20.9‰ considering carbon isotopic fractionation of 8‰ between water and gas)."

Line 219, corrected, "The more negative $\delta^{13}$C$co_2$ in the Xueyu cave air are close to −23.3 ‰ in summer, , discarding the deep $CO_2$ and the human respired $CO_2$ as sources."

Line 228, normally, the degassing companies with precipitating calcite. We just said that the precipitation is not significant.

**Figure 7**

- How is this model different from those proposed/used by other you cite? Might be better just to cite/describe the model.

- I did not understand that the river flowed from inside the cave to outside the cave until this figure – this information should be up front in the study area description

**Answers:**

This figure was abstracted from our study area. We know that there are other models in previous studies, however, few figures with suitable streams. And we think this figure can help readers to understand the main text better. More information about the stream we also explained in the 'Study area' section.

Line 277-278

This sentence is unclear: what does 'resulting in warm surface air into the cave companying with rainfall events' refer to?

Line 283

The terms 'S-pCO2' and 'C-pCO2' are confusing. I recommend not using them

Line 287

Delete the final sentence of this paragraph.

**Answer:**

Line 277-278, we should have described it better, I mean that high-temperature water infiltration in summer always accompany with the rainfall events. "The stream running through Xueyu Cave increased its discharge dramatically in the wet season, companying with increased water temperature during storm events."

Line 283, we accept the comments to avoid using the 'S-pCO2' and 'C-pCO2'. Instead, we use $pCO_{2(cave\ air)}$ and $pCO_{2(stream)}$.

Line 287, the sentence has been cancelled.

Figure 8

- Mark months of the year on the x-axis, not the 20th of each month

- Mark when the cave switches between summer and winter modes

**Answer:**

We accept the suggestions to mark the months and transitions.

[Figure]

Line 290

This section largely repeats what has been already said

Line 303

Where is the CO2 data for stream water at the two LF and DK sites? We are only presented with one dataset

Line 305

- This section is difficult to understand. I'm not sure what I am supposed to get out of it.

- Define the metrics 'before rain' and 'after rain,' response time, intensity, and equilibrium time

Line 309

-'in consistent' should be 'inconsistent'

Lines 311-316 do not seem to add to the section. If you are reporting results, they should be in the Results section

**Answer:**

Line 290, we cancelled the repeated part.

Line 303, the pattern between the air and water is similar in LF and DK, that is why we only put one (LF site) to present the trend.

Line 305, we want to find if there is relationship between cave air $p\text{CO}_2$ the intensity and amount of rainfall events. 'before rain' and 'after rain' just the time that we collected the data. Response time and intensity refer to the lasting time of the rainfall events and their intensity, equilibrium time refers to the time it takes to get balance between the stream and cave air $\text{CO}_2$.

Line 309, accepted, 'inconsistent' in the revised version.

"The accumulated rainfall amount in 2015 was higher than that in 2016, which was inconsistent with the general increasing trend of cave air $p\text{CO}_2$ variations during the two years."

**6 Conclusions**

Line 331

Measurements were made two times in the year, not 'throughout the year'

Line 322

'$^{13}\text{C}$' should be '$\delta^{13}\text{C}$'

Line 333 The stated percentage contributions do not match Table 1

**Answer:**

Line 331, ok, we accepted this expression, that in two intervals not throughout the year. 'throughout the year' has been cancelled.

Line 332, corrected. "$\delta^{13}\text{C}_{\text{DIC}}$ showed higher values in winter but lower values in summer."

Line 333, the table and the text are checked to make sure that the information is consistent.

**Table 1**

- The time transgressive values do not give the reader the idea that a single value of d[13]C -soil CO2 is assumed for the whole time period.

- The table should include the d[13]C values for cave air, stream DIC, and calculated CO2 in equilibrium with stream DIC

**Answer:**

We added the value of d[13]C-soil $CO_2$ in the new table.

Table1 $\delta^{13}$C values from cave air and stream and the contribution of cave $CO_2$ from soils

| Time | Cave air (‰) | | Stream DIC (‰) | | Stream degassing (‰) | | The proportion from soils* (%) | |
|---|---|---|---|---|---|---|---|---|
| | MZ | LF | MZ | LF | MZ | LF | MZ | LF |
| 2014/10/30-09:00 | -18.2 | -19.1 | -10.6 | -12.9 | -18.6 | -20.9 | 59.2 | 63.9 |
| 2014/10/31-09:00 | -19.2 | -19.1 | -12.2 | -12.8 | -20.2 | -20.8 | 48.2 | 61.3 |
| 2014/11/1-09:00 | -19.0 | -19.2 | -12.2 | -13.0 | -20.2 | -21 | 56.2 | 60.7 |
| 2014/11/2-09:00 | -19.3 | -19.4 | -12.1 | -13.1 | -20.1 | -21.1 | 57.9 | 56.7 |
| 2014/11/3-09:00 | -19.1 | -19.1 | -12.6 | -12.6 | -20.6 | -20.6 | 60.1 | 56.9 |
| 2014/11/4-09:00 | -19.0 | -18.9 | -12.3 | -12.6 | -20.3 | -20.6 | 58.1 | 68.3 |
| 2014/11/5-09:00 | -18.3 | -18.5 | -12.1 | -12.5 | -20.1 | -20.5 | 84.4 | 82.6 |
| 2014/11/6-09:00 | -18.4 | -18.6 | -11.0 | -12.2 | -19 | -20.2 | 61.8 | 74.2 |
| 2014/11/7-09:00 | -18.4 | -18.4 | -11.7 | -12.3 | -19.7 | -20.3 | 75.5 | 82.8 |
| 2014/11/8-09:00 | -18.3 | -18.4 | -11.8 | -12.9 | -19.8 | -20.9 | 85.8 | 88.2 |
| Mean values | -18.7 | -18.9 | -11.9 | -12.7 | -19.9 | -20.7 | 64.7 | 69.5 |
| 2015/6/15-09:00 | -23.4 | -23.6 | -13.2 | -13.3 | -21.2 | -21.3 | 82.4 | 89.7 |
| 2015/6/16-09:00 | -23.3 | -23.2 | -13.4 | -13.9 | -21.4 | -21.9 | 77.9 | 68.6 |
| 2015/6/17-09:00 | -23.4 | -23.6 | -13.5 | -13.6 | -21.5 | -21.6 | 81.6 | 90.6 |
| 2015/6/18-09:00 | -23.4 | -23.4 | -13.9 | -13.8 | -21.9 | -21.8 | 79.3 | 79.4 |

| | | | | | | | |
|---|---|---|---|---|---|---|---|
| 2015/6/19-09:00 | -23.4 | -23.6 | -13.5 | -13.6 | -21.5 | -21.6 | 81.2 | 88.1 |
| 2015/6/20-09:00 | -23.4 | -23.3 | -13.0 | -13.2 | -21 | -21.2 | 85.2 | 81.7 |
| 2015/6/21-09:00 | -23.3 | -23.1 | -13.4 | -13.7 | -21.4 | -21.7 | 80.3 | 64.5 |
| 2015/6/22-09:00 | -22.7 | -23.2 | -12.8 | -13.5 | -20.8 | -21.5 | 63.7 | 71.9 |
| 2015/6/23-09:00 | -22.9 | -23.3 | -13.1 | -13.3 | -21.1 | -21.3 | 65.1 | 81.1 |
| 2015/6/24-09:00 | -23.4 | -23.3 | -12.9 | -13.3 | -20.9 | -21.3 | 86.0 | 77.0 |
| Mean values | -23.3 | -23.4 | -13.3 | -13.5 | -21.3 | -21.5 | 78.3 | 79.3 |

*The average $\delta^{13}C$ values of soils in November and June are 18.0‰ and 23.9‰ respectively.

**Table 2**

- This table does not mean much to the reader as the parameters are not defined (intensity, response time, equilibrium time)

- This table can be moved to the supplemental information

**Answer:**

We moved this part to the supplemental material.

**References:**

Baldini, J. U. L., Bertram, R. A., & Ridley, H. E., (2018). Ground air: a first approximation of the earth's second largest reservoir of carbon dioxide gas. Science of The Total Environment, 616-617, 1007-1013.

Pu, J., Wang, A., Shen, L., Yin, J., Yuan, D., & Zhao, H., (2016). Factors controlling the growth rate, carbon and oxygen isotope variation in modern calcite precipitation in a subtropical cave, southwest china. Journal of Asian Earth Sciences, 119(2), 167-178.

Xu, S.Q., (2013). Thesis of master, Southwest University, China.

---

## Author Comment (AC6) · 17 Jun 2019

**Response to Referee #1**

**General comments**

The review of the article "Constraining the soil carbon source to cave-air $CO_2$: evidence from the high-time resolution monitoring soil $CO_2$, cave-air $CO_2$ and its $\delta^{13}C$ in Xueyudong, Southwest China"by Min Cao, Yongjun Jiang, Jiaqi Lei, Qiufang He, JiaxinFan, Ze Zeng. The authors present the data on $CO_2$ in the soil, cave stream, and cave atmosphere (Xueyu Cave,China) and its surrounding. The data weregathered during the period of 2015-2016. The aim of the article is (1) to understand the quantitative relationship between all the forms of $CO_2$, (2) to reveal their sources, and (3) to understand the factors that control the cave air $CO_2$ variations. The topic of the article is important and is worthy of publication. In the article, however, there are some aspects that require revision and other ones that could be substantially improved before publishing. My main reservation is that the conclusions should be better proved by a data analysis (e.g., Cross-correlation Analysis). The results of the data analysis should be presented and discussed in detail. The data sets are nice, but they could be much better presented. The x-axis should be more extended in order to be better distinguishable individual fluctuations in the variables.

**Answer to general comments:**

We would like to thank the referee for his generally positive comments. We will pay more attention in presenting and explaining the our data in the final version.

We posted a table of the correlation analysis:

Table 1 The correlation matrix of environmental parameters in Xueyu system

| | Soil M | Soil T | Prep | Cave T | Soil CO$_2$ | Discharge | pH | MZ stream CO$_2$ | MZ air CO$_2$ | LF stream CO$_2$ | LF air CO$_2$ | Spc | TOC |
|---|---|---|---|---|---|---|---|---|---|---|---|---|---|
| Soil M | 1.00 | | | | | | | | | | | | |
| Soil T | .285** | 1.00 | | | | | | | | | | | |
| Prep | -.023** | -.013** | 1.00 | | | | | | | | | | |
| Cave T | .326** | .367** | -.040** | 1.00 | | | | | | | | | |
| Soil CO$_2$ | .263** | .639** | -.027** | .116** | 1.00 | | | | | | | | |
| Discharge | -.062** | .011* | .217** | -.122** | -.027** | 1.00 | | | | | | | |
| pH | .044** | 0.00 | -.094** | .296** | -.052** | -.278** | 1.00 | | | | | | |
| MZ stream CO$_2$ | .189** | .416** | .073** | -.192** | .294** | .224** | -.735** | 1.00 | | | | | |
| MZ air CO$_2$ | -.589** | .518** | .052** | -.795** | .683** | .222** | -.989** | .868** | 1.00 | | | | |
| LF stream CO$_2$ | .030** | .402** | .054** | -.237** | .263** | .304** | -.926** | .655** | .877** | 1.00 | | | |
| LF air CO$_2$ | -.030** | .423** | .059** | -.210** | .237** | .253** | -.904** | .768** | .963** | .952** | 1.00 | | |
| Spc | .134** | .227** | .077** | -.305** | .062** | .253** | -.740** | .610** | .957** | .749** | .710** | 1.00 | |
| TOC | .190** | -.540** | -.023** | -.447** | -.176** | -.046** | -.194** | -.596** | -.570** | -.080** | -.209** | .111* | 1.00 |

**. $P<0.01$; *. $P<0.05$; Soil M=Soil moisture, Soil T=Soil temperature, Prep=Precipitation, Cave T=Cave temperature

We updated the text in the dicussion part '4.3 Environmental parameters and their correlation':

"There are significant correlations between stream $pCO_2$ and cave air $CO_2$, especially at LF site ($R^2=0.95$, $p<0.01$). The correlation between soil $CO_2$ and soil temperature is significant too ($R^2=0.64$, $p<0.01$)."

We put more details in the study area part, the method part and discussion part, and refined the conclusions too:

" 1) Two-year monitoring study of soil $CO_2$, subterranean stream and cave air $CO_2$ concentration reveals that a dynamic equilibrium between $CO_2$ sources and sinks.   Seasonal dynamics took place with the minimum cave air $CO_2$ concentrations during winter and peaks in November.

2)High-resolution monitoring of $CO_2$ concentrations in the soil and cave system may allow us to estimate the potential cave air $CO_2$ sources in Xueyu Cave (subterranean stream, air from vadose/soil zone). Throughout the year, $\delta^{13}C_{DIC}$ showed higher values in winter but lower values in summer. $\delta^{13}C$ of different endmembers showed that soil $CO_2$ made more contribution of C to the cave air $CO_2$ in June (75.6%) than in November (65.9%), and the second source was the degassing of stream. The accumulation of cave air $CO_2$ concentration maintains the high values in summer due to the confined space.

3)The seasonal variations in cave air $CO_2$ concentration were very similar to that in stream $pCO_2$, showing which shows high but fluctuated values in summer and steady but low values in winter. Stream water seems to be a constant source of $CO_2$ as an increase of up to 5800 ppm in 2 hours was observed and $CO_2$ degassing occurred after strong rain events. In winter, stream water is the carbon sink of cave $CO_2$.

4)Cave air $CO_2$ concentrations are similar at different sites in Xueyu Cave. The anthropogenic impact of visitors to cave air $CO_2$ concentrations is evident from the hourly fluctuations, but not significant on daily or longer time scales."

**Other comments:**   Throughout the text, it is important to distinguish $CO_2$ itself from $CO_2$ concentration and $pCO_2$ (e.g., the lines/paragraph 85). The expression "$PCO_2$ in the water" (stream $pCO_2$ is acceptable only as an abbreviation in the text. Furthermore, it is important to explain that it means $pCO_2$ of gaseous $CO_2$ that would be in equilibrium with aqueous carbonates. In   principle, $pCO_2$ is dimensionless variables (or it has units of pressure). If the $CO_2$ quantity is given in ppmv units, it means "$CO_2$ concentration".

Some soil characteristics should be given in the paragraph Study Area. More detail information should be given in monitoring/calculating of the stream $CO_2$ in the paragraph Methods and Materials.

The x-axes in the plots (Fig. 2, 3, 4, 5) should be better divided (e.g., by one month, three months, etc.). The secondary y-axis in Fig. 4 should represent "Precipitation". I do not understand what the conceptual model in Fig. 7 brings new/beneficial. In the text, there are missing the citation: Liu and Zhao 2000, and Baker et al., 1998 and 2014, referenced in the Reference list.

**Answer to other comments:**

1. We checked the use of $CO_2$ itself and $pCO_2$, $CO_2$ concentration to make sure that they are expressed in the correct form in the revised text. Actually, $pCO_2$ has unit, such as Pa, but we use ppm in $CO_2$ quantity to make it simple and comparable between air and water.

2. In most cases, it is not in equilibrium with aqueous carbonates, which can be seen in Fig. 8.

3. The x-axis had been adjusted in order to be better distinguishable individual fluctuations in the variables. The figures 2 and 3 can be seen in the supplimentary material.

4. Regarding to the references, we checked all the manuscript to make sure that the citations in the maintext are all consistent with the ones in the reference list. References by Liu and Zhao 2000, and Baker et al., 1998 and 2014 were cited in the previous manuscript but then cancelled in the maintext without removing from the reference list.

5. In the revised version, we cancelled the following part:

Baker, A., Genty, D., Dreybrodt, W., Barnes, W. L., Mockler, N. J., and Grapes, J.: Testing theoretically predicted stalagmite growth rate with recent annually laminated samples: implications for past stalagmite deposition, Geochim. Cosmochim. Ac., 62(3), 393-404, https://doi.org/10.1016/S0016-7037(97)00343-8, 1998.

Baker, A. J., Mattey, D. P., and Baldini, J. U. L.: Reconstructing modern stalagmite growth from cave monitoring, local meteorology, and experimental measurements of dripwater films, Earth Planet. Sc. Lett., 392(392), 239-249, https://doi.org/10.1016/j.epsl.2014.02.036, 2014.

Liu, Z., and Zhao, J.: Contribution of carbonate rock weathering to the atmospheric $CO_2$, sink, Environ. Geol., 39(9), 1053-1058, https://doi.org/10.1007/s002549900072, 2000.

**Figure 1** (A) Chongqing Municipality, SW China and geographical location of study area (red shape), (B) Monthly air and precipitation in Xueyu Cave, (C) The location of the Xueyu Cave, its surrounding strata and the soil sampling site (modified from Wu et al. (2015)), (D) Sketch map of the Xueyu Cave and locations of the monitoring sites: X1 and X5 for cave air and stream $pCO_2$ monitoring.

**Figure 2** Cross section of Xueyu Cave passages and the sampling locations, Chongqing, SW China

(modified from Pu et al., 2016).

**Figure 3:** (a) Precipitation, (b) air temperature and soil temperature, (c) soil moisture, (d) $pCO_2$ values in the soil air, cave air and stream water of Xueyu system in the years 2015-2016.

**Figure 4** Variations of monitoring items (precipitation, temperature, $\delta^{13}C$ and $pCO_2$) during rainfall events in October-November, 2014 and June 2015

**Figure 5** Conceptual model for subsurface carbon cycling in Xueyu karst cave. $CO_2$ respired in soils is transported into caves by gaseous form or infiltrated in rainwater. Changes of ventilation patterns which might be correlated to soil moisture overlying can help to accumulate cave air $CO_2$ or make it dispersed in summer and winter. Sketch of the seasonally controlled airflow of the Xueyu Cave system and resulting in $pCO_2$ changes.

**Figure 6** The $pCO_2$ variability ($\theta(pCO_2)=pCO_{2(stream)}-pCO_{2(air)}$ in the Xueyu stream and cave air system

**Figure 7** The relationships between $\delta^{13}C_{V\text{-}PDB}$ and $1/CO_2$ during the occurring of rainfall events in November (A) and June (B)

Figure 1

[Figure]

| | |
|---|---|
| T₂l Middle Triassic Leikoupo Formation | Stratigraphic boundary |
| T₁j Lower Triassic Jialingjiang Formation | Fault |
| T₁f Lower Triassic Feixianguan Formation | River |
| P₃c-w Upper Permian series | Outlet of cave stream |
| P₁₊₂ Lower Permian series | Xueyu Cave outline |
| | Soil sampling site |

★ Monitoring sites for the cave air and stream

✳ Entrance of Xueyu Cave

Figure 2

[Figure]

Figure 3

[Figure]

Figure 4

[Figure]

Figure 5

[Figure]

Figure 6

[Figure]

Figure 7

[Figure]

---

## Author Comment (AC7) · 17 Jun 2019

**Response to Referee #3**

In this contribution, the authors measured partial pressure of cave and soil gas CO2 and DIC concentrations in a cave stream. Those measurements were complemented by temperature (cave, soil and atmosphere) and precipitation observations. The logging was performed for more than two years. Furthermore, they provide the stable C isotopic composition of various components (stream water, soil and cave air) for two shorter periods of time in lower resolution (10 days, one sample per day). The aim of this study is to investigate the major sources of $CO_2$ in the cave. Unfortunately, the manuscript is relatively difficult to read and I am not always sure, I understood what the authors wanted to say. In a possible revision, one focus should be on improvements with respect to the readability. Apart from this, I think, the study is not in a shape to be published. To my opinion, the text should be considerably improved before it could be considered for publication. There are many details, which should be improved and some sections should be reworked. However, I want to emphasize, that I think that a lot of the findings appear to not to be valid but that there are some potential within the data, to make it a quite nice contribution to the scientific community, which is worth to publish once the problems are solved, the text reworked and new ideas implemented.

**Answer:**

We would like to say thanks to the referee for all his valuable comments for this paper. We paid more attention to solve the problems about readability and the data explanation.

**Major points**

Throughout the text (e.g., line 9) when the authors write of 'p $CO_2$ of cave water'. To me, this sounds very sloppy and should be prevented. I mean, gaseous $CO_2$ (which can be well expressed as pCO2) is dissolved in water and then forms the different carbon species often referred to as dissolved inorganic carbon (DIC). The DIC cannot be expressed as pCO2. It is all dissolved and should be expressed in terms of concentrations. To my opinion the authors most likely want to express with their wording something similar to: 'the $pCO_2$ the cave water is in equilibrium with'. This wording is much more precise and should be applied throughout the whole text.

**Answer:**

$$PCO_2 = \frac{(HCO_3^-)(H^+)}{K_1 K_{CO_2}}$$

Yes, $pCO_2$ is assumed to be in equilibrium with the sampled waters by the equation

I do not understand the 'the $pCO_2$ the cave water is in equilibrium with'.

The abstract should be reworked. In line 11 you are arguing that pc of the cave air depend on wet and dry periods. But as I understood your data, the main influence is on the temperature. Only if temperature is warm enough, the wet/dry relationship is important. The last sentence of the abstract, appears weird for me as well after having read the whole paper. In this sentence, you are referring to stalagmite records and what your findings imply for those. However, in the manuscript you did not discuss the influence of the important cave processes on the stalagmite stable carbon proxy. So either you should delete this sentence or add some discussion about this in the text. I guess the latter would be the more valid approach, as this way your story will have more impact.

**Answer:**

Ok, we have to rewrite the abstract. The $CO_2$ production is mainly depending on temperature, but in hot summer, water is a very important factor to control it and the transport is also dominated by rainfall events. We wanted to make it more meaningful with the last sentence but it seems that it lacks some correlation. In the revised version, we cancelled it.

"Cave $CO_2$ plays an important role in carbon cycle in a karst system, largely influencing the formation of speleothems in caves. Gaseous $CO_2$ and aqueous $CO_2$ in Xueyu Cave were monitored from October 2014 to February 2017. The cave air $pCO_2$ and aqueous $CO_2$ over two years showed very similar variations in seasonal patterns, with high fluctuated $CO_2$ concentrations in the wet season and low steady $CO_2$ concentrations in the dry season. Soil $CO_2$ which is largely controlled by soil temperature and soil moisture as well as stream degassing are main origins for the Xueyu cave air $pCO_2$. The average values of $\delta^{13}C_{soil}$, $\delta^{13}C_{DIC}$ in June were -23.9±2‰ and -13.4±0.3‰, respectively; $\delta^{13}C_{CO2}$ of atmospheric air was -10.0‰ and $\delta^{13}C_{CO2}$ of cave air was -23.3±0.3‰. The average values of $\delta^{13}C_{soil}$, $\delta^{13}C_{DIC}$ in November were -18.0±0.5‰ and -12.2±0.4‰, respectively; $\delta^{13}C_{CO2}$ of atmospheric air was -9.6‰ and $\delta^{13}C_{CO2}$ of cave air was -18.8±0.4‰. Moreover, the contribution from soil $CO_2$ is higher in June (78.8±13.0%) than in November (67.1±6.8%) based on the model of stable carbon isotopes. The contribution of C from the soil was larger in summer than in winter. The very similar (negative) values of carbon isotopes between

soil and cave air $CO_2$ suggests that there were no potential geological/deeper sources which show more positive $\delta^{13}C_{CO2}$. Aqueous $CO_2$ degases from upper stream to downstream in the cave, resulting in slightly decreased $pCO_2$ but increased carbon isotope values in the downstream."

As the stream seems to play an important role in the cave, I think it is important to let the reader know some more details. E.g., for which distance the stream is in contact with the cave atmosphere, how much water is transported, how fast is its velocity and something like that. This could be well included in Section 2. This is important, as this way other researchers investigating other caves can set their observations in better context and comparison with your results is easier. I recognized that the cave is ascending into the host rock. However, with respect to cave $CO_2$ and the derived ventilation regime, it behaves, according to your data, like

those caves which are going downwards into the rock. This is quite interesting and could be much more emphasized. Maybe the stream is the key for this behavior? You should put some efforts here and elaborate more on this.

**Answer:**

Thanks for your suggestions, we added more information about the stream in the 'Study area' section. There are three lays in the cave space, which can be highly related to stream flow in different geological periods. In modern time, the stream flows through in the bottom layer. In the revised version, we will put more information about the geology and topography.

"Most parts of the cave are narrow, deep passages (canyon passages), which are developed along strata and is composed of three levels of passages: at 233–236 m (Level I), 249–262 m (Level II) and 281–283 m (Level III), separately. A cave stream flows at the bottom level (Pu et al., 2016). There is no allogenic stream sinking underground at the head of Xueyu Cave (Pu et al., 2015). The cave stream catchment is about 8–9 km². Previous investigations by Zhu et al. (2004) and Pu et al. (2016) have described the hydrogeological and hydrochemical functioning of the Xueyu Cave stream. The stream is the only entrance of Xueyu Cave with an explored length of 1644 m. The discharge of the underground river ranges from 4.1 L/s in dry period to 26.6 L/s in wet period, the velocity of stream flow is 0.27 m/s on average."

Section three should be completely reworked, as some information are given twice (e.g., about the CO2-

sensor; Line 111 and 116) and others incomplete (How was soil $CO_2$ measured? How was the sampling performed for soil gas d13C analysis? How was the pCO2, the stream water is in equilibrium with, determined?) or no information at all (e.g., about measurements of cave and soil temperature, soil moisture, d13C values of plants [according to line 170 they have been measured]).

**Answer:**

We were planning to introduce the equipment in Line 111 and present the monitoring frequency in Line116.

"A set of system for continuous and automatic soil $CO_2$ measurement with a $CO_2$ sensor was fixed in the soils above Xueyu Cave (Fig.2). The soil temperature and soil $CO_2$ concentrations were obtained from October 2014 by a composite measurement system, including a $CO_2$ sensor (GMM221, made by VAISALA in Finland with the resolution of 1 ppm and the range of 0-20000 ppm) and temperature and humidity sensor (AV-10T and AV-EC5 produced by AVALON, U.S.A with a resolution of 0.1 °C and 0.1%). All sensors were imbedded into the soil at the depth of 40 cm by drilling in the soil sampling site which is located about 40 m on the top with an elevation of about 300 m a.s.l. and 400 m a.s.l. in horizontal distance from the entrance of the cave (Fig.2). Above the cave, soils range from 0 to 50 cm in thickness. These soils are stony clays-rich and yellow soils that support evergreen forest and grainland (Field survey)."

"Two sites inside the Xueyu Cave for $p$CO$_2$ monitoring of cave air and the subterranean stream have been located at LF (X1) and MZ (X5), respectively (Fig. 1). The data were recorded each quarter based on a GMM221 sensor (within the range 0~20000 ppm, precision ±1%) connected with RR-1008 data receiving terminal, which were installed in the cave and stream to measure $CO_2$ concentrations. The sensor in the soil was calibrated on a monthly basis by the portable equipment that can insert into the soil for $CO_2$ measurement (portable pump-suction infrared $CO_2$ gas detector, measuring range $20 \times 10^{-6} \sim 20,000 \times 10^{-6}$ ppm). Cave air $CO_2$ was determined with a calibrated CDU 440 $CO_2$ meter (measuring range $10 \times 10^{-6} \sim 20,000 \times 10^{-6}$ ppm with the resolution of 10 ppm, made by Industrial Scientific, Pittsburgh, PA, USA). The sensor in the stream was difficult to be calibrated by another equipment, but logging data have been compared with $p$CO$_2$ that was calculated by hydrochemical parameters. To obtain the detailed hydrochemical variations, a CDTP300 multi-parameter water quality meter (made in Greenspan Corporation in Australia) was installed to record water temperature, water level, Ec and pH with

resolutions of 0.01 °C, 0.01cm, 0.01 µS/cm and 0.01 pH units."

Within the manuscript you are claiming: 'When the temperature is suitable in summer, soil moisture works as the main constraining factor for variations in soil CO2.' (e.g., Line 134). First, please don't discuss this in the results, but put it to the discussion and please be more detailed here. By eye, it is easily visible, that soil gas $CO_2$ might be changed by soil moisture during the warm season. However, there seem to be a clear time delay between soil moisture and pCO2. I feel, you have to discuss this. Where is this delay coming from? Please, do some statistics, e.g., calculate correlation coefficient, determine time lag. You might be even able to make some statements about the sensitivity of soil $CO_2$ variations with respect to soil humidity under various temperature ranges, as soil $pCO_2$ seems also to react on soil humidity under lower temperatures. However, the sensitivity appears to be different when the variations in the cold and warm seasons are compared. This all, will have some influence on section

**Answer:**

Ok, we will separate results and discussion. Yes, there is a time lag we did not pay more attention to. We have monitored but not put in this paper, the relationship between soil temperature, soil moisture and soil $CO_2$ on daily or seasonal scales.

Fig. 6: What exactly is plotted for the stream water? The measured d13C, ok. But howhave you determined the x-axis value? Is this the $pCO_2$ value, the stream water is in equilibrium with? But not all of this C can degas from the stream (due to the chemical limits of degassing) and thus cannot contribute to the cave air CO2. So the $pCO_2$ what comes from the stream is for sure lower than the $pCO_2$ value the water is in equilibrium with. Thus, those values should be put more to the right (the question is how much?).

Due to all this, I think, providing this stream C source in the Keeling plot this way is quite bold without arguing why this can be done. This is then changing the available $CO_2$ which can degas and is thus also changing the according mCi in your equation (line 205). As those plots contain the basis of your argumentation according to your present discussion, a change here, might have major influences on your final findings.

**Answer:**

In Fig. 6, We updated the figure. The x-axis refers the inverted $CO_2$ concentration. The stream water is in equilibrium with $pCO_2$ value. Yes, not all of this C can degas from the stream (due to the chemical limits of degassing) and thus cannot contribute to the cave air $CO_2$. However, there is no need to degas all, when it comes to be in equilibrium, no more aqueous $CO_2$ sources can transfer to gaseous $CO_2$.

The more positive values in the downstream (MZ) can indicate that degassing had occurred. Because more $d^{13}C$ was going to precipitate, and more negative $d^{13}C$ will degas into air. We use the Keeling plot to show that the contributions might be from soil and stream. Why these two sources? We excluded human respiration, deep source and atmospheric air according to their concentrations or carbon isotopes.

**Technical comments:**

Line 26: 'always with higher values in summer and lower values in winter' I want to point out that this is not true. This is only true for caves, which lead downwards with increasing distance from the entrance. For those caves, which leads upwards, the opposite will be observed. But you have indicated this correctly, e.g., in line 43. So please, be consistent and more precise here.

**Answer:**

Thanks for your comments, that is true, what we said is just correct for some caves, especially the caves from the cited study. However, different cave modes show different circulation/ventilation characteristics, resulting in variations of cave $CO_2$.

"$CO_2$ concentrations in temperate karst soils range from 1000 to 15000 ppm, showing higher values in summer and lower values in winter in some caves (Spötl *et al*., 2005; Frisia *et al*., 2011)."

Line 27-28: Unfortunately, I do not understand this sentence. Can you reword this?

Line 32: It seems to me, that 'although' is not fitting here.

Line 64: 'carbon' with a small first letter.

Line 65: 'Especially, in descending caves where carbon dioxide is heavier than the other main atmospheric components : : :' I am pretty sure you are meaning this differently than stated here. CO2 is always heavier than the main atmospheric components (O2 and N2), not only in descending caves. Maybe reword this to something like that: 'As carbon dioxide is heavier than the other main atmospheric components, CO2 accumulates in descending caves during the hot season due to the "cold trap effect".

**Answer:**

Line 27-28, this sentence was going to explain how soil $CO_2$ goes into the cave system.

"The $CO_2$ inputs that penetrate caves and become part of the karstic atmosphere via directly in gaseous form, dissolved $CO_2$ in infiltrated waters from the soil matter" was changed into "Soil $CO_2$ can directly enter the cave by soil gaseous form or be dissolved in the water and move with the infiltration water'.

Line 32, 'Although' was cancelled.

"A few studies revealing very high $CO_2$ concentrations exist in deep and confined karst caves."

Line 64, 'Carbon' has been changed to 'carbon'.

"According to their study, carbon dioxide produced by the soil biomass is accumulated into underground voids due to gravitational drainage from cracks and fissures. As carbon dioxide is heavier than the other main atmospheric components, $CO_2$ accumulates in descending caves during the hot season due to the "cold trap effect".

Line 65, Ok, we accepted your expression, which is much clearer.

Line 73: The 'Thus' does not seem to fit here, as the previous sentence does not provide a reason for what you are stating in the following sentence, which you begun with 'Thus'.

Line75: Please define R/Ra. What is this?

Line 76-77: Please be more precise and reword this sentence as 222Rn is not produced from the decay of U and other radioactive atoms as you have stated here. It would be more precise to write something similar to: '222Rn is a radioactive isotope that is naturally produced within the 238U decay chain. As it is heavier than air it is accumulating in the ….

**Answer:**

Line 73, cancelled. "Soil is commonly considered as the light end-member, and this assumption is correct as long as the $CO_2$ concentration in the cave is lower than the soil (Peyraube *et al.*, 2013)."

Line 75, R/Ra should be '$^{226}Ra/^{228}Ra$', which is the isotopic ratios of Ra.

"Moreover, the $^{226}Ra/^{228}Ra$ versus $\delta^{13}C_{CO2}$ plot, traditionally used to estimate crustal versus mantle components of $CO_2$."

Line 76-77, new sentence is like this: "$^{222}Rn$ is a radioactive isotope that is naturally produced within the $^{238}U$ decay chain. As it is heavier than air it is accumulating in the subterranean atmosphere and usually covaries with $CO_2$ concentration."

Line 90: 'multiyear average precipitation' sounds a bit strange. Do you mean 'average annual precipitation'?

Line 91: Is the term 'secondary speleothems' correct? It seems to me, you might mean 'secondary carbonates' instead?

Line 93: Please provide information, of how constant cave air T is. Give its variation throughout the year and plot it maybe even in Fig. 2

**Answer:**

Line 90, Yes, it should be expressed as 'average annual precipitation'.

"This region has a typical subtropical monsoon climate with an average annual precipitation of approximately 1072 mm."

Line 91, Ok, normally, speleothems are secondary carbonates.

"The geological formation and secondary carbonate deposits"

Line 93: We added the cave temperature in the Fig.3

Fig. 1: It is not clear, why the monitored drip sites are shown in d), as these are not discussed in the text. In addition, I am very confused about the naming of your river drip-sites. Here in Fig. 1 they are labelled as X1 and X5. In Tab. 1 they are labeled MZ and LF while throughout the text and in the figures they are labeled DK and LF. Please stay consistent.

**Answer:**

We corrected in the revised version. X1 and X5 are LF and MZ (DK). We did not discuss drip water, so we have to remove the drip water sites in the Fig.1d.

Line 103: 'cave' appears to be unnecessary here.

Line 105: Please, replace ',' by '.' after ')'.

Line 106: 'drip rate' instead of 'drip water rate'.

Line 126: 'amount' instead of 'amounts'

Line 128 and 131: Here you quite often make some statements, which should be carefully discussed before. To my opinion those statements do not belong in the results section, but should be shifted towards the discussion section.

Line 133-134: This belongs also to the discussion and it is not quite clear, what 'suitable' is meaning

here. Please, be more precise. I also wonder if this statement is justified (see above).

**Answer:**

Line 103, cancelled, the new sentence "The relationships between specific conductance (Spc), $Ca^{2+}$ and $HCO_3^-$ have been established and variations of $CO_2$ concentrations in the cave atmosphere and cave stream showed different changes in wet and dry season due to the ventilation".

Line 105, 106, 126, corrected. The changed sentences are

"(Pu *et al*., 2015, 2018)."

"Seasonal variations of calcite growth rate are primarily controlled by variations of cave air $pCO_2$ and drip rate".

"During 2015-2016, the mean annual rainfall amount was 1149 mm, …"

Line 128-131, Line 133-134, we will separate better the results without comments.

Fig. 2: Please indicate the cave air temperature.

**Answer:** we added the cave air temperature in new Figure 3

Line 137: Capital letters of 'Air' and 'Soil' should be small ones.

Line 146-147: Sorry, but I do not understand this sentence. Could you rewrite this under the correction of the 'stream00000' typo.

Line 156-163: This should be shifted to the discussion part. And I do not understand the sentence in Line 157. Also the following sentence (line 157-159) is not clear to me.

Line 169: Typo. Should be '-10 permil', shouldn't it?

Line 170: Please include here, that you are talking about soil gas, which has values of -18 permill to -23.9.

**Answer:**

Line 137, corrected. "**Figure 3:** (a) Precipitation, (b) air temperature and soil temperature, (c) soil moisture, (d) $pCO_2$ values in the soil air, cave air and stream water of Xueyu system in the years 2015-2016."

Line 146-147, we rewrote the sentence "According to Pu et al. (2018), the seasonality of cave $CO_2$ variations occurred based on monthly monitoring, showing high values in summer and low values in winter."

Line 156-163, we moved this part to the discussion part and improved like this "During the transitional

periods between different seasons, $CO_2$ concentration varied sharply. Especially with rainfall events, cave $CO_2$ concentrations changed largely due to increased high-$CO_2$ flow. A high-frequency monitoring in November 2014 and June 2015 showed the detailed changes of $pCO_2$ and carbon isotopes during rainfall events."

Line 169, corrected. "The $\delta^{13}C$ values of background atmospheric $CO_2$ ranged from -9.6 to -10.0‰."

Line 170, yes, we were talking about soil gas. It is described as "During the two monitoring intervals, $\delta^{13}C$ values were -18.0‰±0.8‰ in the overlying soil gas in October-November but -23.9±0.9‰ in June on average."

Line 183-185: I am sorry. From my side, this does not look as contemporaneously as you described it. Please provide some shaded rectangles in the appropriate figures to allow to better follow your argumentation. For me it even seems, that there is no rain event in October, which was also covered with DIC and pCO2 measurements at DK. In November are some rain events, but there, I could not see the described relationship. DK-d13C is already decreasing before the rain event. Thus, it appears the decrease in d13C has nothing to do with the single rain event. And to be honest, the observation of such behavior (if there would be indeed one) during only one single event is not very convincing.

**Answer:**

Line 183-185, there was a little rain in October, but we mainly monitored the rainfall event at the end of October. Of course, it is not so convincing with one/two rainfall events. However, it needs a lot of work to monitor many rainfall events. At present, we have monitored several rainfall events. However, we do not have enough samples for analysis of carbon isotopes. The decreasing trend of $CO_2$ concentrations are significant in the season transitional period.

Line 186-188: Please also show this in a figure. With only seeing the numbers in the table, I am not able to follow.

Line 205: Please subscript the 'i' in the equation.

Line 207-209: Why has the mixing model only two endmembers? What is with the atmospheric input? Why can this be neglected. Please explain. But then, an endmember modelling with three sources is much more difficult. (even with the assumed two end-members, the equation in line 205 needs an additional equation to order to be solved. [Sum of all mCi = 1])

Line 211-212: Please, cite the work you are taking the fractionation factor from.

**Answer:**

Line 186-188, It is in Figure3

Line 205, 'i' in the equation was edited by equation editor, now we used the non-text expression.

$$\delta^{13}C_{CO2} = \sum_{0}^{i} (mC_i)(\delta^{13}C_i) / \sum_{0}^{i} (mC_i)$$

Line 207-209, We excluded other endmembers. Atmospheric input is not considered as a direct input due to the carbon isotope. If atmospheric input lowers the winter $CO_2$ concentration, the cave $CO_2$ concentration and isotope should have closed to atmospheric end. We excluded human interference and geological sources based previous field exploration, monitoring and also the stable carbon isotopes.

Line 211-212, cited. "Zhang, J., Quay, P.D., Wilbur, D.O.: Carbon isotope fractionation during gas–water exchange and dissolution of $CO_2$. Geochim. Cosmochim. Acta, 59, 107-114, https://doi.org/10.1016/0016-7037(95)91550-D, 1995."

Line 212-216: Even with your two end-member modelling, I am somewhat confused, about your numbers for the contribution of the C sources to the cave air. There is quite some scatter in the $d^{13}C$ of soil gas, cave air and DIC (or degassed $CO_2$) of the stream. The problem then is that the difference between stream degassed $CO_2$ and soil air is quite small compared to the scatter in your values. This makes the calculation quite difficult. At least, I would require to give some error estimates in your values here.

Line 230: Why are you mentioning roots here. Are there roots in your cave? I assume not, otherwise I would have expected some description of them earlier in Section 2. So it is not clear to me, why you are mentioning them here with respect to other caves. This sentence is also completely out of context with the sentence before and after.

**Answer:**

Line 212-216, yes, the calculation is difficult as the $d^{13}C$ values from stream degassing and soil are not largely different. However, they are not the same even considered errors. We put the error zone in the figure 3.

Line 230, yes, no obvious roots were found in the cave, we removed this part.

Line 280: Mawmluh cave was not investigated by Riechelmann et al., 2017. They investigate Bunker Cave. Please change the citation accordingly.

Line 292: Why do you say that cave CO2 and stream DIC are in equilibrium with each other? For me this looks quite different in Fig. 8. There is barely a time, when the difference between both is 0. Even if this would be the case, all your argumentation from above that the stream is a significant CO2 source is destroyed by this sentence. If both would be in equilibrium, no CO2 can be degassed from the stream and could not provide CO2 to the cave atmosphere. And if all the CO2 of the stream would have been degassed earlier, you could also not do any end-member modelling as you do not have the initial conditions of the water.

Line 308: Please define and explain, what the response time is. How have you calculated

this? Can you show a plot of the observed relationship?

**Answer:**

Line 280, corrected. "Increased $CO_2$ absorption from the slow-moving or stagnant cave air by carbonate weathering and potentially the stream might explain the low cave-air $pCO_2$, which is similar to the observation in Bunker Cave of India (Riechelmann *et al*., 2017)."

Line 292, we said they are in equilibrium because the variations in aquatic and air $CO_2$ are very similar in Figure 3. Actually, they are not in equilibrium, we can see that they fluctuated within 1000 ppm in most of time, resulting in degassing or absorbing. We can say that degassing occurred during the two intervals, but we do not know if it is true throughout the year. However, we can make sure that there is exchange between air and water during the monitoring intervals. We think that aqueous and gaseous $CO_2$ should be always in dynamical balance.

Line 308, the response time means that the time it takes to be in equilibrium. We can see rainfall events make the large difference of $CO_2$ concentrations between air and water, degassing does not always take place until its concentration become stable.